# Mineralogical Diversity of Ca₂SiO₄-Bearing Combustion Metamorphic Rocks in the Hatrurim Basin: Implications for Storage and Partitioning of Elements in Oil Shale Clinkering

**Ella V. Sokol [1], Svetlana N. Kokh [1,\*], Victor V. Sharygin [1,2] , Victoria A. Danilovsky [1], Yurii V. Seryotkin [1,2], Ruslan Liferovich [3], Anna S. Deviatiiarova [1], Elena N. Nigmatulina [1] and Nikolay S. Karmanov [1]**

1. V.S. Sobolev Institute of Geology and Mineralogy Siberian Branch Russian Academy of Sciences, 3 Koptyug Avenue, Novosibirsk 630090, Russia
2. Department of Geology and Geophysics, Novosibirsk State University, Novosibirsk, 2 Pirogov Street, Novosibirsk 630090, Russia
3. MAPEI Corporation, 1144 E Newport Center Dr, Deerfield Beach, FL 33442, USA
* Correspondence: s.n.kokh@gmail.com or zateeva@igm.nsc.ru; Tel.: +7-383-330-21-49

**Abstract:** This is the first attempt to provide a general mineralogical and geochemical survey of natural Ca₂SiO₄-bearing combustion metamorphic (CM) rocks produced by annealing and decarbonation of bioproductive Maastrichtian oil shales in the Hatrurim Basin (Negev Desert, Israel). We present a synthesis of data collected for fifteen years on thirty nine minerals existing as fairly large grains suitable for analytical examination. The Hatrurim Ca₂SiO₄-bearing CM rocks, which are natural analogs of industrial cement clinker, have been studied comprehensively, with a focus on several key issues: major- and trace-element compositions of the rocks and their sedimentary precursors; mineral chemistry of rock-forming phases; accessory mineralogy; incorporation of heavy metals and other trace elements into different phases of clinker-like natural assemblages; role of trace elements in stabilization/destabilization of Ca₂SiO₄ polymorphic modifications; mineralogical diversity of Ca₂SiO₄-bearing CM rocks and trace element partitioning during high-temperature–low-pressure anhydrous sintering. The reported results have implications for mineral formation and element partitioning during high-temperature–low-pressure combustion metamorphism of trace element-loaded bituminous marine chalky sediments ("oil shales") as well as for the joint effect of multiple elements on the properties and hydration behavior of crystalline phases in industrial cement clinkers.

**Keywords:** cement clinker; Ca₂SiO₄ modifications; larnite; flamite; ye'elimite; mayenite; trace element partitioning; threshold limits; combustion metamorphism; Hatrurim Formation

## 1. Introduction

High-temperature calcic combustion metamorphic (CM) rocks, ubiquitous in the Hatrurim Formation (so-called Mottled Zone) in Israel (Judean and Negev Deserts) and Jordan (Transjordanian Plateau), are strikingly similar in mineralogy to industrial cement clinkers with high trace-element (Zn, Cd, U, Ni, Cr) content [1–8]. In this paper, the formation of phases and element partitioning in natural high-temperature–low-pressure metacarbonate CM rocks akin to clinkers are studied in the 4 Ma to 100 Kyr geological structures of the Hatrurim Basin (Table 1) used as test sites, with the natural analog approach successfully tested in the Mottled Zone [5,9–14].

In clinker production, which is a key step in cement manufacturing, inorganic raw materials are annealed at temperatures of 1300–1500 °C. Many primary input materials have been substituted in the last two decades by variable proportions of secondary raw materials and alternative fuels with elevated trace element (TE) contents. The clinker production involves carbonate-marl rocks, clay, and bauxite, as well as metallurgical wastes rich in P, Cr, Cu, Zn, Cd, Pb, As and/or Ni [15–19]. Heavy metals are incorporated in cement clinker during the cement kiln co-processing of hazardous wastes either by formation of accessory phases ($PbSO_4$; $(Mg,Zn,Ni,Cu)O$; Cr- and Ni-bearing spinels and others) or by cation substitutions into the structure of main clinker-forming minerals ($Cd^{2+} \rightarrow Ca^{2+}$; $Ni^{2+} \rightarrow Mg^{2+}$; $Ni^{2+} \rightarrow Ca^{2+}$; $Ni^{2+} \rightarrow Fe^{2+}$; $Cr^{3+} \rightarrow Al^{3+}$; $Zn^{2+} \rightarrow Al^{3+}$). Their further immobilization during cement hydration and concrete aging is mainly due to the presence of a hydrated phase of low solubility [5,14,17–19].

Prediction of element partitioning between rock-forming and accessory clinker phases in systems of complex compositions is an essential issue of modern material science. Clinker mineralogy is commonly investigated in experiments with a limited number of well characterized phases [20–23], whereas the joint effect of multiple elements on the stability, melting point, and hydration behavior of crystalline clinker phases is mostly complicated and analytically challenging to unravel. Gross [1] and Kolodny [2], who were the first to study the Hatrurim rocks, noticed their similarity with cements and concretes in the 1970s.

Due to the recent progress in analytical facilities, sixty nine new mineral species have been discovered altogether since 2004 in typical clinker-like larnite-bearing CM rocks from different Mottled Zone areas, out of which fifteen species have been approved by IMA as new minerals [24]. The Hatrurim Basin hosts numerous CM outcrops where all members of the perovskite–brownmillerite family have been found, including six mineral species that belong to the perovskite–brownmillerite pseudo-binary series, with four new minerals among them: shulamitite ($Ca_3TiFe^{3+}AlO_8$), sharyginite ($Ca_3TiFe_2O_8$), nataliakulikite ($Ca_4Ti_2FeFeO_{11}$), and $Fe^{3+}$-rich perovskite $CaTi_{1-2x}Fe_{2x}O_{3-x}$ ($x < 0.25$) [25–28]. The accessories include recently discovered vorlanite ($Ca(UO_4)$, cubic) and vapnikite ($Ca_3UO_6$, $P2_1/n$) [29,30], nabimusaite ($KCa_{12}(SiO_4)_4(SO_4)_2O_2F$) [31], silicocarnotite ($Ca_5((SiO_4)(PO_4))(PO_4)$) [32], and harmunite ($CaFe_2O_4$) [33]. Other accessory minerals are periclase, magnesioferrite, barite, eucairite ($CuAgSe$), berzelianite or bellidoite ($Cu_2Se$), and a number of sulfides, including oldhamite and K-bearing species [4].

The mineralogy and geochemistry of $Ca_2SiO_4$-bearing CM rocks have never been characterized comprehensively. All cited publications dealt with a limited number of samples, or even with single holotype samples, and often focused on specific minerals rather than considering a whole assemblage. At the time of writing, larnite–ye'elimite rocks from Har Parsa locality (Hatrurim Basin, Negev Desert) remain the only documented natural analog of belite sulfoaluminate cement clinkers [4]. We are trying to bridge the knowledge gaps with this study, which is the first attempt to provide a geochemical survey of natural $Ca_2SiO_4$-bearing CM rocks produced by annealing and decarbonation of oil shales in the Hatrurim Basin. It encompasses analytical data we collected for fifteen years on thirty-nine minerals (Table 2) existing as fairly large (10–50 μm) well crystallized grains suitable for measuring the contents of their major and trace elements. Relative percentages of minerals have implications for the mineralogical diversity of clinker derived from compositionally complex protolith/raw mixtures, as well as for the influence of mineralogy on trace element partitioning.

This study covers the following issues:

- major- and trace-element compositions of $Ca_2SiO_4$-bearing CM rocks and their sedimentary protoliths;
- mineral chemistry of rock-forming phases in the natural analogs of cement clinker;
- accessory mineralogy in CM rocks;
- incorporation of heavy metals and other trace elements (TE) into different phases of clinker-like natural assemblages;
- role of trace elements in stabilization/destabilization of $Ca_2SiO_4$ polymorphic modifications;

- mineralogical diversity of $Ca_2SiO_4$-bearing CM rocks and TE partitioning during high-temperature–low-pressure anhydrous sintering.

**Table 1.** Minerals in Hatrurim Fm Ca-rich combustion metamorphic (CM) rocks (Israel) and their cement chemist notation (CCN) notations.

| Mineral | Chemical Formula | Symmetry | CCN Notation |
|---|---|---|---|
| Silicates | | | |
| Hatrurite | $Ca_3SiO_5$ | trigonal | $C_3S$ (Alite) |
| Flamite | $\alpha'\text{-}Ca_2SiO_4$ | orthorhombic | $C_2S$ (Type I belite) |
| Larnite | $\beta\text{-}Ca_2SiO_4$ | monoclinic | $C_2S$ (Type II belite) |
| Calcioolivine | $\gamma\text{-}Ca_2SiO_4$ | orthorhombic | $\gamma\text{-}Ca_2SiO_4$ |
| Rankinite | $Ca_3Si_2O_7$ | monoclinic | $C_3S_2$ |
| Pseudowollastonite | $\alpha\text{-}Ca_3Si_3O_9$ | monoclinic | $\alpha\text{-}CS$ |
| Wollastonite | $\beta\text{-}Ca_3Si_3O_9$ | triclinic | $\beta\text{-}CS$ |
| Parawollastonite | $Ca_3Si_3O_9$ | monoclinic | $CS$ |
| Bredigite | $Ca_7Mg(SiO_4)_4$ | orthorhombic | $C_7MS_4$ |
| Merwinite | $Ca_3Mg(SiO_4)_2$ | monoclinic | $C_3MS_2$ |
| Gehlenite | $Ca_2Al_2SiO_7$ | tetragonal | $C_2AS$ |
| Polyanionic minerals | | | |
| Nagelschmidtite [a] | $Ca_7Si_2P_2O_{16}$ | hexagonal | $C_7S_2P_2$ |
| Spurrite | $Ca_5(SiO_4)_2(CO_3)$ | monoclinic | $C_5S_2\overline{C}$ |
| Ye'elimite [b] | $Ca_4Al_6O_{12}(SO_4)$ | isometric | $C_4A_3\overline{S}$ |
| Cuspidine | $Ca_4(Si_2O_7)F_2$ | monoclinic | $C_4S_2\overline{F}_2$ |
| Fluorapatite | $Ca_{10}(PO_4)_6F_2$ | hexagonal | $C_{10}P_6\overline{F}_2$ |
| Fluorellestadite | $Ca_{10}[(SiO_4)(PO_4),(SO_4)]_6F_2$ | hexagonal | $C_{10}S_3\overline{S}_3\overline{F}_2$ |
| Ternesite (sulfospurrite) | $Ca_5(SiO_4)_2(SO_4)$ | orthorhombic | $C_5S_2\overline{S}$ |
| Silicocarnotite | $Ca_5[(SiO_4),(PO_4)](PO_4)$ | orthorhombic | $C_5SP_2$ |
| Fluormayenite–Fluorkyuygenite | $Ca_{12}Al_{14}O_{32}[(H_2O)_{0-4}F_2]$ | isometric | $C_{12}A_7H_n\overline{F}_2$ |
| Aluminate | | | |
| Grossite | $CaAl_4O_7$ | monoclinic | $CA_2$ |
| Multiple oxides | | | |
| Srebrodolskite | $Ca_2Fe_2O_5$ | orthorhombic | $CF$ |
| Brownmillerite | $Ca_2(Fe_{2-x}Al_x)O_5$ | orthorhombic | $C_4AF$ |
| Shulamitite | $Ca_3TiFe^{3+}AlO_8$ | orthorhombic | $C_3FTA$ |
| Sharyginite | $Ca_3TiFe_2O_8$ | orthorhombic | $C_3F_2T$ |
| Perovskite | $CaTiO_3$ | orthorhombic | $CT$ |
| Magnesioferrite | $MgFe^{3+}_2O_4$ | isometric | $MF$ |
| Spinel | $(Mg,Fe)Al_2O_4$ | isometric | $MA$ |
| Simple oxide | | | |
| Periclase | $MgO$ | isometric | $M$ |

[a] nagelschmidtite formula according to [20]; [b] symmetry of $Ca_4Al_6O_{12}(SO_4)$ polymorphic modification according to single-crystal determination. CAS notation for sharyginite according to [34], all other notations according to [35,36].

**Table 2.** Assemblages of primary minerals found in Hatrurim larnite-bearing CM rocks (Israel).

| Mineral/Sample | Larnite β-Ca$_2$SiO$_4$ | Flamite α'$_L$-Ca$_2$SiO$_4$ | α'$_H$-Ca$_2$SiO$_4$ | Gehlenite | Mayenite Group | Ye'elimite | Fluorellestadite | Fluorapatite | Silicocarnotite Ternesite | Barite Hashemite | Gazeevite | Brownmillerite Srebrodolskite | Shulamitite Sharyginite | Perovskite Nataliakulikite | Fe–Mg Spinel | Magnesioferrite Magnetite | Periclase | Bredigite | Other Primary Minor and Accessory Minerals | Secondary Phases |
|---|---|---|---|---|---|---|---|---|---|---|---|---|---|---|---|---|---|---|---|---|
| Y-5-1 | ▲57.8 | □0.3 | □1.0 | ●4.4 | | ●6.8 | ●3.2 | ▲21.4 | | □tr | □0.4 | | | | | ●4.2 | | | Vrl | □0.3 |
| Y-6-3 | ▲68.7 | | | | | ●8.5 | □1.9 | | | □0.1 | □1.2 | □1.3 | □1.6 | | □tr | ●9.6 | | | Lak, Hem, Arc, Anh | ●7.2 |
| Y-8 | ▲ | | | | | ▲ | ● | ● | | □ | □ | ● | □ | | | ● | | | Mer, Hem | □ |
| M5-30 | ▲ | | | | ● | ▲ | ● | | ▲ | | | ▲ | | | | | □ | | | □ |
| M5-31 | ▲ | | | | ▲ | ● | | ▲ | ▲ | □ | | ▲ | □ | | ▲ | ▲ | | | Vrl | □ |
| M4-217 | ▲ | | | | | ▲ | ▲ | | | | | ▲ | | | | | ● | | | □ |
| YV-412 | ▲ | | | | | ▲ | ▲ | | | | | ▲ | | | | | ● | | Berz/Bell, Euc, Vrl | □tr |
| YV-411 | ▲ | | | | | ▲ | | ▲ | | ● | | | ● | | | ● | | | | □tr |
| YV-410 | ▲ | | | ▲ | | ● | | ▲ | | | | ▲ | □ | | | □ | | | Hat, Po | □tr |
| H-201 | ▲ | | | | | ▲ | ● | ▲ | | | | ▲ | ● | | | ▲ | | | | □ |
| MP-10-1 | ▲ | | | | | ▲ | | | | | | ▲ | ● | | ● | | | | | □tr |
| Y-10-5 | ▲23.2 | | | ▲53.0 | | | | □2.9 | | | | | | □0.2 | | ●3.0 | | ▲11.5 | Po, Andr | ●6.2 |
| W-11-3 | ▲32.4 | □1.6 | □2.4 | ▲45.1 | | | ●6.0 | | | □0.1 | | | | □0.6 | | ●5.6 | | | | ●6.5 |
| Y-7 | ▲43.2 | ▲15.2 | □1.3 | □0.7 | ●6.0 | □tr | ▲10.6 | | | | | ▲10.5 | | | | | | | | ▲11.8 |
| Y-9-1 | ▲73.3 | | | □1.5 | □1.2 | □0.5 | ●8.6 | □1.8 | | □0.1 | □0.3 | | □1.4 | | □tr | □0.3 | | | | ▲11.4 |
| G-7 | ▲51.4 | ●4.7 | ●3.0 | ▲17.7 | □2.5 | | | ●3.0 | | | | | □4.7 | ●6.5 | | ●7.1 | | | Vrl, Cpp | □tr |
| W-10-2 | ▲73.6 | | | ●8.7 | ●9.9 | | | □1.5 | | □tr | | □2.9 | □tr | □0.1 | □1.3 | □tr | □tr | | Vrl | □1.9 |
| W-12-1 | ▲ | | | | ▲ | | | | | □ | □ | ▲ | | | | □ | | | | □tr |
| CONCR | ▲ | □ | □ | ▲ | | | | □ | | | | ▲ | | | ▲ | ▲ | | | | □ |
| M5-32 | ▲ | □ | □ | | ▲ | | | □ | | | | ▲ | □ | | | | | ● | Vrl | □ |
| M4-215 | ▲ | | | | ▲ | | | □ | | | | | □ | ● | ▲ | | | ● | | □ |
| M4-218 | ▲ | | | | ▲ | □ | | □ | | | | | ▲ | | ▲ | ▲ | | ● | Old, Po, K-Fe-sulf | □ |
| M4-251 | ▲ | □ | □ | ▲ | ▲ | | | | | | | | □ | | ▲ | ▲ | | | | □ |
| Y-2-1 | ▲ | | □ | ▲ | | | | | | | | ▲ | | | | □ | | ● | Cus | □tr |
| H-401 | ▲ | ▲ | □ | ▲ | ▲ | | | □ | | | | | | | ▲ | ▲ | □ | | Hem | □ |

New data of this study (Rietveld analysis of powder XRD, SEM) and published evidence [4,25–27,32]. ▲ main phase (>10 vol. %); ● minor phase (3–10 vol. %); □ accessory phase (<3 vol. %); trace phase (<0.1 vol. %); digits show average results of Rietveld analysis. Mineral names are abbreviated as Vrl = vorlanite; Lak = lakargiite; Hem = hematite; Arc = arcanite; Anh = anhydrite; Mer = merwinite; Hat = hatrurite; Berz/Bell = berzelianite/bellidoite; Euc = eucairite; Po = pyrrhotite; Cpp = chalcopyrite; Old = oldhamite; K–Fe-sulf = K–Fe–Cu–Ni-sulfide; Cus = cuspidine; Andr = Ti-rich andradite.

## 2. Analytical Techniques

About ninety bulk samples of various $Ca_2SiO_4$-bearing CM rocks were collected in the course of field trips to the Hatrurim Basin (Negev Desert) from 2004 through 2007. After the initial scrutiny, twenty-five most representative and fresh samples were selected for detailed phase and chemical analyses (Tables 2 and 3). All selected samples were analyzed by X-ray powder diffraction (XRD), scanning electron microscopy (SEM), and electron probe microanalyses (EPMA), methods at the Analytical Center for multi-elemental and isotope research SB RAS, Sobolev Institute of Geology and mineralogy (IGM), Novosibirsk, Russia. The bulk major- and trace-element compositions of the rocks were determined by atomic emission spectrometry with inductively-coupled plasma (ICP-AES) on an Intertechs IRIS Advantage atomic emission spectrometer (ThermoJarrell Intertechs Corporation, Atkinson, WI, USA), to an analytical precision of ~10–15%. The preconditioning procedure included fusion of powdered whole rock samples with lithium borate as in [37]. The concentrations of Fe(II), $H_2O$ and $CO_2$ were determined by wet chemistry [38]. Trace elements in some samples were analyzed additionally at the Siberian Synchrotron and Terahertz Radiation Centre (SSTRC) at the Budker Institute of Nuclear Physics (Novosibirsk) using precise synchrotron radiation X-ray fluorescence analysis (SR XRF) with energy-dispersion spectroscopy (EDS), at 23 kV and 42 kV excitation energies following [39].

Main and minor mineral phases (≥3%) were identified by XRD in samples crushed and pulverized under isopropanol. After overnight drying and homogenization, the powders were front-loaded in a dimpled sample holder and surface-finished with a glass slide. All specimens were analyzed on a DRON-3 diffractometer, using bulk CuK$\alpha$ radiation with $\lambda$ = 1.54178 Å. Scans were recorded from 6 to 60° 2$\theta$ at 0.05° 2$\theta$ increments with 5 s scanning time per step.

Double-polished thin sections (20–30 μm) of $Ca_2SiO_4$-bearing rocks were prepared using the standard protocol (see [4] for details), with petroleum as coolant/lubricant to accommodate the preparation of hydrophobic materials. The thin sections were finished by polishing using 0.25 μm diamond paste. Microfabric and phase distribution were assessed by thin section optical petrography in both transmitted and reflected light, as well as by scanning electron microscopy. The polished sections were sputter coated with ~30 nm carbon (C) for EPMA and for SEM elemental mapping.

In addition to XRD, the minerals, including accessories, were identified according to energy-dispersive spectra (EDS), back-scattered electron (BSE) images, and elemental maps (EDS system), using a MIRA 3MLU scanning electron microscope (TESCAN, Brno, Czech Republic) equipped with an INCA Energy 450 XMax 80 microanalysis system (Oxford Instruments, High Wycombe, UK), at IGM. The instruments were operated at an accelerating voltage of 20 kV and a beam current of 1 nA in low- (40–60 Pa) or high-vacuum modes. The results were checked against reference standards of simple compounds and metals for most of the elements.

Electron probe microanalyses (EPMA) using the wavelength-dispersive (WDS) mode were performed using a Camebax Micro (Cameca Ltd., Gennevilliers, France) microprobe at IGM. The operating conditions were: 1–2 μm beam diameter, 20 kV accelerating voltage, 20 nA (silicates) and 50–60 nA (opaque minerals) beam currents, and count time 20 s (10 s peak and 10 s background). For details of standards chosen for different systems, see [4,8,25,26,40]. Matrix correction using the PAP routine was applied to raw data prior to recalculation into major oxides. The analytical accuracy was better than 2%-relative for >5 wt % elements and about 5%-relative for ≤2 wt % elements and F.

Nine typical samples of $Ca_2SiO_4$-bearing rocks were studied by quantitative X-Ray analysis at the MAPEI Corporation (Newport, RI, USA). The samples were immersed in ethanol and powdered using a Retch PM-200 planetary ball mill. Step-scanned powder X-ray diffraction (XRD) patterns were obtained at room temperature in a spinning zero-background holder (specially-cut silicon monocrystal) with a 10-mm diameter central well. To minimize preferred orientation effects, the exposed side of samples was textured using a fine grained sand paper (SiC 1200 grit, ~15 μm). Scans were performed in 5–80° 2$\theta$ range using a X'Pert Pro diffractometer (PANalytical, Almelo, the Netherlands) (Ni-filtered CuK$\alpha$ radiation; Δ2$\theta$scan step 0.0167°; time per step 100 s; 4608 steps/scan).

**Table 3.** Secondary mineral assemblages in Ca$_2$SiO$_4$-bearing CM rocks (Hatrurim Basin, Israel).

| Mineral/Sample | Calcite | Portlandite | Hydrogarnet | Hydrocalumite | Hillebrandite | Straetlingite | Afwillite | Ettringite | Apophyllite-(OH) | Chlorkyuygenite | Gibbsite | Other |
|---|---|---|---|---|---|---|---|---|---|---|---|---|
| Y-5-1 | | | □0.3 | | | | | | | | | |
| Y-6-3 | □0.8 | | □2.6 | | □0.8 | | ●3.0 | □(Cr) | | | | Chromatite |
| Y-8 | | | | | | □ | | | | | | |
| YV-412 | | | | | | | | | | | □tr | |
| YV-411 | | | | | | | | | | | □tr | Tobermorite |
| YV-410 | | | | | | | | | | | □tr | Tobermorite |
| Y-7 | | | ●7.8 | | □2.2 | □1.8 | | | | | | |
| Y-9-1 | | | ●8.3 | □1.3 | | | | | □1.8 | □tr | | |
| G-7 | | | | | □tr | □tr | | | | | | |
| W-10-2 | □0.2 | □1.0 | □0.4 | | □0.3 | □tr | | | | | | |
| W-12-1 | | | | | □ | | | | | | | |
| M4-218 | | □ | □ | | □ | □ | | | | | | Foshagite, Hem |
| Y-2-1 | | | | | □ | □ | | | | □tr | | |
| Y-10-5 | | | ●5.2 | | □0.7 | | | | | | | Hem□0.3 |
| W-11-3 | □0.7 | □0.7 | ●3.0 | | □1.9 | | □0.2 | □tr | | | | |

No secondary phases have been identified in samples M5-30, M5-31, M4-217; CONCR; M5-32, M4-215. Hem = hematite. ● minor phase (3–10 vol. %); □ accessory phase (<3 vol. %); trace phase (<0.1 vol. %).

The XRD patterns were inspected using the PANalytical Highscore Plus software, version 3.0c, coupled to crystallographic databases ICSD FIZ Karlsruhe 2011-1 [41,42] and COD [43] to identify the phases present in the test samples. The XRD-based identification agreed perfectly with the SEM and EPMA data. Following the identification of minerals in the samples, the X-Ray scans were further analyzed by the Rietveld technique [44] using the same software for mineral quantification. Quantitative analysis was performed semi-automatically using the fundamental parameter Rietveld refinement [44].

Single-crystal X-ray study of fluorellestadite grains was carried out using an Xcalibur Gemini diffractometer (MoK$\alpha$ radiation, 0.5 mm collimator, graphite monochromator, $\alpha$ scan with step of 1°, 22 s per frame) (Oxford Diffraction, Abingdon, UK). Data reduction, including the background correction and Lorentz and polarization corrections, was performed with the CrysAlis Pro 171.38.43 program package. A semi-empirical absorption correction was applied using the multi-scan technique. The structure was solved and refined with the SHELX program package [45]. All atoms were refined with anisotropic displacement parameters.

Ye'elimite crystals were studied using a STADI IPDS 2T single-crystal diffractometer (MoK$\alpha$, $\alpha$ scan with step of 1°, 1200 s per frame) (STOE & Cie GmbH, Darmstadt, Germany). The collected diffraction data were processed with the CrysAlis Pro 171.38.43 program package. They were used to determine unit-cell parameters and space group symmetry, but their quality was too low for crystal structure refinement.

## 3. The Hatrurim Formation

### 3.1. General Information

The terms Mottled Zone (MZ) sequence [46] or the Hatrurim Formation [1] refer to a specific non-stratigraphic unit which lies 30–40 to 120 m below the surface at the top of the Upper Cretaceous-Low Paleogene section of restricted areas in the vicinity of the Dead Sea-Jordan Transform. It mainly consists of diverse sedimentary rocks affected by post-depositional alteration and unevenly distributed brecciated sediments (mostly chalks enclosing phosphorite, chert, and marl) heavily cut by veins of calcite, aragonite, gypsum, zeolite, ettringite, and various Ca silicate hydrates (CSHs), with multiple foci of diverse Ca-rich anhydrous high- to ultrahigh-temperature CM rocks, frequently with clinker-like mineralogy, at shallow depths [1,47–49].

The rocks produced by highest-temperature annealing and complete decarbonation of carbonate and marly sediments ($T$ = 1000–1500 °C) can be grouped into two main groups: (1) products of solid-state reactions, such as hornfelses with pyroxenes and plagioclases (after marly sediments) and $Ca_2SiO_4$-bearing rocks (after chalky sediments); (2) diverse molten rocks (paralavas). The high-temperature CM rocks are mainly found at the top of the Mesozoic sequence west of the Dead Sea rift, in Negev and Judean Deserts [1,4,25,26,32,40,47,50–59]. The CM rocks in the Transjordanian Plateau farther in the east are mostly medium-temperature ($T$ = 700–850 °C) spurrite marbles, while the $Ca_2(SiO_4)$-bearing rocks and paralavas are rare [3,5–8,13,57,60].

The Hatrurim Fm CM rocks have often been interpreted as products of in situ combustion of low-calorific fuel, specifically, disseminated bituminous matter of marine chalk [6–8,47–49,61,62]. This interpretation appears reasonable for most of CM areas in the Transjordanian Plateau, which are mainly composed of spurrite marbles [5–8]. The higher-temperature CM overprints in the western side of the Dead Sea Transform may rather result from local breakthrough of hydrocarbon gases, mainly methane, possibly associated with dormant mud or gas volcanism [4,25–27,50,51,54,63–65]. Once exposed to air, methane reacts with atmospheric oxygen and ignites spontaneously. The burning of high-calorific fossil fuel released enough heat to maintain high-temperature CM alteration and even resulted in local melting of marly sediments. Past emissions of hydrocarbon gases left their imprint as abundant foci of high- (800–1100 °C) and ultrahigh-temperature (1200–1500 °C) combustion metamorphism of sediments in the Hatrurim Basin and elsewhere in the Negev and Judean Deserts.

According to $^{40}$Ar/$^{39}$Ar, K/Ar and $^{230}$Th–$^{234}$U dating of different MZ rocks [11,66,67], fluid flow and further gas ignition occasionally occurred between 16 and 0.2 Ma (mainly in the interval 7–0.5 Ma), which is the time span of the most active Dead Sea rifting [68,69]. Despite the controversy about their genesis, all CM rocks of the Hatrurim Basin and other MZ localities are considered as products of high-temperature (800–1200 °C) solid-state reactions during organic matter combustion, sometimes with subsequent local melting (1250–1500 °C).

## 3.2. Geological Background

The Hatrurim Formation (Mottled Zone) complexes undoubtedly belong to the tectonic system of the Dead Sea-Jordan Transform [11,46,65,68,69], which is a prominent shear zone separating the Arabian plate from the Sinai microplate (Figure 1). Neotectonic deformation was accompanied by expulsion of hydrocarbon-rich fluids, which was likely a response to the activity of the Red Sea and Levantine Transform faults [65]. The Dead Sea basin is the largest rhomb-shaped depression bordered by uplifts and filled with up to 10 km thick Miocene to Holocene sediments [67,68]. The Upper Cretaceous–Lower Paleogene marine sediments cover the area almost continuously and consist of limestone, chalk (including bituminous varieties), and dolomite with marl, phosphorite, and chert intercalations [70]. The faulted and fractured damage zones of the Dead Sea-Jordan Transform are meters to hundreds of meters wide and control fluid flow ascending from the subsurface [69,71,72]. In the Late Cenozoic (≤6 Ma), the area was favorable for occasional blowouts of methane venting through deformed sediments. The ongoing activity shows up as methane outbursts from shallow wells in the southern Dead Sea, saline springs, and famous asphalt floating blocks and 'springs' in the southern Dead Sea [73].

Variations in the temperature of CM alteration revealed for the Mottled Zone complexes may result from asymmetrical distribution of fluid flows from the Dead Sea basin, with migration of waters, brines, hydrocarbon gases, and micro-oil mostly to the west, toward the Mediterranean Sea [72]. Consequently, the contribution of high-calorific methane emanations into the heat budged of CM events was much greater in the western rift side than in the east, where combustion metamorphism was mainly maintained by hot gases released from sintering disseminated organic matter in chalks. The hydrocarbon fluid flows rising along nearly vertical faults from the Dead Sea gas generation zone discharged within the faulted Masada-Zohar tectonic block (Figure 1).

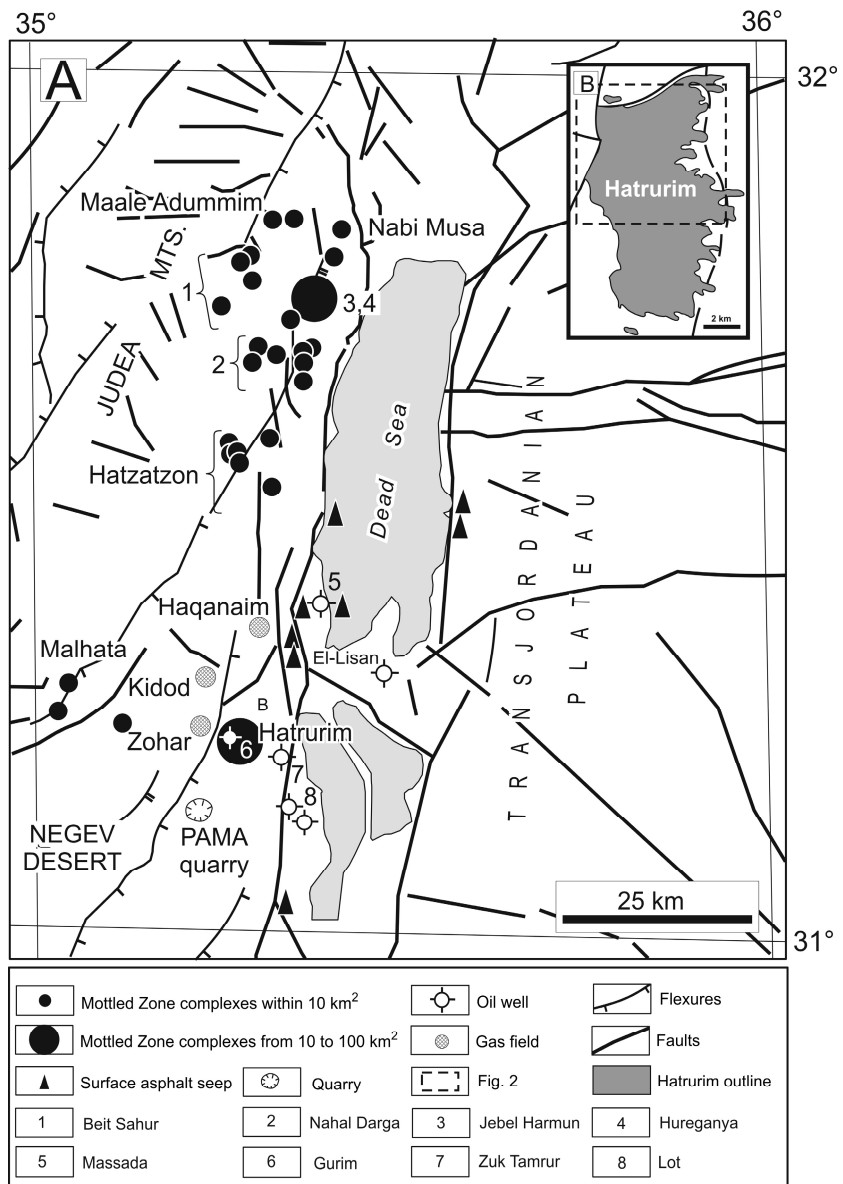

**Figure 1.** (**A**) Generalized geology of the Dead Sea Transform area, with locations of Mottled Zone complexes, gas fields, asphalt shows, and oil wells, after [71,73,74]. (**B**) Hatrurim Basin, simplified after [74].

### 3.3. Combustion Metamorphic Rocks within the Hatrurim Basin

The Hatrurim Basin complex located within the Masada-Zohar block is the largest one in the Hatrurim Formation (Mottled Zone) and is its stratotype [48]. It occupies 47.8 km$^2$ (11.3 × 7.3 km) in the southwestern Dead Sea rift shoulder at 31°12′N, 35°16′E (Figure 1) and is surrounded by tens of small isolated fields in the northern Negev Desert [75]. The Masada-Zohar block stores small commercial gas fields and non-commercial heavy and light oil. The geological descriptions of different MZ complexes and the models explaining their origin have been largely published [1,4,25,40,47–51, 69,74]. The Hatrurim CM complex is remarkable among other MZ in the abundance of high- and ultrahigh-temperature rocks, including diverse paralavas, as well as in the presence of multiple outcrops of Ca$_2$SiO$_4$-bearing clinker-like CM rocks (Figures 2 and 3). All these rock varieties are jointly called larnite rocks in the literature [1,4,31,48,49,64], though they may include other Ca$_2$SiO$_4$ modifications (see below).

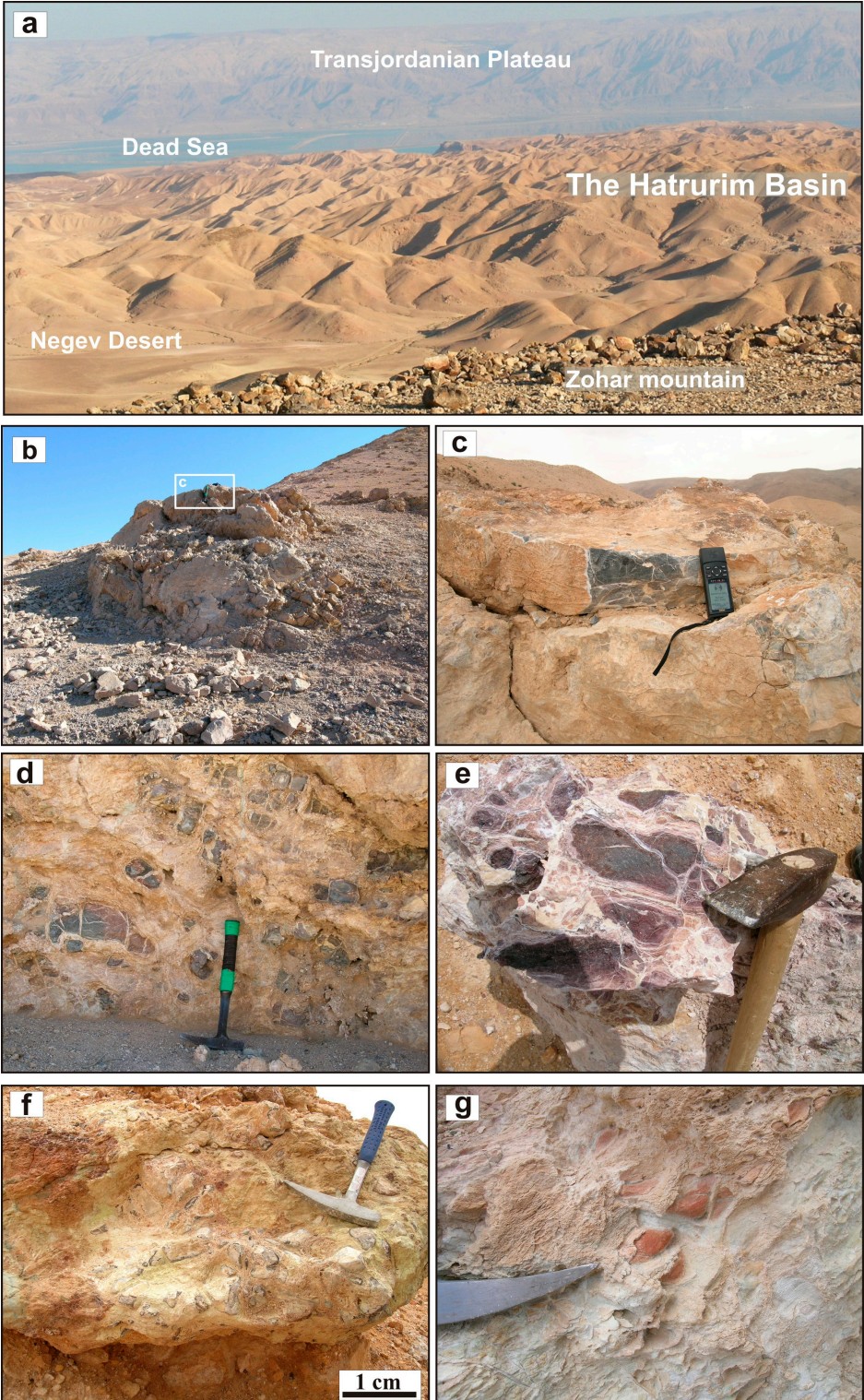

**Figure 2.** Larnite-bearing CM rocks from the Hatrurim Basin. Field images. (**a**) Panoramic view of the Hatrurim Basin. (**b**) Outcrop of larnite CM rocks on a hilltop. (**c**) Massive fresh larnite rocks. (**d**–**g**) Typical appearance of pseudo-conglomerates consisting of larnite- and/or gehlenite-bearing "pebbles" in a light-colored matrix of secondary minerals (mainly calcite, aragonite, gypsum, and ettringite, with lesser percentages of tobermorite, jennite, afwillite, and hydrogarnets). Photographs are ordered according to increasing alteration degrees of rocks. Photographs of 2005 and 2007.

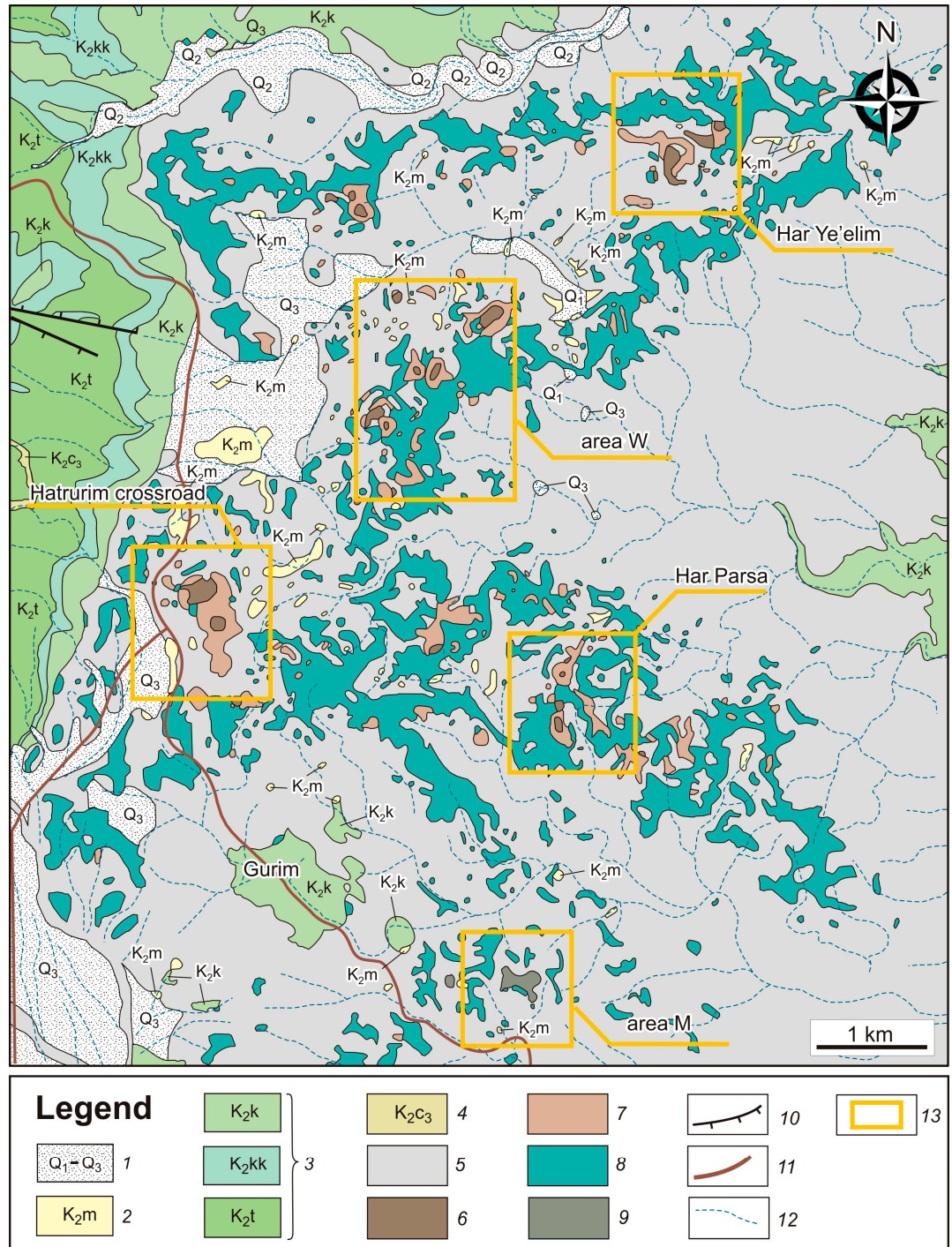

**Figure 3.** Geological map of the northwestern Hatrurim Basin, modified after the 1:50,000 Geological Map of Israel [75]. 1 = Pleistocene terrace conglomerates ($Q_1$, $Q_2$, $Q_3$); 2 = Maastrichtian organic-rich marine chalk ($K_2m$); 3 = Campanian ($K_2k$), Santonian ($K_2kk$), and Turonian ($K_2t$) limestone, chalk, and dolomite with chert and phosphorite intercalations; 4 = Cenomanian ($K_2c_3$) limestone, dolomite, and chalk; 5 = Low-grade Hatrurim Fm rocks; 6 = Larnite rocks (High-grade Hatrurim Fm rocks); 7 = "Olive rocks" (Hatrurim Fm); 8 = Spurrite marbles (medium-grade Hatrurim Fm rocks); 9 = Pseudo-conglomerates; 10 = Faults; 11 = Road;12 = Wadi; 13 = Sampling sites. Areas W and M are sampling sites of W and M series, respectively.

The specific (non-stratigraphic) sequence of the MZ rocks cropped out the Upper Cretaceous-Low Tertiary strata of the area and spread down to depths from 30–40 to 120 m. Term "MZ sequence" involves

highly diverse sedimentary rocks, which underwent post-depositional alteration. The section mainly consists of unevenly distributed brecciated sediments (mainly carbonates that enclose phosphorite, chert, and marl), which have undergone strong low-temperature alteration and are cut by abundant veinlets containing calcite, aragonite, gypsum, various Ca silicate hydrates, zeolites and ettringite group minerals. Numerous foci of diverse anhydrous high- to ultrahigh-temperature combustion metamorphic rocks are mainly concentrated in the uppermost part of this section. They occupy no more than 3–5% of the area of the MZ complexes, reaching about 10% in the Hatruruim Basin [1,4,46–51,61,64,65].

Within the Hatrurim Basin, combustion metamorphism has occurred at thousands of local foci, the number, size and degree of thermal alteration of which, increase toward the surface. This area is characterized by very specific relief of conical hills, many of which are topped by separate blocks and lenses of resistant high-grade combustion metamorphic rocks (Figure 2). Burg et al. [48] divided the sequence of the MZ rocks into four mapping units and revealed spatial distribution of different types of CM rocks within the Hatrurim Basin. The basal strata are mainly composed of diverse brecciated and hydrothermally altered sedimentary (indurated chalk-like masses, indurated marls, limestone, less abundant chert and phosphorite) and metasedimentary rocks [1,50,51,64], which are crisscrossed by numerous veinlets [1,67,69]. At the level of 10–40 m above the top of the sedimentary section, they include numerous subhorizontal bodies of highly-resistant gehlenite hornfelses (thickness up to 2 m and length up to 50–100 m) [48,64]. The following minerals have been distinguished in these rocks: gehlenite-rich melilite, Ti-rich andradite, rankinite, fluorapatite-fluorellestadite solid solutions, $Ca_3Si_3O_9$ modifications (±cuspidine, brownmillerite, Ca ferrites, Cr-spinel; kalsilite, larnite). The middle part of the section is mainly composed by spurrite marbles (T ~700–800 °C). Scarce relatively fresh marbles, as well as their widespread hydrated varieties (up to 4 m in thickness), cover large areas and fringe a system of circular structures, hundreds of meters to kilometers in diameter (Figure 3). Fresh rocks contain calcite, spurrite, brownmillerite, fluorapatite-fluorellestadite solid solutions, (±gehlenite, mayenite supergroup minerals, periclase, Ca ferrites, graphite and scarce Ca, Fe, K, Zn, Cu, Ni, Ba sulfides).

Some of the largest hills in the central and northern part of the basin are topped by the ultrahigh-temperature CM rocks (Figures 2 and 3). Among them, totally decarbonated Ca-rich and Al-poor larnite rocks are relatively widespread, while Al-enriched clinopyroxene-anorthite hornfelses as well as molten rocks-paralavas are sporadic and scarce. Clinopyroxene-anorthite paralavas contain K-feldspar and Fe–Ti–Al oxides (±fluorapatite, orthopyroxene, tridymite; fayalite; pyrrhotite; the whitlockite) [4,40,50,52,64]. Rock-forming minerals of Ca-rich paralavas are gehlenite, $Ca_3Si_3O_9$ and $Ca_2SiO_4$ modifications, rankinite, Ti-rich andradite, kalsilite, fluorapatite-fluorellestadite solid solutions. Their accessory assemblages are highly variable and a lot of rare and new minerals came from this rock type [1,24,32,52,55,59].

Given the extreme complexity of the Hatrurim Basin, it appears reasonable to limit this consideration to larnite-bearing CM rocks found at all hypsometric levels [1,4,32,48,76], mainly in the northern and central parts of the basin near the Har Parsa, Har Ye'eleim and Hatrurim cross-road areas (Figures 2 and 3). They occur as lenses or occasionally as cliff exposures, commonly in zones of brecciated country rocks, and are often brecciated themselves. Within 20 m above the section base, they are from a few tens of cm to a few meters thick and neighbor gehlenite hornfels bodies. At hilltops, in the upper Hatrurim Fm section, monolithic fresh larnite rocks make up separate isometric massive blocks, plates or cliff scarps, up to 50 m across and 10 m thick. One of the most prominent outcrops of larnite rocks was found on Har-Parsa mountain where they coexist with gehlenite hornfels cut by thin paralava veins.

Larnite CM rocks can also exist as remnant "pebbles" or "cobbles" from 1–2 to 15–20 cm in diameter, buried in a light-colored matrix of secondary calcite, aragonite, gypsum, and ettringite, with lesser percentages of tobermorites, jennite, afwillite, and hydrogarnets. The matrix contains remnant grains of opaque minerals or occasionally gehlenite and rankinite. The secondary phases form distinct

laminated aggregates that coat the pebbles of larnitic rocks like shells (Figure 2). Up the section, larnite CM rocks occur as isolated mottles (to 10 m across) among strongly altered brownmillerite-bearing varieties. Their original sizes are hard to estimate because of high-grade retrograde metamorphism. Individual bodies of larnitic CM rocks were described in previous publications [1,4,25,26,32,48,64,76].

## 4. Results and Discussion

### 4.1. Major- and Trace-Element Chemistry of Hatrurim Fm. larnite CM Rocks and Their Sedimentary Protoliths

**Sedimentary rocks**. The Hatrurim Fm CM rocks formed by isochemical metamorphism of bitumen-rich chalky sediments ('oil shales') of the Maastrichtian Ghareb Fm or, less often, Paleocene marls of the Taqiye Fm [62]. Their stratigraphic equivalents in Jordan appear as bioproductive chalky-marly sediments of the Muwaqqar Fm, which are sedimentary precursors of CM spurrite marbles [3,6,77,78]. The Ghareb Fm bituminous chalks (oil shales) are hypersaline shallow marine sediments mainly consisting of biogenic calcite with minor contributions of detrital quartz, clay minerals, apatite or phosphorite matter, and pyrite (Figure 4). The sediments generally have uniform major-element compositions and contain on average 37.3 wt % CaO, 8.84 wt % $SiO_2$, 3.37 wt % $Al_2O_3$, 1.77 wt % $Fe_2O_3$, and 1.34 wt % $P_2O_5$; the $CO_2$ contents are from 24.6 to 34.7 wt %, while $TiO_2$, MgO, $Na_2O$, and $K_2O$ are within 0.60 wt % (Table 4).

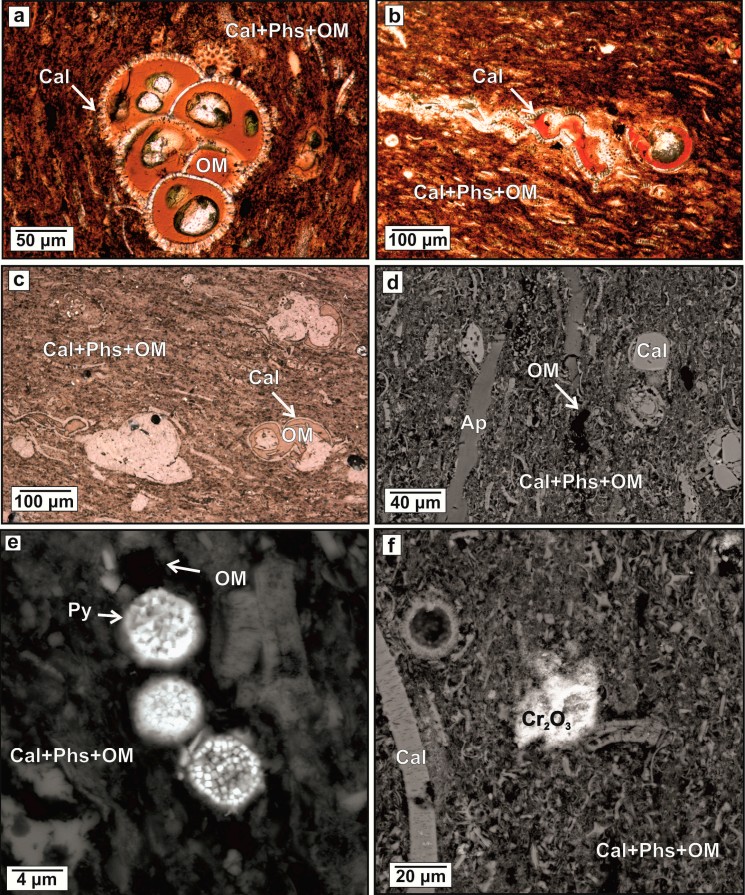

**Figure 4.** (**a–c**) Appearance of Maastrichtian organic-rich marine chalky sediments ('oil shales'): transmitted light images, (**d–f**) back-scattered electron (BSE) images. Mineral names are abbreviated as Ap = francolite; Cal = calcite; $Cr_2O_3$ = Cr (oxy)hydroxides; OM = organic matter; Phs = phosphatic material; Py = pyrite.

The Maastrichtian bioproductive chalks and marls contain 8.52 wt % of total organic carbon (TOC) on average, with S-rich (up to 12 wt %) type II kerogen organic matter consisting of degraded marine

algae and phytoplankton [3,69,73,78,79]. Correspondingly, the rocks are rich in sulfide sulfur (and/or products of its oxidation): 3.77 wt % $SO_3$ on average (Table 4). The $S^{2-}$-TOC correlation is especially evident in fresh (non-oxidized) oil shales sampled from boreholes [78] and is as high as $R^2 = 0.92$ ($N = 9$) in those from Rotem borehole.

The Ghareb Fm chalks are markedly depleted in Mg, Mn, Rb, Ba, and Pb with respect to Post-Archean marine carbonates [80], but have greater enrichment in P, S, Fe, and in TE typical of anoxic bioproductive marine environments, which are on average 95.2 ppm V, 221 ppm Cr, 127 ppm Ni, 61.9 ppm Cu; 158 ppm Zn, 12.7 ppm As, 1112 ppm Sr, 11.9 ppm Mo, and 16.5 ppm U (Tables 4 and 5; Figure 5). Cd contents in the rocks are as low as 0.83 ppm, though phosphate layers have rather high enrichment in Cd (5–154 ppm) and in U [81]; some calcareous layers contain abundant baryte concretions [82]. The coeval Muwaqqar Fm sediments in central Jordan store high or extremely high concentrations of Zn, Cd, U, Ni, and V, due to a specific deposition environment [3,6–8,78].

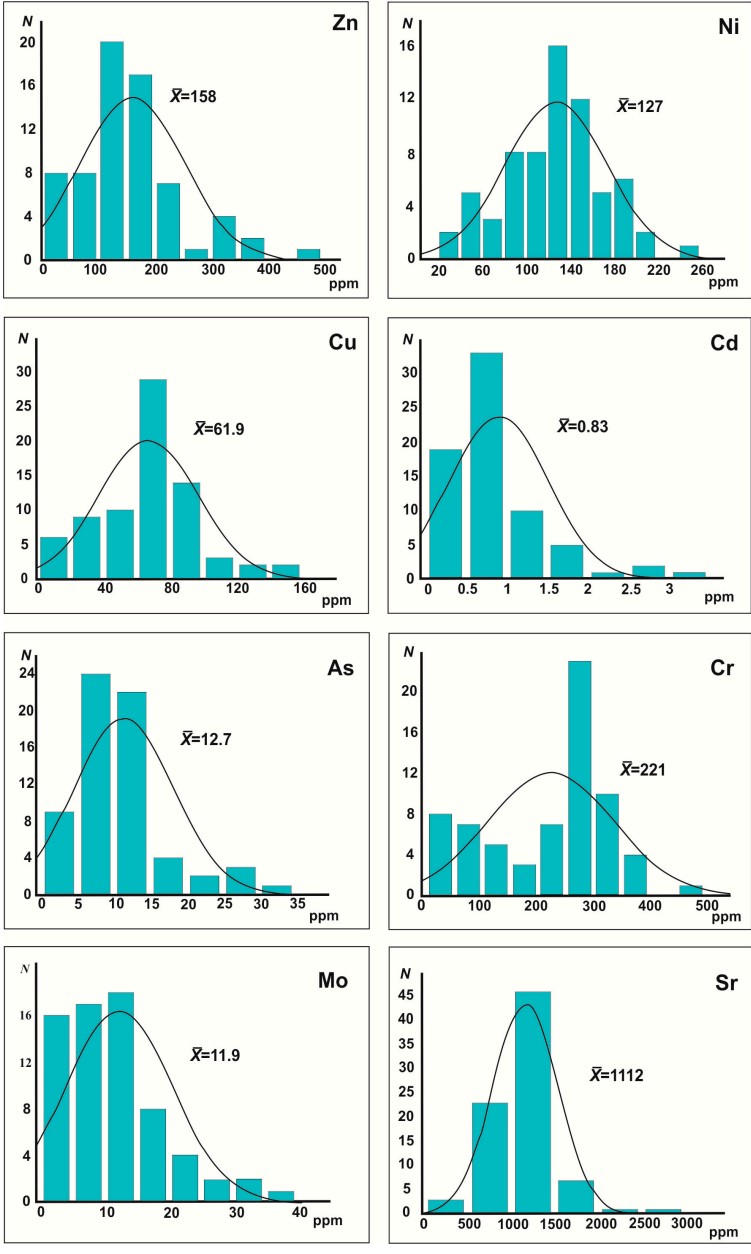

**Figure 5.** Trace-element compositions of the Ghareb Fm bituminous calcareous rocks, compared. *N* = number of samples; $\overline{X}$ = mean value (See data set in Table 4).

**Table 4.** Major-element (in wt %) compositions of sedimentary protoliths (Ghareb Fm "oil shales").

| Locality | | Sample | $SiO_2$ | $TiO_2$ | $Al_2O_3$ | $Fe_2O_3$ | MgO | CaO | $Na_2O$ | $K_2O$ | $P_2O_5$ | $SO_3$ | LOI | Total | $CO_2$ | Corg |
|---|---|---|---|---|---|---|---|---|---|---|---|---|---|---|---|---|
| Pama | 1 | YG-1 | 2.40 | 0.04 | 0.60 | 0.30 | 2.30 | 45.0 | n.a. | n.a. | 11.2 | 3.60 | 33.0 | 98.4 | n.a. | 6.00 |
| | 2 | YG-3 | 4.60 | 0.07 | 1.20 | 0.50 | 0.50 | 32.0 | n.a. | n.a. | 0.90 | 8.00 | n.a. | | n.a. | 21.0 |
| | 3 | YG-13 | 8.90 | 0.15 | 3.80 | 2.00 | 0.30 | 36.0 | n.a. | n.a. | 0.80 | 4.60 | n.a. | | n.a. | 7.00 |
| | 4 | YG-49 | 12.2 | 0.21 | 5.20 | 2.40 | 0.40 | 33.0 | n.a. | n.a. | 1.00 | 5.70 | n.a. | | n.a. | 7.00 |
| | 5 | YG-72p | 16.6 | 0.31 | 7.00 | 2.40 | 0.40 | 36.0 | n.a. | n.a. | 1.00 | 3.00 | n.a. | | n.a. | 8.00 |
| | 6 | A | 9.30 | 0.16 | 3.80 | 1.70 | 0.94 | 34.2 | 0.24 | 0.34 | 2.70 | 6.80 | 35.7 | 95.9 | n.a. | n.a. |
| | 7 | B | 11.1 | 0.21 | 5.10 | 2.30 | 0.93 | 34.1 | 0.21 | 0.35 | 1.80 | 6.20 | 33.6 | 95.9 | n.a. | n.a. |
| | 8 | C | 7.40 | 0.11 | 2.40 | 1.10 | 0.96 | 34.1 | 0.29 | 0.31 | 3.60 | 7.30 | 37.3 | 94.9 | n.a. | n.a. |
| Rotem borehole | 9 | 401/40–41 | 14.0 | n.a. | 5.53 | 2.70 | 0.61 | 31.7 | 0.24 | 0.37 | 0.73 | 6.90 | 38.2 | 94.1 | n.a. | 9.51 |
| | 10 | 423/42–43 | 11.7 | n.a. | 4.53 | 2.26 | 0.52 | 33.2 | 0.22 | 0.32 | 1.95 | 6.65 | 39.6 | 94.3 | n.a. | 10.1 |
| | 11 | 478/47–48 | 9.44 | n.a. | 3.53 | 1.96 | 0.50 | 37.6 | 0.21 | 0.29 | 1.75 | 5.93 | 39.7 | 94.9 | n.a. | 8.53 |
| | 12 | 523/52–53 | 7.63 | n.a. | 2.46 | 1.53 | 0.54 | 38.6 | 0.24 | 0.29 | 2.22 | 6.71 | 41.8 | 95.3 | n.a. | 10.3 |
| | 13 | 534/53–54 | 8.30 | n.a. | 2.87 | 1.74 | 0.58 | 36.0 | 0.24 | 0.28 | 3.07 | 7.48 | 42.0 | 95.1 | n.a. | 11.9 |
| | 14 | 556/55–56 | 9.94 | n.a. | 3.23 | 1.98 | 0.68 | 34.4 | 0.26 | 0.32 | 2.76 | 7.92 | 41.4 | 95.0 | n.a. | 11.8 |
| | 15 | 589/58–59 | 8.08 | n.a. | 1.41 | 0.70 | 0.53 | 35.6 | 0.24 | 0.30 | 4.66 | 7.65 | 44.3 | 95.7 | n.a. | 13.3 |
| | 16 | 601/60–61 | 5.91 | n.a. | 1.14 | 0.54 | 0.45 | 35.5 | 0.24 | 0.28 | 3.28 | 8.26 | 48.3 | 95.6 | n.a. | 15.6 |
| | 17 | 612/61–62 | 6.37 | n.a. | 1.27 | 0.50 | 0.48 | 35.0 | 0.24 | 0.29 | 3.42 | 8.50 | 48.2 | 95.8 | n.a. | 15.8 |
| Ta'alat Hayamim borehole | 18 | AG-47 | 11.7 | 0.10 | 4.30 | 2.10 | 0.70 | 43.0 | 0.30 | 0.50 | 0.19 | 1.20 | 35.1 | 99.2 | n.a. | n.a. |
| | 19 | AG-48 | 11.4 | 0.20 | 4.40 | 2.10 | 0.60 | 36.4 | 0.40 | 0.50 | 2.40 | 3.60 | 36.5 | 98.5 | n.a. | n.a. |
| Hatrurim Basin | 20 | YV-114 | 7.90 | 0.10 | 2.80 | 1.80 | 0.50 | 47.0 | 0.30 | 0.30 | 0.70 | n.a. | 37.9 | 99.3 | n.a. | n.a. |
| | 21 | 4 | 6.90 | 0.10 | 2.00 | 1.00 | 1.00 | 46.7 | 0.50 | <0.20 | 2.50 | n.a. | 38.8 | 99.7 | 24.6 | n.a. |
| Hatrurim Basin | 22 | 11 | 5.60 | <0.1 | 0.90 | 0.60 | 0.50 | 49.0 | 1.20 | 0.30 | 2.80 | n.a. | 39.6 | 100.8 | 31.7 | n.a. |
| | 23 | 26 | 18.5 | 0.30 | 3.60 | 1.80 | 2.20 | 37.9 | 1.40 | 0.30 | 2.80 | n.a. | 31.5 | 100 | n.a. | n.a. |
| | 24 | 28 | 16.9 | 0.20 | 5.30 | 2.20 | 1.60 | 33.5 | 1.70 | 0.30 | 1.40 | n.a. | 36.9 | 100 | n.a. | n.a. |
| | 25 | AB-624 | 8.50 | 0.12 | 3.20 | 3.20 | 0.80 | 42.0 | n.a. | n.a. | <0.50 | <0.50 | n.a. | | n.a. | n.a. |
| | 26 | AB-651 | 13.8 | 0.22 | 5.70 | 2.60 | 1.40 | 36.0 | n.a. | n.a. | 1.20 | <0.50 | n.a. | | n.a. | n.a. |
| | 27 | AB-657 | 2.40 | 0.02 | 0.60 | 1.30 | 2.20 | 43.0 | n.a. | n.a. | <0.50 | <0.50 | n.a. | | n.a. | n.a. |
| | 28 | AB-730 | 3.70 | 0.04 | 1.20 | 1.50 | 0.60 | 51.0 | n.a. | n.a. | <0.50 | <0.50 | n.a. | | n.a. | n.a. |
| Nabi Musa | 29 | 10 | 5.20 | 0.10 | 1.00 | 0.80 | 0.50 | 49.4 | 0.80 | 0.20 | 2.20 | n.a. | 40.3 | 100 | n.a. | n.a. |
| | 30 | AG-27[b] | 3.10 | <0.10 | 0.90 | 0.40 | 0.40 | 41.8 | 0.30 | 0.20 | 1.70 | 0.40 | 48.7 | 97.4 | n.a. | n.a. |
| | 31 | AG-19[b] | 6.30 | <0.10 | 1.00 | 0.40 | 0.30 | 51.0 | 0.30 | 0.10 | 3.60 | 0.90 | 34.6 | 99.4 | n.a. | n.a. |

**Table 4.** *Cont.*

| Locality | Sample | | SiO$_2$ | TiO$_2$ | Al$_2$O$_3$ | Fe$_2$O$_3$ | MgO | CaO | Na$_2$O | K$_2$O | P$_2$O$_5$ | SO$_3$ | LOI | Total | CO$_2$ | Corg |
|---|---|---|---|---|---|---|---|---|---|---|---|---|---|---|---|---|
| Nahal Ayalon | 32 | 438 | 8.10 | 0.20 | 3.50 | 0.60 | 0.90 | 46.5 | 0.20 | <0.20 | 0.70 | 0.8 * | 38.8 | 100 | 34.7 | 0.30 |
| | *N* | | 78 | 69 | 78 | 78 | 78 | 78 | 23 | 23 | 78 | 72 | | | 3 | 41 |
| | Mean | | 8.84 | 0.14 | 3.37 | 1.77 | 0.60 | 37.3 | 0.40 | 0.28 | 1.34 | 3.77 | | | 30.3 | 8.52 |
| | *S* | | 3.77 | 0.08 | 1.84 | 0.77 | 0.42 | 6.70 | 0.33 | 0.11 | 1.02 | 2.76 | | | 5.19 | 3.05 |
| | Min | | 2.40 | 0.02 | 0.60 | 0.30 | 0.30 | 25.0 | 0.20 | 0.10 | 0.19 | 0.10 | | | 24.6 | 0.30 |
| | Max | | 18.5 | 0.31 | 7.00 | 3.50 | 5.10 | 51.0 | 1.70 | 0.50 | 11.2 | 8.50 | | 37.3 | 34.7 | 21.0 |
| | Post-Archean carbonate ** | | 13.7 | 0.10 | 1.76 | 0.91 | 10.6 | 34.5 | 0.17 | 0.62 | 0.05 | n.a. | 37.3 | 99.7 | n.a. | n.a. |

n.a. = not analyzed, *N* = number of samples, Mean = mean value, *S* = standard deviation, Min = minimum value, Max = maximum value, Corg = organic carbon. Cl = 0.17–0.85 wt % in samples 18–22; H$_2$O (in wt %) = 3.80, 3.85 and 2.55 in samples 29, 19 and 20, respectively; * = S–0.2 wt %. ** = composition of Post-Archean marine carbonate [80]. 1–5, 25–28—after [62]; 6–8—after [79]; 9–17—after [83]; 18–19, 30–31—after [84]; 20–24, 29—after [51]; 32—after [85].

**Table 5.** Trace-element compositions (in ppm) of sedimentary protoliths (Ghareb Fm "oil shales").

| Locality | | Sample | V | Cr | Mn | Co | Ni | Cu | Zn | As | Rb | Sr | Mo | Cd | Sb | Ba | Pb | Th | U |
|---|---|---|---|---|---|---|---|---|---|---|---|---|---|---|---|---|---|---|---|
| | 1 | PAMA | 67.3 | 245 | n.a. | n.a. | 139 | 74.9 | 245 | 6.21 | 14.7 | 1370 | 11.9 | n.a. | n.a. | 125 | n.a. | 2.61 | 15.1 |
| | 2 | PAMA | 57.0 | 234 | 49.0 | n.a. | 142 | 67.0 | 220 | 6.00 | 14.0 | 1226 | 8.00 | n.a. | n.a. | 55.0 | n.a. | <2.00 | 27.0 |
| | 3 | YV-195 | 85.0 | 240 | n.a. | n.a. | 115 | 55.0 | 165 | 9.00 | 14.0 | n.a. | 22.0 | n.a. | n.a. | 50.0 | 14.0 | 2.00 | 11.0 |
| | 4 | YG-1 | 83.0 | 190 | 60.00 | 1.70 | 117 | 64.0 | 206 | 54.0 | n.a. | 2079 | 19.0 | 3.10 | 1.90 | 66.0 | 3.30 | 1.50 | 100 |
| Pama | 5 | YG-3 | 140 | 465 | 53.00 | 2.40 | 244 | 129 | 380 | 89.0 | n.a. | 1442 | 30.0 | 2.10 | 4.30 | 59.0 | 4.70 | 1.80 | 42.0 |
| | 6 | YG-13 | 89.0 | 232 | 45.00 | 4.40 | 126 | 61.0 | 126 | 11.0 | n.a. | 1331 | 11.0 | 0.50 | 0.80 | 45.0 | 4.00 | 2.00 | 13.0 |
| | 7 | YG-49 | 99.0 | 259 | 50.00 | 5.80 | 128 | 65.0 | 131 | 13.0 | n.a. | 1051 | 13.0 | 0.60 | 1.10 | 54.0 | 5.30 | 2.30 | 16.0 |
| | 8 | YG-72p | 104 | 321 | 44.0 | 4.90 | 79.0 | 60.0 | 109 | 7.00 | n.a. | 1183 | 6.00 | 0.50 | 0.80 | 74.0 | 6.40 | 2.40 | 16.0 |
| Ta'alat Hayamim borehole | 9 | AG-47[c] | 90.0 | 80.0 | 1300 | 25.0 | 220 | 30.0 | 200 | 3.50 | 20.0 | 930 | 1.10 | 0.90 | 1.80 | 45.0 | 15.0 | 3.00 | 3.50 |
| | 10 | AG-48[c] | 130 | 300 | 80.0 | 15.0 | 150 | 115 | 200 | 15.0 | 19.0 | 1200 | 13.0 | 0.60 | 1.20 | 110 | 7.50 | 2.00 | 15.5 |
| | 11 | YV-114 | 54.0 | 115 | 200 | n.a. | 47.0 | n.a. | 50.0 | n.a. | 11.0 | 1100 | 5.70 | n.a. | n.a. | 46.0 | n.a. | 1.40 | 5.40 |
| | 12 | 4 | 115 | 400 | <100 | n.a. | 190 | n.a. | 175 | n.a. | 1.00 | 500 | 20.0 | n.a. | n.a. | 490 | n.a. | 1.20 | 29.0 |
| | 13 | 11 | 36.0 | 25.0 | <100 | n.a. | 95.0 | n.a. | 45.0 | n.a. | 7.00 | 680 | 10.0 | n.a. | n.a. | 40.0 | n.a. | 0.40 | 9.00 |
| Hatrurim Basin | 14 | 26 | 90.0 | 210 | 100 | n.a. | 150 | n.a. | 360 | n.a. | 13.0 | 1266 | 4.00 | n.a. | n.a. | 755 | n.a. | 2.20 | 17.0 |
| | 15 | 28 | 82.0 | 200 | 95.0 | n.a. | 175 | n.a. | 460 | n.a. | 18.0 | 1389 | 9.00 | n.a. | n.a. | 2200 | n.a. | 1.90 | 16.0 |
| | 16 | AB-624 | 168 | 52.0 | 827 | 18.0 | 152 | 37.0 | 133 | 11.00 | n.a. | 970 | 2.70 | 1.60 | 0.70 | 57.0 | 12.0 | 1.40 | 7.00 |
| | 17 | AB-651 | 111 | 155 | 109 | 8.00 | 153 | 68.0 | 171 | 13.0 | n.a. | 780 | 4.80 | 0.60 | 1.00 | 47.0 | 8.00 | 1.90 | 16.0 |

**Table 5.** *Cont.*

| Locality | | Sample | V | Cr | Mn | Co | Ni | Cu | Zn | As | Rb | Sr | Mo | Cd | Sb | Ba | Pb | Th | U |
|---|---|---|---|---|---|---|---|---|---|---|---|---|---|---|---|---|---|---|---|
| | 18 | AB-657 | 34.0 | 4.00 | 131 | 2.00 | 31.0 | 9.00 | 54.0 | 4.00 | n.a. | 890 | 1.50 | 0.20 | 0.30 | 68.0 | 4.00 | 0.40 | 6.00 |
| | 19 | AB-730 | 37.0 | 15.0 | 218 | 5.00 | 48.0 | 16.0 | 24.0 | 4.00 | n.a. | 966 | 1.20 | 0.20 | 0.20 | 72.0 | 5.00 | 0.70 | 3.00 |
| | 20 | 10 | 72.0 | 250 | <100 | n.a. | 155 | n.a. | 338 | n.a. | 6.00 | 1300 | 22.0 | n.a. | n.a. | 215 | n.a. | 0.70 | 23.0 |
| Nabi Musa | 21 | AG-27[b] | 50.0 | 228 | 15.0 | 4.00 | 84.0 | 74.0 | 340 | 9.00 | 4.00 | 1021 | 13.0 | 1.90 | 1.00 | 86.0 | 3.00 | 25.5 | 0.70 |
| | 22 | AG-19[b] | 83.0 | 138 | 15.0 | 2.00 | 73.0 | 83.0 | 330 | 7.00 | 1.10 | 1220 | 12.4 | 1.40 | 0.60 | 174 | 1.00 | 33.0 | 1.60 |
| | N | | 68 | 68 | 66 | 59 | 68 | 62 | 68 | 62 | 13 | 67 | 68 | 59 | 59 | 68 | 60 | 68 | 68 |
| | Mean | | 95.2 | 221 | 107 | 5.37 | 127 | 61.9 | 158 | 12.7 | 11.1 | 1112 | 11.9 | 0.83 | 1.21 | 88.8 | 5.61 | 2.14 | 16.5 |
| | S | | 30.5 | 107 | 140 | 2.77 | 42.9 | 23.6 | 82.4 | 8.54 | 5.80 | 261 | 7.71 | 0.48 | 0.64 | 112 | 2.40 | 3.05 | 9.11 |
| | Min | | 33.0 | 4.00 | 15.0 | 1.30 | 31.0 | 9.00 | 15.0 | 2.00 | 1.00 | 400 | 0.90 | 0.20 | <0.1 | 24.0 | 1.00 | 0.40 | 0.70 |
| | Max | | 168 | 465 | 1300 | 25.0 | 244 | 129 | 460 | 89.0 | 20.0 | 3000 | 37.0 | 3.10 | 4.30 | 2200 | 15.0 | 33.0 | 100 |
| Post-Archean carbonate ** | | | 20.0 | 9.00 | 464 | 3.00 | 6.00 | 8.00 | 22.0 | 4.80 | 17.0 | 245 | 0.56 | 0.11 | n.a. | 178 | 13.3 | 1.99 | 1.10 |

n.a. = not analyzed, *N* = number of samples, Mean = mean value, *S* = standard deviation, Min = minimum value, Max = maximum value. Cd (in ppm): 0.90, 0.60 and 1.40 in samples 9, 10 and 22, respectively; Zr (in ppm): 32.5, 43, 90 and 60 in samples 1, 2, 12 and 13, respectively; Y: 20–49 ppm in samples 1–3, 9–15, 20–22. ** The composition of Post-Archean marine carbonate [80]. 1, 3—authors data; 2, 11–15, 20—after [51]; 4–8, 16–19—after [62]; 9–10, 21–22—after [84].

The principal component analysis reveals two main generalized variables that account for 67.7% of the total variance (Table 6). One component records high negative correlations among elements in the clastic sediment fraction (Si, Ti, Al, Fe Co, Pb and Th) and positive correlations among other elements that reside in the carbonate and phosphate fractions (Mg, Ca, P, Sr, Zn, As, Mo, Cd, Sb, and U); the other component refers to high positive correlations among TOC and organic-bonded trace elements (V, Cr, Mn, Co, Ni, Cu, Zn, Pb).

**Table 6.** Effective correlation factor for bitumen-bearing calcareous rocks from Hatrurim Basin and Nabi Musa MZ areas, Pama career, Ta'alat Hayamim borehole, Ghareb Fm, Israel.

| Component | Factor | | | | | Communality |
|---|---|---|---|---|---|---|
| | 1 | 2 | 3 | 4 | 5 | |
| $SiO_2$ | **−0.94** | 0.16 | −0.16 | −0.01 | −0.07 | 0.94 |
| $TiO_2$ | **−0.93** | 0.11 | −0.17 | −0.04 | −0.10 | 0.92 |
| $Al_2O_3$ | **−0.95** | 0.15 | −0.19 | 0.03 | −0.04 | 0.97 |
| $Fe_2O_3$ | **−0.89** | 0.31 | −0.19 | −0.02 | −0.04 | 0.93 |
| MgO | **0.62** | −0.24 | **−0.63** | 0.35 | −0.04 | 0.96 |
| CaO | **0.57** | −0.41 | −0.38 | **−0.54** | 0.01 | 0.94 |
| $P_2O_5$ | **0.62** | −0.29 | **−0.65** | 0.27 | −0.04 | 0.97 |
| $SO_3$ | 0.31 | 0.31 | **0.55** | **0.62** | 0.25 | 0.94 |
| TOC | 0.26 | **0.61** | 0.20 | 0.33 | 0.23 | 0.65 |
| V | 0.08 | **0.91** | −0.09 | −0.06 | −0.02 | 0.85 |
| Cr | 0.12 | **0.92** | 0.11 | −0.20 | −0.09 | 0.93 |
| Mn | −0.15 | **0.69** | **−0.56** | 0.21 | 0.21 | 0.91 |
| Co | **−0.74** | **0.53** | −0.28 | 0.09 | 0.10 | 0.92 |
| Ni | 0.40 | **0.82** | 0.16 | −0.04 | −0.02 | 0.86 |
| Cu | 0.38 | **0.77** | −0.01 | −0.41 | −0.07 | 0.91 |
| Zn | **0.73** | **0.60** | 0.13 | −0.03 | −0.09 | 0.92 |
| As | **0.75** | 0.46 | −0.17 | 0.01 | 0.06 | 0.81 |
| Sr | **0.69** | 0.08 | **−0.61** | −0.23 | 0.02 | 0.91 |
| Mo | **0.73** | 0.38 | 0.20 | −0.03 | −0.14 | 0.74 |
| Cd | **0.92** | 0.00 | −0.27 | 0.17 | −0.03 | 0.95 |
| Sb | **0.83** | 0.36 | 0.27 | −0.17 | −0.08 | 0.93 |
| Ba | −0.08 | 0.15 | 0.08 | 0.32 | **−0.89** | 0.94 |
| Pb | **−0.58** | **0.64** | −0.44 | −0.10 | 0.03 | 0.95 |
| Th | **−0.73** | 0.44 | −0.35 | 0.19 | −0.03 | 0.89 |
| U | **0.83** | −0.06 | −0.48 | 0.24 | −0.07 | 0.98 |
| Factor loading, % | 43.20 | 24.52 | 12.20 | 6.32 | 4.20 | |

Bold highlights correlations of more than 0.50. TOC = total organic carbon.

The compositions of **$Ca_2SiO_4$-bearing CM rocks** sampled in different areas within the Hatrurim Basin fall within quite narrow ranges (Tables 7 and 8), with high contents of CaO (from 47.9 to 56.7 wt %), moderate $SiO_2$ (19.8 to 27.1 wt %), $Al_2O_3$ (7.9 to 12.5 wt %), and (FeO + $Fe_2O_3$) (2.4 to 6.3 wt %, with predominant $Fe_2O_3$), and low contents of MgO (0.5 to 1.8 wt %), as well as $TiO_2$, $Na_2O$, and $K_2O$ (<1 wt %). The $Ca_2SiO_4$-bearing CM rocks contain either ye'elimite or mayenite, according to markedly different contents of sulfate sulfur (1.35–4.97 wt % $SO_3$; $x_{mean}$ = 2.71 vs. <0.05–0.73 wt % $SO_3$; $x_{mean}$ = 0.22, respectively). The ye'elimite-bearing rocks have slightly greater enrichment in $Fe_2O_3$ ($x_{mean}$ = 4.45 wt % vs. $x_{mean}$ = 3.17 wt %), MgO ($x_{mean}$ = 1.18 wt % vs. $x_{mean}$ = 0.75 wt %), and some redox-sensitive trace elements, with $x_{mean}$ contents of 51.9 vs. 33.8 ppm Cu; 240 vs. 119 ppm Zn; 13.7 vs. 3.08 ppm Se, as well as in Ba (908 vs. 168 ppm), whereas the mayenite-bearing variety contains more $P_2O_5$ ($x_{mean}$ = 2.33 wt % vs. $x_{mean}$ = 1.88 wt %), Cr (239 vs. 174 ppm), and As (18.4 vs. 12.1 ppm).

The sedimentary protoliths of the two varieties apparently were different though belonged to the same sedimentary sequence. Judging by high Ca, Si, and Al contents coupled with high sulfur enrichment and a characteristic assemblage of redox-sensitive elements (Fe, Cu, Zn, Mo, and Se), the ye'elimite-bearing larnite CM rocks were derived from oil shales with secondary gypsum [4], whereas the mayenite variety with lower S and RSE may result from annealing of organic-poor chalks with uneven phosphatic impregnation.

**Table 7.** Bulk (in wt %) and trace element (in ppm) compositions of Hatrurim ye'elimite–larnite CM rocks.

| Sample | LLD | YV-412 | Y-5-1 | YV-411 | Y-6-3 | Y-8 | M5-30 | YV-410 | M5-31 | M4-217 | MP-10-1 | MP-10-2 | *n* | Mean | *S* | Min | Max |
|---|---|---|---|---|---|---|---|---|---|---|---|---|---|---|---|---|---|
| | | | | | | Bulk composition (in wt %) | | | | | | | | | | | |
| SiO$_2$ | 0.25 | 21.0 | 21.3 | 21.9 | 24.1 | 22.5 | 24.6 | 23.0 | 25.6 | 25.9 | 21.2 | 23.2 | 11 | 23.1 | 1.51 | 21.0 | 25.9 |
| TiO$_2$ | 0.10 | 0.41 | 0.35 | 0.41 | 0.36 | 0.32 | 0.39 | 0.42 | 0.43 | 0.38 | 0.42 | 0.45 | 11 | 0.40 | 0.03 | 0.32 | 0.45 |
| Al$_2$O$_3$ | 0.25 | 9.88 | 9.27 | 10.4 | 9.73 | 7.92 | 11.7 | 11.9 | 12.1 | 11.7 | 10.7 | 10.8 | 11 | 10.7 | 0.95 | 7.92 | 12.1 |
| Fe$_2$O$_3$ | 0.20 | 4.32 | 3.78 | 5.00 | 2.73 | 2.42 | 4.55 * | 4.80 | 5.81 * | 4.49 * | 5.70 * | 4.70 * | 11 | 4.45 | 0.83 | 2.42 | 5.81 |
| FeO | 0.10 | bdl | 0.48 | bdl | 0.48 | 0.42 | n.a. | bdl | n.a. | n.a. | n.a. | n.a. | 6 | 0.28 | 0.20 | <0.10 | 0.48 |
| MnO | 0.01 | bdl | 0.21 | 0.05 | 0.04 | 0.03 | 0.14 | 0.26 | 0.14 | 0.15 | 0.06 | 0.01 | 11 | 0.10 | 0.07 | <0.01 | 0.26 |
| MgO | 0.20 | 0.81 | 1.15 | 1.37 | 1.78 | 1.66 | 0.82 | 1.13 | 0.82 | 0.93 | 1.70 | 1.01 | 11 | 1.18 | 0.33 | 0.81 | 1.78 |
| CaO | 0.25 | 56.0 | 51.9 | 53.0 | 51.6 | 52.9 | 53.6 | 51.1 | 50.8 | 51.9 | 51.3 | 51.3 | 11 | 52.1 | 0.90 | 50.8 | 56.0 |
| N.a.$_2$O | 0.05 | 0.25 | 0.29 | 0.47 | 0.45 | 0.16 | 0.15 | 0.18 | 0.72 | 0.30 | 0.56 | 0.60 | 11 | 0.36 | 0.16 | 0.15 | 0.72 |
| K$_2$O | 0.05 | 0.13 | 0.60 | 0.35 | 1.74 | 0.19 | 0.09 | 2.43 | 0.48 | 0.45 | 0.13 | 0.15 | 11 | 0.47 | 0.51 | 0.09 | 2.43 |
| P$_2$O$_5$ | 0.03 | 1.34 | 3.84 | 1.92 | 0.78 | 0.79 | 2.31 | 1.82 | 2.07 | 2.25 | 2.41 | 1.98 | 11 | 1.88 | 0.52 | 0.78 | 3.84 |
| CO$_2$ | 0.06 | 0.76 | 0.67 | 0.86 | 0.76 | 0.69 | 0.89 | 0.49 | 0.08 | 0.62 | n.a. | n.a. | 9 | 0.69 | 0.12 | 0.08 | 0.89 |
| F | 0.03 | n.a. | bdl | n.a. | bdl | 0.09 | 0.07 | n.a. | n.a. | n.a. | n.a. | n.a. | 4 | 0.14 | 0.09 | <0.03 | 0.26 |
| SO$_3$ | 0.05 | 4.97 | 4.44 | 3.29 | 2.83 | 2.70 | 2.20 | 2.17 | 1.39 | 1.35 | 2.31 | 3.10 | 11 | 2.71 | 0.86 | 1.35 | 4.97 |
| H$_2$O | 0.05 | 0.10 | 0.99 | 0.96 | 2.50 | 7.54 | 2.43 | 0.54 | 0.42 | n.a. | n.a. | n.a. | 8 | 1.31 | 0.93 | 0.10 | 7.54 |
| Total | - | 99.9 | 99.5 | 99.9 | 99.9 | 100.3 | 99.4 | 100.2 | 100.9 | 100.4 | 96.5 | 97.3 | - | - | - | - | - |
| | | | | | | Trace element composition (in ppm) | | | | | | | | | | | |
| B | 1.00 | 26.9 | 42.2 | 43.7 | 45.3 | 37.3 | n.a. | 44.5 | n.a. | n.a. | n.a. | n.a. | 6 | 41.9 | 22.5 | 26.9 | 45.3 |
| V | 2.00 | 40.0 | 87.0 | 112 | 82.0 | 141 | 39.9 | 82.0 | 53.3 | 49.2 | n.a. | n.a. | 9 | 72.2 | 25.6 | 39.9 | 141 |
| Cr | 1.50 | 224 | 178 | 243 | 119 | 40.6 | 182 | 175 | 377 | 95.0 | n.a. | n.a. | 9 | 174 | 52.6 | 40.6 | 377 |
| Ni | 1.00 | 240 | 182 | 151 | 65.0 | 44.2 | 152 | 214 | 205 | 129 | n.a. | n.a. | 9 | 157 | 50.8 | 44.2 | 240 |
| Cu | 1.00 | 96.4 | 50.3 | 57.5 | 17.7 | 24.6 | 103 | 40.4 | 66.2 | 34.8 | n.a. | n.a. | 9 | 51.9 | 25.2 | 17.7 | 103 |
| Zn | 1.00 | 244 | 376 | 343 | 73.0 | 60.5 | 58.7 | 333 | 426 | 249 | n.a. | n.a. | 9 | 240 | 128 | 58.7 | 426 |
| Ga | 0.70 | 8.00 | 7.00 | 9.45 | 8.37 | 6.19 | 9.28 | 11.5 | 8.71 | 8.45 | n.a. | n.a. | 9 | 8.47 | 0.82 | 6.19 | 11.5 |
| Ge | 0.20 | 1.06 | bdl | 1.30 | 1.26 | 0.52 | 2.10 | 3.02 | 1.29 | 2.38 | n.a. | n.a. | 9 | 1.42 | 0.77 | <0.2 | 3.02 |
| As | 0.80 | 13.2 | 15.8 | 14.5 | bdl | 0.90 | 22.1 | 18.6 | 13.0 | 8.72 | n.a. | n.a. | 9 | 12.1 | 6.85 | <0.8 | 22.1 |
| Se | 0.20 | 96.2 | 2.93 | 9.77 | 39.7 | 19.9 | bdl | bdl | bdl | 23.4 | n.a. | n.a. | 9 | 13.7 | 14.7 | <0.2 | 96.2 |
| Rb | 0.20 | 4.92 | 11.7 | 7.72 | 31.9 | 5.00 | 3.45 | 2.65 | 10.4 | 12.0 | n.a. | n.a. | 9 | 7.88 | 3.53 | 2.65 | 31.9 |
| Sr | 1.00 | 2009 | 1453 | 1502 | 1407 | 1243 | 2903 | 1407 | 2050 | 2203 | n.a. | n.a. | 9 | 1719 | 351 | 1243 | 2903 |
| Y | 0.20 | 36.9 | 44.1 | 52.4 | 41.2 | 47.9 | 35.4 | 53.2 | 34.5 | 69.2 | n.a. | n.a. | 9 | 44.4 | 7.09 | 34.5 | 69.2 |

**Table 7.** *Cont.*

| Sample | LLD | YV-412 | Y-5-1 | YV-411 | Y-6-3 | Y-8 | M5-30 | YV-410 | M5-31 | M4-217 | MP-10-1 | MP-10-2 | *n* | Mean | *S* | Min | Max |
|---|---|---|---|---|---|---|---|---|---|---|---|---|---|---|---|---|---|
| | | | | | | Trace element composition (in ppm) | | | | | | | | | | | |
| Zr | 0.50 | 69.4 | 65.7 | 67.0 | 73.7 | 72.0 | 68.5 | 69.0 | 80.4 | 77.0 | n.a. | n.a. | 9 | 70.9 | 3.49 | 65.7 | 80.4 |
| Nb | 0.20 | 9.08 | 6.34 | 9.32 | 7.63 | 6.62 | 5.51 | 10.4 | 8.31 | 10.5 | n.a. | n.a. | 9 | 8.24 | 1.48 | 5.51 | 10.5 |
| Mo | 0.20 | 16.0 | 12.1 | 9.69 | 0.48 | 0.69 | 8.82 | 2.91 | 18.9 | 6.56 | n.a. | n.a. | 9 | 8.11 | 5.25 | 0.48 | 18.9 |
| Ag | 0.01 | 1.81 | 0.50 | 0.45 | 0.22 | 0.39 | n.a. | 0.57 | n.a. | n.a. | n.a. | n.a. | 6 | 0.48 | 0.26 | 0.22 | 1.81 |
| Ba | 1.00 | 210 | 1629 | 1459 | 1994 | 5025 | 19.0 | 451 | 481 | 135 | n.a. | n.a. | 9 | 908 | 761 | 19.0 | 5025 |
| Pb | 0.80 | 18.5 | 3.60 | 46.7 | 14.7 | 9.30 | bdl | 9.28 | 8.58 | 2.77 | n.a. | n.a. | 9 | 9.53 | 6.19 | <0.8 | 46.7 |
| Th | 1.00 | 11.2 | 9.40 | 9.20 | 7.10 | 5.20 | bdl | 8.60 | 11.9 | 8.30 | n.a. | n.a. | 9 | 8.43 | 3.46 | <1.0 | 11.9 |
| U | 1.00 | 23.5 | 14.3 | 24.5 | bdl | bdl | 12.0 | 13.7 | 26.7 | 2.39 | n.a. | n.a. | 9 | 13.1 | 9.15 | <1.0 | 26.7 |

LLD = low limit detection; n.a. = not analyzed; bdl = below detection limit; *n* = number of samples; Mean = average value; *S* = standard deviation; Min = minimum value; Max = maximum value. * All Fe as $Fe_2O_3$. I is 37.5 ppm in sample Y-8; Cd, In, Sn, Sb, Cs, Te are <1 ppm (LLD).

**Table 8.** Bulk (in wt %) and trace element (in ppm) compositions of Hatrurim mayenite–larnite and gehlenite-larnite CM rocks.

| Sample | LLD | Y-7 | CONCR | Y-2-1 | W-10-2 | W-12-1 | M4-218 | G-7 | M4-215 | M5-32 | *n* | Mean | *S* | Min | Max | Y-10-5 | W-11-3 |
|---|---|---|---|---|---|---|---|---|---|---|---|---|---|---|---|---|---|
| | | | | | | | Bulk composition (in wt %) | | | | | | | | | | |
| $SiO_2$ | 0.25 | 19.8 | 25.5 | 22.5 | 24.9 | 22.4 | 27.1 | 24.4 | 26.8 | 24.5 | 9 | 24.4 | 1.60 | 19.8 | 27.1 | 26.4 | 25.1 |
| $TiO_2$ | 0.10 | 0.35 | 0.47 | 0.38 | 0.44 | 0.42 | 0.45 | 0.46 | 0.38 | 0.41 | 9 | 0.42 | 0.03 | 0.35 | 0.47 | 0.37 | 0.47 |
| $Al_2O_3$ | 0.25 | 8.17 | 11.6 | 9.23 | 11.0 | 9.59 | 12.5 | 8.67 | 12.5 | 11.7 | 9 | 10.6 | 1.45 | 8.17 | 12.5 | 9.86 | 10.8 |
| $Fe_2O_3$ | 0.20 | 2.94 | 4.51 | 3.29 | 2.08 | 2.55 | 3.40 | 3.26 | 2.21 | 4.56 | 9 | 3.17 | 0.73 | 2.08 | 4.56 | 2.92 | 4.68 |
| FeO | 0.10 | 0.47 | n.a. | 1.03 | 0.67 | 0.89 | n.a. | 0.94 | n.a. | n.a. | 5 | 0.83 | 0.45 | 0.47 | 1.03 | 0.47 | 0.62 |
| MnO | 0.01 | 0.03 | 0.14 | bdl | bdl | bdl | 0.13 | bdl | 0.13 | 0.13 | 9 | 0.06 | 0.06 | <0.01 | 0.14 | 0.04 | 0.02 |
| MgO | 0.20 | 0.94 | 0.71 | 0.80 | 0.78 | 0.66 | 0.75 | 0.86 | 0.70 | 0.51 | 9 | 0.75 | 0.07 | 0.51 | 0.94 | 1.77 | 0.68 |
| CaO | 0.25 | 51.2 | 51.1 | 56.7 | 54.0 | 55.8 | 51.1 | 55.6 | 52.2 | 52.8 | 9 | 53.3 | 1.93 | 51.1 | 56.7 | 47.9 | 48.6 |
| $N.a._2O$ | 0.05 | 0.13 | 0.75 | 0.24 | 0.45 | 0.44 | 0.27 | 0.35 | 0.15 | 0.43 | 9 | 0.33 | 0.12 | 0.13 | 0.75 | 0.22 | 0.33 |
| $K_2O$ | 0.05 | 0.07 | 0.49 | 0.13 | 0.54 | 0.22 | 0.27 | 0.97 | 0.17 | 0.86 | 9 | 0.38 | 0.26 | 0.07 | 0.97 | 0.20 | 0.69 |
| $P_2O_5$ | 0.03 | 2.19 | 2.58 | 2.08 | 2.20 | 2.28 | 2.27 | 2.30 | 2.49 | 2.62 | 9 | 2.33 | 0.15 | 2.08 | 2.62 | 0.98 | 2.22 |
| $CO_2$ | 0.06 | 0.98 | 1.15 | 1.01 | 0.65 | 1.06 | 0.89 | 0.73 | 0.89 | 0.62 | 9 | 0.89 | 0.15 | 0.62 | 1.15 | 0.83 | 0.13 |
| F | 0.03 | 0.19 | n.a. | 0.24 | 0.19 | 0.21 | n.a. | 0.09 | n.a. | n.a. | 5 | 0.20 | 0.11 | 0.09 | 0.24 | bdl | bdl |
| $SO_3$ | 0.05 | 0.73 | 0.60 | 0.38 | 0.20 | 0.18 | 0.11 | 0.05 | 0.03 | 0.02 | 9 | 0.22 | 0.20 | <0.05 | 0.73 | 0.25 | 0.13 |
| $H_2O$ | 0.05 | 12.1 | 0.67 | 2.19 | 1.69 | 2.95 | 1.16 | 1.25 | 1.47 | 1.23 | 9 | 1.71 | 0.65 | 0.67 | 12.1 | 8.10 | 5.50 |
| Total | - | 100.3 | 100.3 | 100.2 | 99.8 | 99.6 | 100.4 | 99.9 | 100.1 | 100.5 | - | - | - | - | - | 100.3 | 100 |

**Table 8.** *Cont.*

| Sample | LLD | Y-7 | CONCR | Y-2-1 | W-10-2 | W-12-1 | M4-218 | G-7 | M4-215 | M5-32 | *n* | Mean | *S* | Min | Max | Y-10-5 | W-11-3 |
|---|---|---|---|---|---|---|---|---|---|---|---|---|---|---|---|---|---|
| | | | | | | Trace element composition (in ppm) | | | | | | | | | | | |
| B | 1.00 | 16.8 | 75.8 | 15.4 | 28.8 | 11.9 | 31.4 | 17.5 | n.a. | 103 | 8 | 31.0 | 24.0 | 11.9 | 103 | 38.8 | 60.2 |
| V | 2.00 | 24.2 | 55.9 | 41.8 | 42.7 | 52.8 | 46.6 | 47.5 | 64.9 | 60.3 | 9 | 49.7 | 6.91 | 24.2 | 64.9 | 63.7 | 51.0 |
| Cr | 1.50 | 135 | 361 | 234 | 220 | 198 | 250 | 284 | 202 | 284 | 9 | 239 | 35.6 | 135 | 361 | 44.9 | 123 |
| Ni | 1.00 | 93.0 | 230 | 95.0 | 60.8 | 163 | 58.0 | 175 | 64.4 | 263 | 9 | 126 | 64.1 | 58.0 | 263 | 69.0 | 139 |
| Cu | 1.00 | 23.9 | 86.0 | 18.0 | 11.7 | 26.8 | 23.8 | 100 | 16.1 | 41.9 | 9 | 33.8 | 24.5 | 11.7 | 100 | 24.4 | 80.0 |
| Zn | 1.00 | 100 | 445 | 71.3 | 191 | 26.5 | 161 | 55.1 | 42.0 | 210 | 9 | 119 | 68.2 | 26.5 | 445 | 44.6 | 134 |
| Ga | 0.70 | 6.20 | 8.70 | 6.44 | 8.40 | 7.55 | 10.3 | 4.79 | 8.17 | 8.64 | - | 9 | 7.73 | 1.04 | 4.79 | 10.3 | - |
| Ge | 0.20 | bdl | 1.84 | 0.72 | bdl | 0.54 | 2.30 | 0.96 | 2.61 | 1.58 | - | 9 | 1.16 | 0.82 | <0.2 | 2.61 | - |
| As | 0.80 | 9.90 | 10.7 | 31.3 | 12.5 | 17.7 | 22.8 | 7.00 | 33.9 | 24.1 | - | 9 | 18.4 | 8.01 | 7.00 | 33.9 | - |
| Se | 0.20 | 7.55 | 5.71 | 2.82 | bdl | 3.87 | bdl | 1.24 | 9.71 | bdl | - | 9 | 3.08 | 2.83 | <0.2 | 9.71 | - |
| Rb | 0.20 | 2.57 | 8.59 | 2.74 | 4.44 | 2.30 | 2.00 | 4.73 | 1.85 | 9.86 | - | 9 | 3.91 | 2.32 | 1.85 | 9.86 | - |
| Sr | 1.00 | 1146 | 1880 | 1454 | 1547 | 1453 | 1830 | 1550 | 1465 | 1843 | - | 9 | 1592 | 172 | 1146 | 1880 | - |
| Y | 0.20 | 55.7 | 42.5 | 30.7 | 32.0 | 36.6 | 39.1 | 31.2 | 43.0 | 41.0 | - | 9 | 37.9 | 4.82 | 30.7 | 55.7 | - |
| Zr | 0.50 | 57.9 | 78.0 | 57.9 | 59.0 | 58.0 | 69.4 | 75.0 | 51.9 | 72.2 | - | 9 | 64.2 | 7.67 | 51.9 | 78.0 | - |
| Nb | 0.20 | 6.32 | 7.64 | 3.10 | 6.54 | 4.49 | 6.48 | 10.7 | 6.05 | 7.71 | - | 9 | 6.46 | 1.08 | 3.10 | 10.7 | - |
| Mo | 0.20 | 1.27 | 23.2 | 3.16 | 5.63 | 15.0 | 8.22 | 6.22 | 13.4 | 20.2 | - | 9 | 10.3 | 6.10 | 1.27 | 23.2 | - |
| Ag | 0.01 | 0.65 | 0.67 | 0.05 | 0.03 | 0.02 | 2.89 | 0.09 | n.a. | 0.43 | - | 8 | 0.32 | 0.30 | 0.02 | 2.89 | - |
| Ba | 1.00 | 26.5 | 290 | 101 | 225 | 428 | 56.0 | 103 | 47.2 | 350 | - | 9 | 168 | 121 | 26.5 | 428 | - |
| Pb | 0.80 | 1.50 | 2.30 | 4.20 | 2.10 | bdl | bdl | 5.80 | bdl | bdl | - | 9 | 1.79 | 1.31 | <0.8 | 5.80 | - |
| Th | 1.00 | 5.00 | bdl | 7.10 | 8.70 | 6.70 | 7.50 | 12.6 | 6.78 | 9.80 | - | 9 | 7.37 | 2.97 | <1.0 | 12.60 | - |
| U | 1.00 | 7.20 | 22.0 | bdl | 4.80 | 8.40 | 6.00 | 62.4 | 6.00 | 19.1 | - | 9 | 10.5 | 7.47 | <1.0 | 62.4 | - |

LLD = low limit of detection; n.a. = not analyzed; bdl = below detection limit; n = number of samples; Mean = average value; S = standard deviation; Min = minimum value; Max = maximum value. * All Fe as $Fe_2O_3$. I is 3 ppm in sample CONCR; Cd, In, Sn, Sb, Cs, Te <1 ppm (LLD). Mayenite–larnite CM rocks: Y-7, CONCR, Y-2-1, W-10-2, W-12-1, M4-218, G-7, M4-215, M5-32. Gehlenite-larnite CM rocks: Y-10-5, W-11-3..

### 4.2. Mineralogy of Hatrurim Fm. Larnite CM Rocks

Fresh $Ca_2SiO_4$-bearing rocks have deep brown or grey to black colors, fine grain sizes, a massive structure and high density, and are viscous and homogeneous (Figure 2c). Rocks of the **ye'elimite–larnite** variety are microcrystalline (1–30 μm), with rather uniformly distributed minerals (Figure 6): 40–50% larnite, 7–15% opaque minerals (predominant brownmillerite and less abundant magnesioferrite), 25–15% Ca phosphate–sulphates, and 15–20% ye'elimite (Table 2). Fluorellestadite–fluorapatite and brownmillerite occur as large (up to 200 μm) crystals and concretions (Figures 7 and 8). All minerals, except the opaques, host numerous mineral inclusions and, in their turn, may be enclosed in any other mineral. Usually, the rocks contain sporadic grains of α'-$Ca_2SiO_4$ and gehlenite. Few small grains of hatrurite (alite) and merwinite have been identified in a single sample. Shulamitite–sharyginite, Fe–Mg spinel, Fe-enriched periclase, and Cr-bearing barite are the main accessory phases. Other identified phases include sporadic tiny grains of eucairite, $Cu_2Se$, vorlanite, lakargiite, and arcanite. Some ye'elimite grains are decomposed and replaced by $Al(OH)_3$ (Table 3). Tobermorite is present in trace amounts. Opaque minerals are usually fresh or slightly altered. Secondary ($CrO_4$)-bearing ettringite with afwillite fills thin cracks in one sample (Figure 9). Upon further alteration, ettringite (within 1.5 wt % $Cr_2O_3$) develops into chromatite ($CaCrO_4$) which is found as the latest phase (Figure 9I). Details of the process were discussed previously by Sokol et al. (2011) [86].

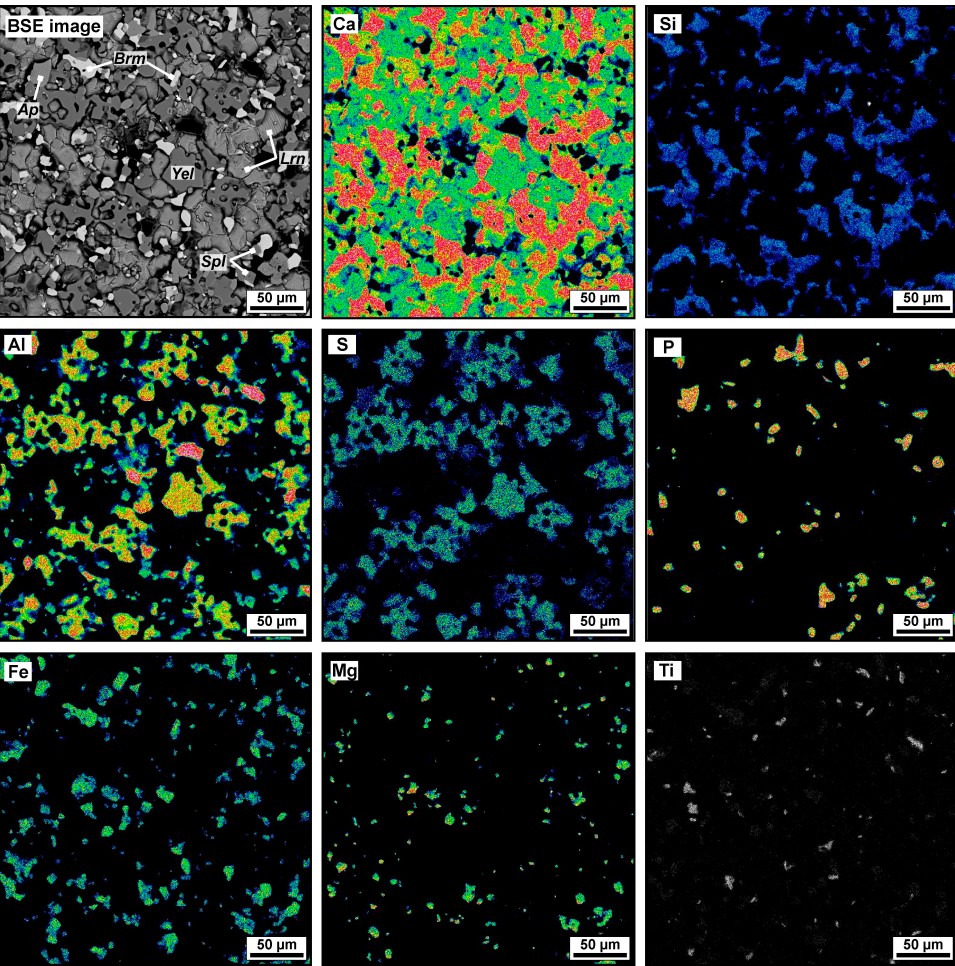

**Figure 6.** BSE image and elemental maps (Ca, Si, Al, S, P, Fe, Mg, Ti) of typical ye'elimite–larnite CM rock: uniformly distributed main, minor, and accessory phases. Sample MP-10-1. Mineral names are abbreviated as Ap = fluorapatite, Brm = brownmillerite, Lrn = larnite, Spl = spinel, Yel = ye'elimite.

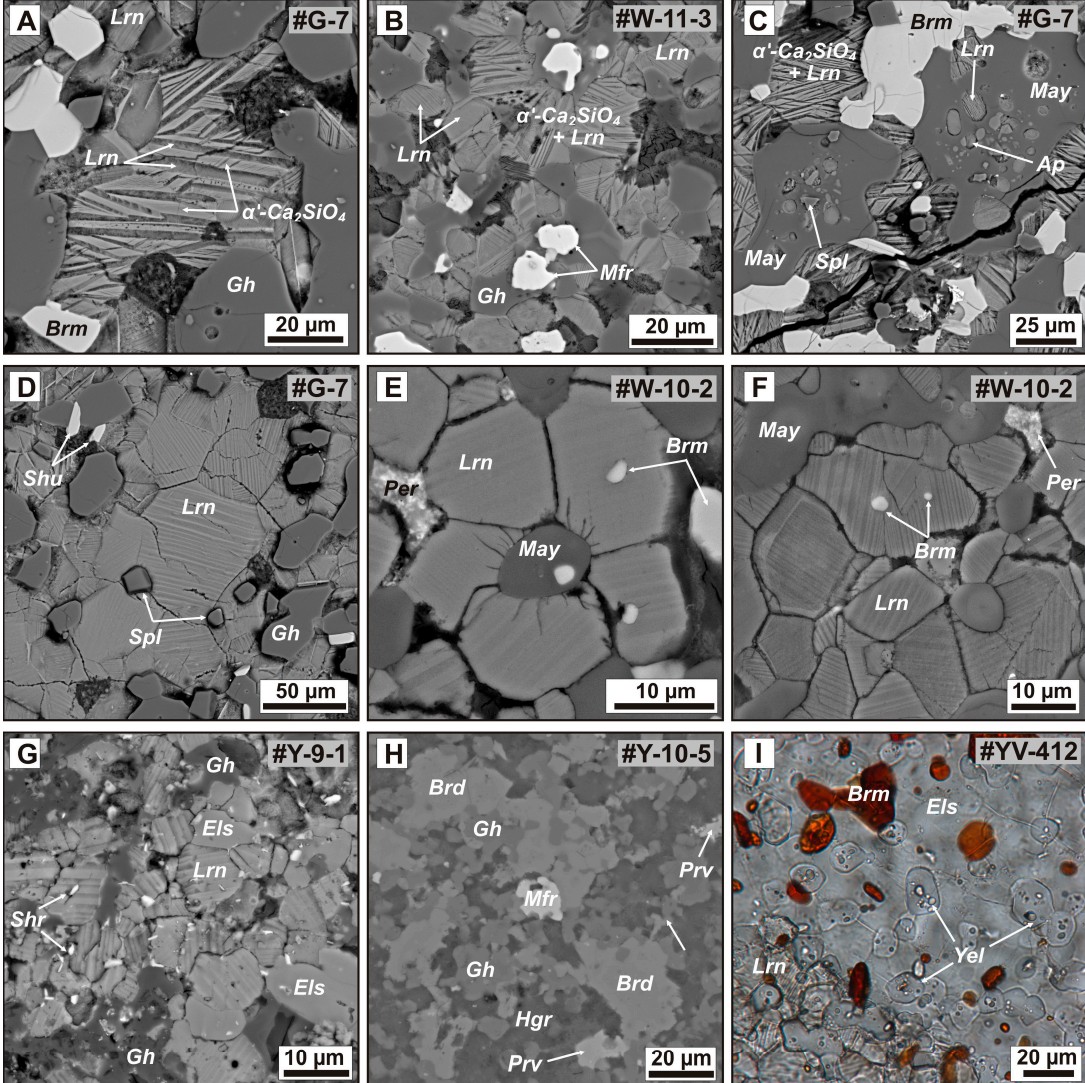

**Figure 7.** Morphological diversity of the main phases in ye'elimite– and mayenite-bearing larnite CM rocks. (**A**) Close association of round larnite grains with {100} twinning and aggregates of larnite (β-Ca$_2$SiO$_4$) lamelli in the flamite (α'-Ca$_2$SiO$_4$) matrix. The α'-Ca$_2$SiO$_4$ and β-Ca$_2$SiO$_4$ lamelli are oriented as {11$\bar{2}$0} α' ∥ {100}$_β$ and <0001>$_{α'}$ ∥ <010>β. (**B**) Typical assemblage of Ca$_2$SiO$_4$-bearing rocks consisting of larnite, larnite (β-Ca$_2$SiO$_4$) and flamite (α'-Ca$_2$SiO$_4$) aggregates, gehlenite, and magnesioferrite. (**C**) Mayenite grains filled with larnite, fluorapatite, and spinel inclusions. Mayenite associated with brownmillerite and larnite-flamite aggregates. (**D**) Large grains of larnite associated with gehlenite, spinel and scarce shulamitite. (**E**) Mayenite grain surrounded by larnite with radiating cracks. (**F**) Mosaic of anhedral larnite in association with mayenite, brownmillerite, and periclase. (**G**) Intergrown rock-forming gehlenite, larnite, and fluorellestadite with numerous fine grains of accessory sharyginite. (**H**) Local area of bredigite-enriched larnite CM rock. (**I**) Large fluorellestadite crystals filled with ye'elimite and brownmillerite inclusions. A–H = back-scattered electron (BSE) images; I = photomicrograph in polarized transmitted light. Mineral names are abbreviated as Ap = fluorapatite, Brd = bredigite, Brm = brownmillerite, Els = fluorellestadite, Hgr = hydrogarnet, Gh = gehlenite, Lrn = larnite, May = mayenite supergroup minerals, Mfr = magnesioferrite, Per = periclase, Prv = perovskite, Shr = sharyginite, Spl = spinel, Yel = ye'elimite.

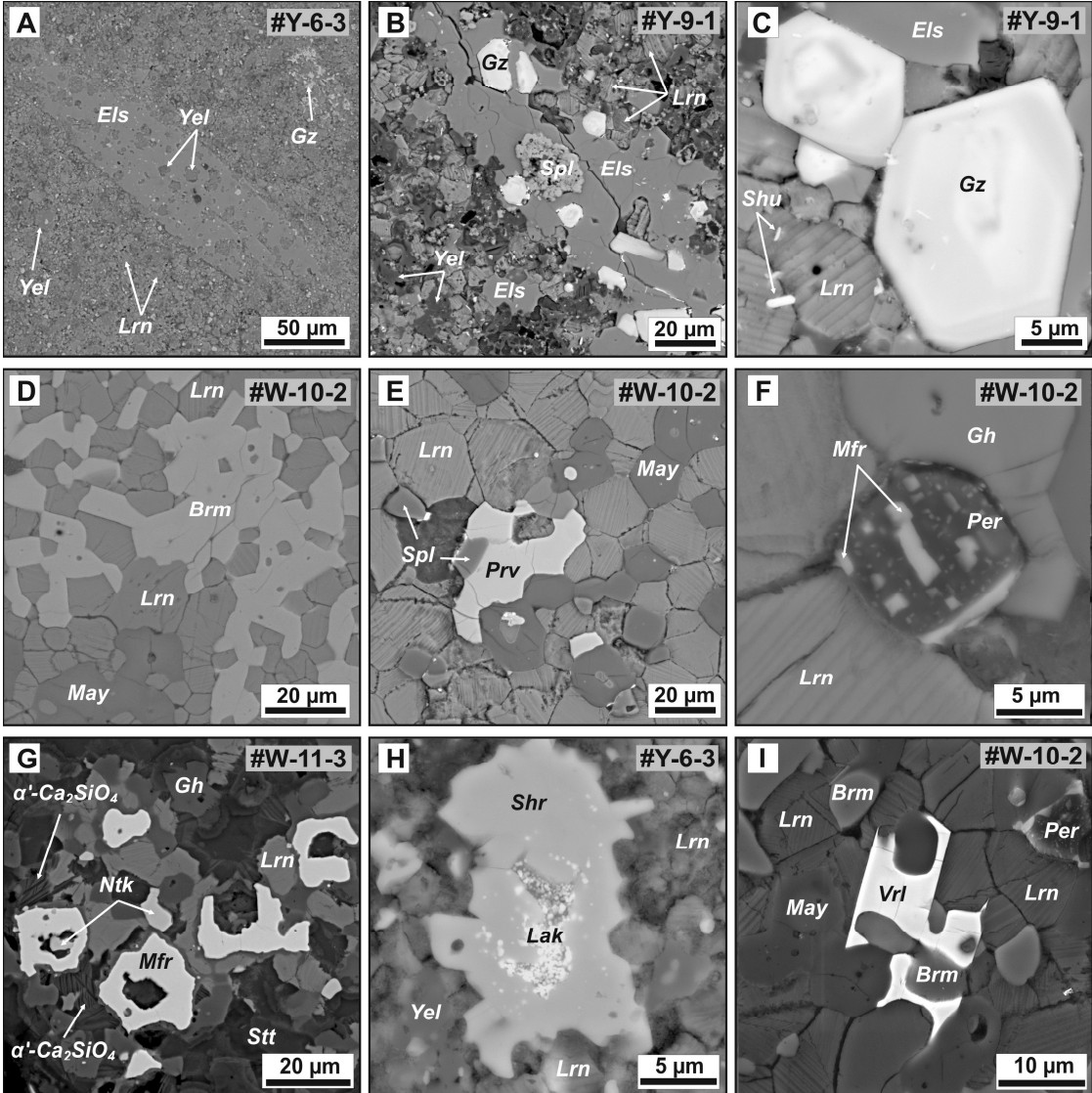

**Figure 8.** BSE images of accessory minerals in ye'elimite- and mayenite-bearing larnite CM rocks. Nest of anhedral grains (**A**) and small faceted crystals (**B**,**C**) of gazeevite associated with larnite, fluorellestadite, ye'elimite, shulamitite, and spinel. (**D**) Mosaic intergrowth of larnite and brownmillerite. (**E**) Anhedral segregation of perovskite and spinel grains in mayenite–larnite rock. (**F**) Magnesioferrite microinclusions in periclase. (**G**) Close association of magnesioferrite and nataliakulikite in gehlenite–larnite rock. (**H**) Ultrafine lakargiite inclusions in sharyginite. (**I**) Grains of vorlanite in interstitials between larnite, mayenite and brownmillerite. Mineral names are abbreviated as Brm = brownmillerite, Els = fluorellestadite, Gh = gehlenite, Gz = gazeevite, Lak = lakargiite, Lrn = larnite, May = mayenite supergroup minerals, Mfr = magnesioferrite, Ntk = nataliakulikite, Per = periclase, Prv = perovskite, Shr = sharyginite, Spl = spinel, Vrl = vorlanite, Yel = ye'elimite.

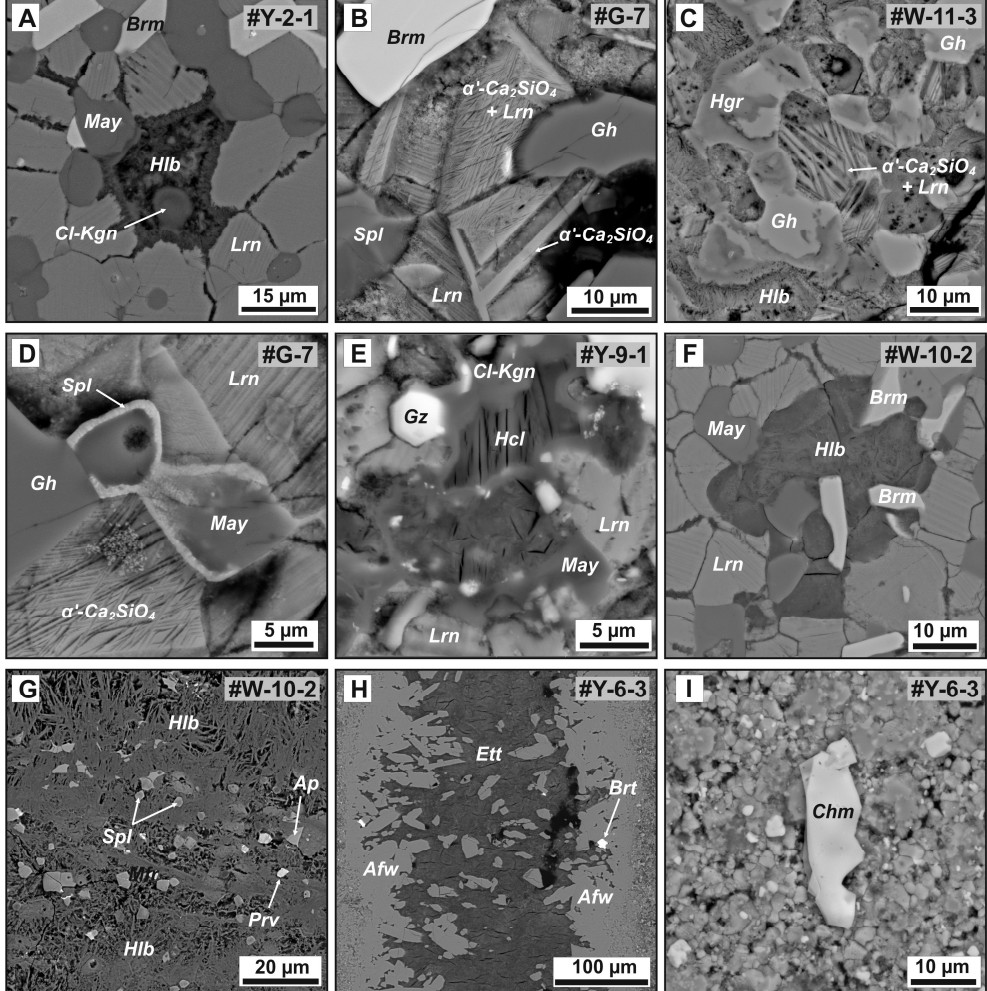

**Figure 9.** Morphology and distribution of secondary minerals in larnite CM rocks. BSE images. (**A–D**) Microstructure of slightly hydrated (etched) larnite with a single set of polysynthetic twins and a lamellar aggregate of larnite and flamite; pore space filled with hillebrandite and chlorkyuygenite; gehlenite grains partially replaced by hydrogarnet. (**E**) Aggregate of chlorkyuygenite and hydrocalumite after mayenite. (**F**) Vugs filled with hillebrandite. (**G**) Aggregate of hillebrandite fibers with fresh fluorapatite, spinel, and perovskite. (**H**) Ettringite and afwillite filling a crack in larnite rock. (**I**) Small particle of supergene chromatite. Mineral names are abbreviated as Ap = fluorapatite, Afw = afwillite, Brm = brownmillerite, Brt = barite, Chm = chromatite, Cl-Kgn = chlorkyuygenite, Ett = ettringite, Gh = gehlenite, Gz = gazeevite, Hcl = hydrocalumite, Hgr = hydrogarnet, Hlb = hillebrandite, Lrn = larnite, May = mayenite supergroup minerals, Prv = perovskite, Spl = spinel.

**Mayenite–larnite rocks** have slightly coarser grain sizes (15–50 µm) and a different mineralogy, with up to 74% of rock-forming larnite, 18% of gehlenite, and 10–15% of opaque (mainly brownmillerite and Fe–Mg spinel) and mayenite–supergroup (mainly fluorkyuygenite) minerals, as well as accessory shulamitite–sharyginite (Table 2) and minor percentages of fluorapatite (<3%) and trace amounts of ye'elimite, but without fluorellestadite/fluorapatite and silicocarnotite. Unlike the ye'elimite variety, the mayenite-bearing rocks commonly contain $\alpha'_L$-Ca$_2$SiO$_4$ and $\alpha'_H$-Ca$_2$SiO$_4$ modifications (up to 16.5% in total) (Figure 7A,C), and their sulfate contents correlate with the percentages of β-Ca$_2$SiO$_4$ which appears instead of α-modifications, as was previously inferred for industrial clinkers [87].

The accessory phases are commonly Fe–Mg spinel, perovskite, periclase, barite-hashemite solid solution, and vorlanite (Figure 8). One sample contains pyrrhotite, oldhamite, and K–Fe sulfide. In these rocks, larnite commonly keeps fresh while fluormayenite is markedly hydrated and replaced

by fluorkyuygenite. Further hydration produces 50–120 μm nests of secondary phases in place of mayenite grains (namely hydrocalumite ($Ca_2Al(OH)_6(Cl_{1-x}(OH)_x)\cdot 3(H_2O)$) and chlorkyuygenite) and involves also the neighbor grains of other minerals. Hydration follows cracks but is weaker there than at local isometric sites where secondary products are nested (Figure 9). Alteration products are hydrogarnet and CSHs with high Ca:Si ratios (Table 3); calcite forms crusts upon pseudo-pebbles.

Larnite CM rocks with predominant gehlenite (up to 53%) are rare within the Hatrurim Basin. They have a simpler phase composition and are more strongly altered than both ye'elimite– and mayenite-bearing varieties (Table 2; Figures 7B and 8G). The CSHs (Na and Al-bearing) and hydrogarnet with trace amounts of portlandite, ettringite and afwillite formed at the expense of primary clinker phases (Table 3). The cores of gehlenite grains and bredigite may have remained intact, but the gehlenite rims are commonly hydrated (Figure 9C). Main opaque phases are magnesioferrite and nataliakulikite (Figure 8G); other accessories are fluorapatite, Ti-andradite, barite and pyrrhotite.

**Hatrurite** ($Ca_3SiO_5$), a natural analog of alite, an important phase of ordinary Portland cement clinker, has a composition close to $Ca_3SiO_5$, with 51.8 wt % Ca, 11.1 wt % Si, 1.1 wt% Al, 0.4 wt % Mg, 0.6 wt % P, and 35.0 wt % O. It is extremely rare in the Hatrurim Fm clinker-like CM rocks: we found it in a single sample of our collection as few ≤10 μm vuggy grains. This rarity cannot be due to the temperature of metamorphism as the $Ca_3SiO_5$ phase is stable within the range from 2150 to 1250 °C [20], while the carbonate protolith was most often exposed to 1300–1250 °C during CM events [4,40,51,54]. Hatrurite may be lacking from the high-temperature Hatrurim CM rocks for two reasons: (1) decomposition of $Ca_3SiO_5$ into CaO and $Ca_2SiO_4$ during gradual cooling (without quenching above 1250 °C); (2) extremely high hydration reactivity of $Ca_3SiO_5$ which thus cannot survive prolonged alteration processes. Previously, Gross [1] found out experimentally that hatrurite from larnite rocks was prone to hydration for half an hour at room temperature with $Ca(OH)_2$ release into the solution. Till recently, the mineral was thought to be endemic to the Hatrurim Fm, but it was also discovered in the Eifel metacarbonate (Ettringen, Germany) (www.mindat.org/min-1828.html).

**Ca disilicates** occur in the Hatrurim Fm CM rocks as two polymorphic modifications: rock-forming larnite ($\beta$-$Ca_2SiO_4$) and subordinate $\alpha'$-$Ca_2SiO_4$ (flamite) (Table 2).

**Larnite** is a main phase (23–73%) and is similar in appearance to type II belite in production clinkers. It has round grains, typically 20 to 45 μm, with a single set of distinct striation (Figure 7D–G, Figure 8E,F and Figure 9D), which are polysynthetic twins after (100) or (010) forming by $\alpha'_L$-$Ca_2SiO_4$ to $\beta$-$Ca_2SiO_4$ transformation [20]. In thin sections, the mineral appears colorless or greyish. Its grains are commonly cracked, free from zonation, pores, or overgrowth structures.

Larnite of ideal $Ca_2SiO_4$ stoichiometry (≤0.05 wt % $P_2O_5$) was found in a few gehlenite-bearing samples only (Table 9, Table 10 and Table S1). Silica is commonly notably below the theoretical values (33.36 ± 0.77 wt % and 32.70 ± 0.50 wt % $SiO_2$ in ye'elimite and mayenite varieties, respectively) and is often substituted by $P_2O_5$ (0.38–3.33 wt %). Aluminum occurs as impurity in the greatest part of the analyzed larnite grains (≤0.80 wt % $Al_2O_3$, with average contents of 0.21 and 0.16 wt %), while sulfur is above the detection limit in a few grains (≤0.39 wt % $SO_3$, 0.05 and <0.03 wt % on average). The lack of significant correlation between Al and S speaks against their conjugate incorporation into the structure of natural larnite akin to clinker belite ($2Al^{3+} + S^{6+} \rightarrow 3Si^{4+}$) [20]. The CaO content in natural larnite is also below the $Ca_2SiO_4$ stoichiometry (64.13 ± 0.72 wt % and 63.71 ± 0.88 wt % in ye'elimite and mayenite rocks, respectively). The elements that substitute for Ca form a descending series of Na, K, Sr, Mg, and Ba, while $K^+$ and $Mg^{2+}$ mainly substitute for $Ca^{2+}$ in clinker belite [20]. $Na_2O$ in natural larnite reaches 1.25 wt % and correlates with the bulk rock content, and $K_2O$ is the highest (1.35 wt %) in larnite from mayenite-bearing rocks, whereas larnite from the ye'elimite-bearing variety is generally poorer in both $Na_2O$ (0.44 wt % against 0.62 wt %) and $K_2O$ (0.09 wt % against 0.44 wt %). The mayenite- and ye'elimite-bearing rocks have notably different average larnite compositions: $(Ca_{1.94}Na_{0.04}K_{0.02})(Si_{0.94}P_{0.05}Al_{0.01})$ and $(Ca_{1.97}Na_{0.02})(Si_{0.96}P_{0.03}Al_{0.01})$ (Table 10).

**Table 9.** Mineral chemistry (WDS-EDS, wt %) of larnite and $\alpha'$-Ca$_2$SiO$_4$ phase from Hatrurim larnite CM rocks, compared with other Ca$_2$SiO$_4$ polymorphs.

| Type | Sample | Polymorph | n | SO$_3$ | P$_2$O$_5$ | SiO$_2$ | TiO$_2$ | Al$_2$O$_3$ | FeO | MgO | CaO | SrO | Na$_2$O | K$_2$O | Total | S | P | Si | Ti | Al | Fe | Mg | Ca | Sr | Na | K |
|---|---|---|---|---|---|---|---|---|---|---|---|---|---|---|---|---|---|---|---|---|---|---|---|---|---|---|
| Y | Y-5-1 | Larnite | 2 | n.a. | 0.46 | 34.58 | n.a. | 0.37 | 0.64 | n.a. | 63.62 | n.a. | 0.39 | n.a. | 100.05 |  | 0.01 | 0.99 |  | 0.01 | 0.02 |  | 1.95 |  | 0.02 | 0.00 |
| Y | Y-6-3 | Larnite | 3 | n.a. | 0.82 | 34.49 | n.a. | 0.32 | 0.47 | <0.02 | 63.29 | n.a. | 0.39 | 0.11 | 99.89 |  | 0.02 | 0.99 |  | 0.01 | 0.01 | 0.00 | 1.94 |  | 0.02 | 0.00 |
| Y | M5-30 | Larnite | 3 | 0.10 | 0.98 | 32.98 | 0.11 | 0.63 | 0.11 | <0.02 | 64.30 | 0.30 | 0.19 | 0.04 | 99.75 | 0.00 | 0.02 | 0.95 | 0.00 | 0.02 | 0.00 | 0.00 | 1.98 | 0.00 | 0.01 | 0.00 |
| Y | M5-31 | Larnite | 6 | 0.10 | 1.88 | 32.59 | 0.07 | 0.13 | 0.06 | <0.02 | 63.75 | 0.04 | 0.78 | 0.08 | 99.49 | 0.00 | 0.05 | 0.94 | 0.00 | 0.00 | 0.00 | 0.00 | 1.97 | 0.00 | 0.04 | 0.00 |
| Y | M4-217 | Larnite | 4 | 0.04 | 0.97 | 33.41 | 0.09 | 0.31 | 0.07 | <0.02 | 64.30 | 0.21 | 0.28 | 0.09 | 99.76 | 0.00 | 0.02 | 0.96 | 0.00 | 0.01 | 0.00 | 0.00 | 1.98 | 0.00 | 0.02 | 0.00 |
| M | Y-9-1 | Larnite | 3 | n.a. | 0.32 | 34.95 | n.a. | 0.46 | 0.37 | 0.24 | 63.14 | n.a. | 0.34 | 0.23 | 100.05 |  | 0.01 | 1.00 |  | 0.02 | 0.01 | 0.01 | 1.93 |  | 0.02 | 0.01 |
| M | G-7 | Larnite | 4 | n.a. | 2.12 | 31.68 | n.a. | 0.20 | 0.08 | 0.04 | 64.22 | 0.25 | 0.41 | 0.79 | 99.77 |  | 0.05 | 0.92 |  | 0.01 | 0.00 | 0.00 | 1.99 | 0.00 | 0.02 | 0.03 |
| M | G-7 | $\alpha'$-Ca$_2$SiO$_4$ | 5 | n.a. | 7.85 | 27.26 | n.a. | 0.11 | 0.03 | 0.10 | 59.90 | 0.29 | 0.65 | 3.08 | 99.26 |  | 0.19 | 0.79 |  | 0.00 | 0.00 | 0.00 | 1.85 | 0.00 | 0.04 | 0.11 |
| M | W-10-2 | Larnite | 5 | n.a. | 1.43 | 33.80 | n.a. | 0.32 | n.a. | n.a. | 63.52 | n.a. | 0.48 | 0.63 | 100.19 |  | 0.03 | 0.96 |  | 0.01 |  |  | 1.94 |  | 0.03 | 0.02 |
| M | W-12-1 | Larnite | 5 | n.a. | 2.15 | 32.29 | n.a. | 0.61 | 0.13 | 0.05 | 63.81 | 0.26 | 0.51 | 0.19 | 99.99 |  | 0.05 | 0.93 |  | 0.02 | 0.00 | 0.00 | 1.96 | 0.00 | 0.03 | 0.01 |
| M | W-12-1 | $\alpha'$-Ca$_2$SiO$_4$ | 2 | n.a. | 5.05 | 30.23 | n.a. | 0.25 | <0.02 | 0.06 | 62.97 | 0.30 | 0.63 | 0.18 | 99.65 |  | 0.12 | 0.86 |  | 0.01 | 0.00 | 0.00 | 1.93 | 0.00 | 0.03 | 0.01 |
| M | CONCR | Larnite | 24 | <0.03 | 2.09 | 33.03 | 0.08 | 0.11 | 0.02 | 0.05 | 63.11 | 0.19 | 1.07 | 0.10 | 99.84 | 0.00 | 0.05 | 0.95 | 0.00 | 0.00 | 0.00 | 0.00 | 1.94 | 0.00 | 0.06 | 0.00 |
| M | M5-32 | Larnite | 11 | 0.04 | 2.47 | 32.20 | 0.18 | 0.27 | 0.06 | <0.02 | 62.79 | 0.32 | 0.57 | 0.83 | 99.73 | 0.00 | 0.06 | 0.93 | 0.00 | 0.01 | 0.00 | 0.00 | 1.93 | 0.01 | 0.03 | 0.03 |
| M | M5-32 | $\alpha'$-Ca$_2$SiO$_4$ | 2 | <0.03 | 7.51 | 27.99 | 0.07 | 0.09 | 0.04 | <0.02 | 59.05 | 0.36 | 0.95 | 3.58 | 99.62 | 0.00 | 0.18 | 0.81 | 0.00 | 0.00 | 0.00 | 0.00 | 1.82 | 0.01 | 0.05 | 0.13 |
| M | M4-215 | Larnite | 4 | <0.03 | 0.91 | 33.10 | 0.09 | 0.19 | 0.02 | <0.02 | 65.19 | 0.01 | 0.23 | 0.19 | 99.92 | 0.00 | 0.02 | 0.95 | 0.00 | 0.01 | 0.00 | 0.00 | 2.01 | 0.00 | 0.01 | 0.01 |
| M | M4-218 | Larnite | 5 | <0.03 | 1.69 | 32.70 | 0.16 | 0.25 | 0.07 | <0.02 | 64.19 | 0.22 | 0.41 | 0.29 | 99.99 | 0.00 | 0.04 | 0.94 | 0.00 | 0.01 | 0.00 | 0.00 | 1.98 | 0.00 | 0.02 | 0.01 |
| M | M4-251 | Larnite | 4 | <0.03 | 1.80 | 32.75 | 0.19 | 0.22 | 0.05 | <0.02 | 63.64 | 0.27 | 0.31 | 0.75 | 99.96 | 0.00 | 0.04 | 0.94 | 0.00 | 0.01 | 0.00 | 0.00 | 1.96 | 0.00 | 0.02 | 0.03 |
| M | M4-251 | $\alpha'$-Ca$_2$SiO$_4$ | 1 | 0.10 | 4.83 | 31.54 | 0.02 | 0.17 | <0.02 | <0.02 | 61.37 | 0.28 | 0.35 | 1.15 | 99.84 | 0.00 | 0.12 | 0.90 | 0.00 | 0.01 | 0.00 | 0.00 | 1.87 | 0.00 | 0.02 | 0.04 |
| M | Y-2-1 | Larnite | 3 | n.a. | 0.96 | 33.77 | n.a. | <0.02 | <0.02 | n.a. | 64.67 | n.a. | 0.40 | <0.01 | 99.80 |  | 0.02 | 0.97 |  | 0.00 | 0.00 |  | 1.99 |  | 0.02 | 0.00 |
| M | H-401 | $\alpha'$-Ca$_2$SiO$_4$ | 22 | <0.03 | 4.64 | 31.05 | 0.02 | <0.02 | 0.07 | 0.03 | 61.83 | 0.21 | 0.41 | 1.46 | 99.73 | 0.00 | 0.11 | 0.89 | 0.00 | 0.00 | 0.00 | 0.00 | 1.90 | 0.00 | 0.02 | 0.05 |
| Gh | Y-10-5 | Larnite | 1 | n.a. | 0.05 | 34.55 | n.a. | n.a. | n.a. | n.a. | 64.54 | n.a. | n.a. | n.a. | 99.14 |  | 0.00 | 1.00 |  |  |  |  | 2.00 |  |  |  |
| Gh | W-11-3 | Larnite | 26 | n.a. | 2.26 | 33.03 | n.a. | 0.11 | 0.10 | <0.02 | 63.22 | 0.27 | 0.65 | 0.65 | 100.31 |  | 0.05 | 0.94 |  | 0.00 | 0.00 | 0.00 | 1.93 | 0.00 | 0.04 | 0.02 |
| Gh | W-11-3 | $\alpha'$-Ca$_2$SiO$_4$ | 18 | n.a. | 6.98 | 29.14 | n.a. | 0.14 | 0.24 | <0.02 | 59.54 | 0.37 | 0.94 | 2.49 | 99.85 |  | 0.17 | 0.83 |  | 0.00 | 0.01 | 0.00 | 1.82 | 0.01 | 0.05 | 0.09 |
| PL-1 | YV-402 | Flamite | 21 | n.a. | 7.38 | 28.87 | n.a. | 0.04 | 0.15 | 0.16 | 59.76 | 0.24 | 1.55 | 1.73 | 100.03 |  | 0.18 | 0.82 |  | 0.00 | 0.00 | 0.01 | 1.82 | 0.00 | 0.09 | 0.06 |
| PL-2 | G2-G4 | $\alpha$-Ca$_2$SiO$_4$ | 82 | n.a. | 10.43 | 26.44 | 0.01 | 0.02 | 0.27 | 0.14 | 58.29 | 0.29 | 1.60 | 2.40 | 99.90 |  | 0.25 | 0.75 | 0.00 | 0.00 | 0.01 | 0.01 | 1.77 | 0.00 | 0.09 | 0.09 |

Formula is based on four oxygens. MnO and BaO are below the detection limits (<0.05 wt %). n = number of analyses; n.a. = not analyzed. Y = ye'elimite–larnite rocks; M = mayenite–larnite rocks; Gh = gehlenite-rich larnite rocks; PL-1 = holotype flamite (ideally (Na,K)Ca$_9$(SiO$_4$)$_4$(PO$_4$), $\alpha'_L$-Ca$_2$SiO$_4$) from Hatrurim larnite-gehlenite-rankinite paralava [52] (total includes 0.10 wt % V$_2$O$_5$ and 0.05 wt % BaO); PL-2—"nagelschmidtite" inclusions (ideally (Na,K)Ca$_7$(SiO$_4$)$_3$(PO$_4$), $\alpha$-Ca$_2$SiO$_4$) in Ti–andradite, rankinite and melilite from Ti-andradite-rich paralava, Gurim, Hatrurim Basin [64]. Data for some samples YV-410, 411, 412 are from [4,32]. See Supplementary (Table S1) for a complete chemical database of Ca$_2$SiO$_4$ phases from studied larnite CM rocks.

**Table 10.** Average composition of larnite, $\alpha'$-Ca$_2$SiO$_4$(ss) and bredigite from the Hatrurim larnite rocks (WDS-EDS, wt %).

| Mineral | Larnite ($n = 41$) | | | | Larnite ($n = 46$) | | | | $\alpha'$-Ca$_2$SiO$_4$(ss) ($n = 50$) | | | | Bredigite ($n = 15$) | | | |
|---|---|---|---|---|---|---|---|---|---|---|---|---|---|---|---|---|
| Rock | Ye'elimite Rocks | | | | Mayenite Rocks | | | | Ye'elimite and Mayenite Rocks | | | | Gehlenite Rocks | | | |
| | Mean | S | Min | Max | Mean | S | Min | Max | Mean | S | Min | Max | Mean | S | Min | Max |
| SiO$_2$ | 33.36 | 0.77 | 31.92 | 35.13 | 32.70 | 0.50 | 31.68 | 34.95 | 29.84 | 1.35 | 27.34 | 32.45 | 34.00 | 0.83 | 32.4 | 35.65 |
| TiO$_2$ | 0.09 | 0.06 | 0.04 | 0.39 | 0.12 | 0.07 | <0.02 | 0.23 | 0.08 | 0.07 | <0.02 | 0.25 | 0.11 | 0.12 | <0.02 | 0.33 |
| Cr$_2$O$_3$ | <0.02 | 0.03 | <0.02 | 0.12 | 0.04 | 0.05 | <0.02 | 0.15 | 0.14 | 0.26 | <0.02 | 0.70 | <0.02 | 0.01 | <0.02 | 0.03 |
| Al$_2$O$_3$ | 0.21 | 0.17 | <0.02 | 0.80 | 0.16 | 0.12 | <0.02 | 0.61 | 0.08 | 0.33 | <0.02 | 2.10 | 0.58 | 0.73 | <0.02 | 2.19 |
| FeO | 0.09 | 0.11 | <0.02 | 0.71 | 0.06 | 0.13 | <0.02 | 0.69 | 0.05 | 0.12 | <0.02 | 0.63 | 0.21 | 0.20 | <0.02 | 0.54 |
| MgO | 0.05 | 0.06 | <0.02 | 0.31 | 0.04 | 0.08 | <0.02 | 0.24 | 0.04 | 0.05 | <0.02 | 0.10 | 5.59 | 0.27 | 5.06 | 6.12 |
| CaO | 64.13 | 0.72 | 62.23 | 65.71 | 63.23 | 0.88 | 61.82 | 65.50 | 59.86 | 1.78 | 56.36 | 62.97 | 58.26 | 0.55 | 57.14 | 58.98 |
| SrO | 0.07 | 0.11 | <0.04 | 0.30 | 0.22 | 0.10 | <0.04 | 0.34 | 0.09 | 0.10 | <0.04 | 0.36 | <0.04 | | | |
| Na$_2$O | 0.44 | 0.22 | 0.08 | 0.94 | 0.62 | 0.33 | <0.03 | 1.25 | 0.69 | 0.29 | 0.35 | 1.29 | 0.18 | 0.19 | <0.03 | 0.46 |
| K$_2$O | 0.09 | 0.04 | 0.02 | 0.18 | 0.44 | 0.33 | 0.06 | 1.35 | 2.12 | 0.62 | 1.23 | 3.58 | <0.01 | 0.01 | <0.01 | 0.04 |
| P$_2$O5 | 1.16 | 0.46 | 0.38 | 2.28 | 2.17 | 0.60 | 0.50 | 3.33 | 6.54 | 1.73 | 4.26 | 9.46 | 0.56 | 0.09 | <0.02 | 0.62 |
| SO$_3$ | 0.05 | 0.05 | <0.03 | 0.17 | <0.03 | 0.06 | <0.03 | 0.39 | <0.03 | | | | 0.03 | 0.02 | <0.03 | 0.06 |
| V$_2$O$_5$ | <0.03 | 0.03 | <0.03 | 0.09 | <0.03 | | | | <0.03 | 0.06 | <0.03 | 0.32 | <0.03 | | | |
| BaO | <0.05 | 0.07 | <0.05 | 0.26 | <0.05 | | | | <0.05 | | | | <0.05 | 0.03 | <0.05 | 0.10 |
| Total | 99.74 | | | | 99.80 | | | | 99.53 | | | | 99.52 | | | |
| | 4 oxygen atoms, apfu | | | | | | | | | | | | 16 oxygen atoms, apfu | | | |
| Si | 0.96 | | | | 0.94 | | | | 0.85 | | | | 3.82 | | | |
| Al | 0.01 | | | | 0.01 | | | | 0.00 | | | | 0.09 | | | |
| Fe | 0.00 | | | | 0.00 | | | | 0.00 | | | | 0.03 | | | |
| Mg | 0.00 | | | | 0.00 | | | | 0.00 | | | | 0.94 | | | |
| Ca | 1.97 | | | | 1.94 | | | | 1.83 | | | | 7.00 | | | |
| Na | 0.02 | | | | 0.04 | | | | 0.04 | | | | 0.04 | | | |
| K | 0.00 | | | | 0.02 | | | | 0.08 | | | | 0.00 | | | |
| P | 0.03 | | | | 0.05 | | | | 0.16 | | | | 0.06 | | | |
| S | 0.00 | | | | 0.00 | | | | 0.00 | | | | 0.02 | | | |

$n$ = number of samples, Mean = mean value, $S$ = standard deviation, Min = minimum value, Max = maximum value, apfu = atoms per formula unit.

Strontium, a principal industrial stabilizer of α-, α′- and β-$Ca_2SiO_4$ modifications is within ≤0.34 wt % SrO. Barium, another β-$Ca_2SiO_4$ stabilizer, is restricted to a few larnite grains (≤0.26 wt % BaO), as well as Mg (≤0.31 wt % MgO) and Fe (≤0.71 wt % FeO). V, Cr, Zn, Ni and Co were reported [17,18] to incorporate into main clinker phases (alite, belite, and brownmillerite ss). The $V_2O_5$ contents in natural larnite are commonly below the detection limit (≤0.09 wt %), while Zn, Ni and Co have never been revealed by EPMA. Note, however, that the bulk contents of all these elements in the analyzed CM rocks (Tables 7 and 8) are much lower than in experimental systems. Chromium is quite often found in larnite (average 0.04 ± 0.05 wt % in that from mayenite rocks; up to 0.15 wt % $Cr_2O_3$) but is more common to the α′-$Ca_2SiO_4$ modification (Table 10).

**Flamite**, a natural analogue of the α′-$Ca_2SiO_4$ phase [52,59,88], occurs as a symmetry-related domain structure of α′-$Ca_2SiO_4$ (ss) in the β-$Ca_2SiO_4$ (ss) matrix (Figure 7A–C and Figure 9B–D). The striations intersect at an angle of 60° or 120° in cross sections perpendicular to the $c_α$-axis of the host. In other sections, lamelli cross at ~27°, 54° or 81°, which is consistent with data reported for synthetic analogs [89]. The orientation relationships between the host β-$Ca_2SiO_4$ and the α′-$Ca_2SiO_4$ lamelli are $\{11\bar{2}0\}_α\|\{100\}_β$ and $<0001>_α\|<010>_β$ which corresponds to the relative matrix lamellar orientations found in synthetic $Ca_2SiO_4$ [90]. Most β-lamelli show {100} twinning. Such a structure forms by cooling when the hexagonal α-$Ca_2SiO_4$ modification (stability range 1425–2130 °C) transforms into rhombic α′-$Ca_2SiO_4$ and preserves upon clinker quenching at >1280 °C [20,87,89].

Flamite contains more P, K and Na than the coexisting larnite and has a wide range of element compositions (Tables 9 and 10) with 56.36 to 62.97 wt % CaO, 27.34 to 32.45 wt % $SiO_2$, 4.26 to 9.46 wt % $P_2O_5$, 1.23–3.58 wt % $K_2O$, 0.35-1.29 wt % $Na_2O$, and up to 0.36 wt % SrO. Other minor elements (Al, Mg, Fe, V, Ba) are in minute amounts. Calcium and silicon contents decrease with increasing P, K and Na (Figure 10). The empirical formula is $(Ca_{1.83}Na_{0.04}K_{0.08}(Mg,Fe,Sr)_{0.01})_{Σ1.96}(Si_{0.85}P_{0.16})_{Σ2.01}O_4$. Although containing quite a high percentage of impurities (~9.5–11.5 wt % on average, mainly P, K, and Na), the natural α′-$Ca_2SiO_4$ modification is rarely stabilized in natural conditions and remains within 3 wt % according to the Rietveld analysis (Table 2). Note for comparison that clinker belite (mainly a mixture of the α- and α′-forms) contains 4–6 wt % of impurities in total, mainly Al, Fe and Na [20,87].

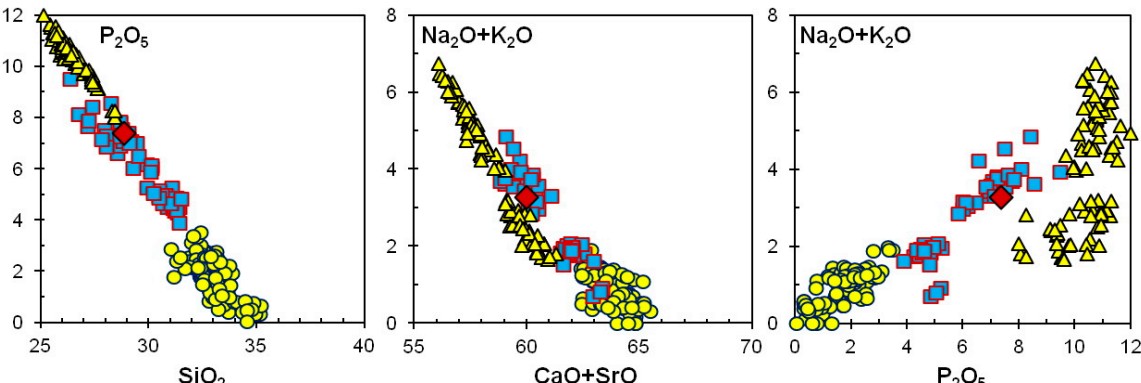

**Figure 10.** Compositional variations (wt %) of $CaSiO_4$ polymorphs from the Hatrurim larnite CM rocks. Symbols stand for: circles = larnite; squares = α′-$Ca_2SiO_4$ polymorphs; triangles = α-$Ca_2SiO_4$ $((Na,K)Ca_7(SiO_4)_3(PO_4))$, from garnet-rich paralava, Gurim, Hatrurim Basin; rhombs = average flamite composition (holotype, $(Na,K)Ca_9(SiO_4)_4(PO_4)$, α′$_L$-$Ca_2SiO_4$) from larnite–gehlenite–rankinite paralava, Hatrurim Basin [52].

## 4.2.1. Phase Relations and Transitions in the $Ca_2SiO_4$ System

Polymorphism in the $Ca_2SiO_4$ system has been comprehensively investigated because its phases are largely responsible for hydraulic activity of cement clinker, while α-, α′$_H$-, α′$_L$-, β-, and γ-$Ca_2SiO_4$ modifications are drastically different in hydration reactivity. Furthermore, γ-$Ca_2SiO_4$ which is unreactive with water is much less dense than highly reactive β-$Ca_2SiO_4$; hence, the β-to-γ transition

is highly undesirable for clinker production as it leads to β-$Ca_2SiO_4$ disintegration into powder (a phenomenon known as dusting) [20,87,89–91]. The α-to-$α'_H$ polymorphic transformation first occurs upon cooling without chemical composition changes of the phases. The transition is the first-order transformation realized as nucleation and growth processes accompanied by the formation of a specific intracrystalline lamellar structure. The $α'_H$-phase occurs as six sets of crossing lamelli related by twinning within the parent α-$Ca_2SiO_4$ (ss) crystal. In the high-temperature region, the resultant microtextures show lamellar intergrowth consisting of both α- and $α'_H$-$Ca_2SiO_4$ modifications of the same composition. The process strongly depends on the kinetics of cooling: quenching of the α-$Ca_2SiO_4$ phase is possible only at a cooling rate above 30 °C/s, and α-$Ca_2SiO_4$ entirely inverts to the $α'_H$-phase at cooling slower than 10 °C/s. The time and the temperature for the start and finish of the α-to-$α'_H$ transition have been reported as a TTT (time-temperature-transformation) diagram with two C-shaped curves [89]. Both phases have orthorhombic symmetry (instead of hexagonal α-$Ca_2SiO_4$ and monoclinic β-$Ca_2SiO_4$), because the $α'_H$-to-$α'_L$- transition does not affect the relevant microtextures, and are thus hardly identifiable even by routine XRD powder diffraction; thus, they are simply referred to as $α'$-$Ca_2SiO_4$ [89].

As was previously found for cement and ceramic systems [20,87], the high-temperature $Ca_2SiO_4$ modifications are usually stabilized by dopants Ba, Sr, P (±Na, K), Fe, Al, and B, which effectively depress the transition rate of α-$Ca_2SiO_4$ (ss) → $α'_H$ → $α'_L$; the transformation temperature of $α'_L$ → β also prevents clinker dusting caused by the β-$Ca_2SiO_4$ → γ-$Ca_2SiO_4$ transition. The $α'_H$ phase (both pure and doped) rapidly inverts upon cooling to the β-$Ca_2SiO_4$ phase passing through the $α'_L$-phase. The stabilizing effect of P, K and Na dopants and the upper temperature limit of $α'_H$–phase quenching (~1280 °C) were revealed for industrial belite-rich cement clinkers [89]. The $α'_L$↔β inversion is athermal martensitic transformation leading to polysynthetic twinning at the submicroscopic level within each lamella [89,90]. Figures 7A and 9B illustrate the complex structure resulting from the combination of two twinning systems that formed during cooling of natural rocks as a consequence of advanced α-to-$α'_H$ transition and subsequent $α'_L$-to-β inversion. The discovery of such textures in natural samples provides unambiguous evidence that the primary phase $Ca_2SiO_4$ (ss) was growing in the stability field of the α-modification at $T$ >1000 °C [4,32,52,88]. However, precise temperature estimation based solely on the textural features of $Ca_2SiO_4$ (ss) is hardly possible without knowledge of the solid solution compositions (for some reasons discussed below).

The cooling rate greatly influences the degree of phase transformations in the $Ca_2SiO_4$ system, which, together with lowering of the transition temperatures, may lead to quenching of metastable phases [20,87,89]. The $α'_H$ phase existing in equilibrium with α-$Ca_2SiO_4$ (ss) at 1280 °C forms at the maximum rate around 1100 °C regardless of the chemical composition [89]. Episodically, the cooling rate in natural CM processes at $T$ >1200 °C becomes rapid enough to ensure quenching of flamite [4,32,52,88], but usually the $α'$-$Ca_2SiO_4$ phase inverts completely to larnite (β-$Ca_2SiO_4$) (Table 2).

The high-temperature $Ca_2SiO_4$ (ss) modifications can tolerate much larger temperatures than pure $Ca_2SiO_4$. Different dopants cause different effects: e.g., the α-$Ca_2SiO_4$ (ss) → $α'_H$ phase transition temperature decreases from 1425 °C for pure $Ca_2SiO_4$ to 1395 °C for $Ca_2SiO_4$ (ss) doped with $Al_2O_3$ [89] and to 1290–1280 °C in the case of $Fe_2O_3$ doping [92,93]. Phosphorus is the most important dopant for natural $Ca_2SiO_4$ (ss) (Table 9): there is a large region of solid solutions in the $Ca_2SiO_4$–$Ca_3(PO_4)_2$ system [52,89,94], where quenching is possible for P-doped $α'$-$Ca_2SiO_4$ (ss) and even α-$Ca_2SiO_4$ (ss).

A thorough study of intricate structural characteristics and changes in P-bearing $Ca_2SiO_4$ within the composition range of $(Ca_{2-x/2}\square_{x/2})(Si_{1-x}P_x)O_4$ ($x$ = 0.03–0.40), with samples heated in the stable region of α-$Ca_2SiO_4$ and then quenched [95], showed that β-$Ca_2SiO_4$ and $α'$-$Ca_2SiO_4$ (ss) modifications can coexist in the $x$ ≤ 0.10 region (Figure 11). Within the 0.125 ≤ $x$ ≤ 0.150 range, the α-to-$α'_H$ transition is incomplete, and further cooling of $Ca_2SiO_4$ (ss) leads to $α'_H$–to-$α'_L$-phase inversion, while the residual α-phase inverts to an incommensurate phase, which is structurally similar to $α'$-$Ca_2SiO_4$ (ss) but is not its full analog. The quenching crystals with 0.175 ≤ $x$ ≤ 0.225 are composed

exclusively of the incommensurate phase, and crystals with $0.275 \leq x \leq 0.300$ are isostructural with $\alpha\text{-Ca}_2\text{SiO}_4$. The hexagonal phase with $0.350 \leq x \leq 0.400$ may be a modulated structure after $\alpha\text{-Ca}_2\text{SiO}_4$ (with $N = 2$ along the $a$-axis and $N = 3$ along $c$-axis). In all cases, the $\alpha\text{-Ca}_2\text{SiO}_4$ (ss)-to-$\alpha'\text{-Ca}_2\text{SiO}_4$ (ss) transition leads to the formation of a typical system of intersecting polysynthetic twins, which allows the high-temperature precursor phase to be easily detected in petrographic thin plates after etching (Figure 7A–C and Figure 9B–D). However, precise identification of the highly symmetrical structures $\alpha\text{-Ca}_2\text{SiO}_4$ (ss) and $\alpha'\text{-Ca}_2\text{SiO}_4$ (ss) and their unambiguous discrimination from similar incommensurate phases and modulated structures are nontrivial procedures, unfeasible even with X-ray powder detection and thus requiring single crystal XRD data [89].

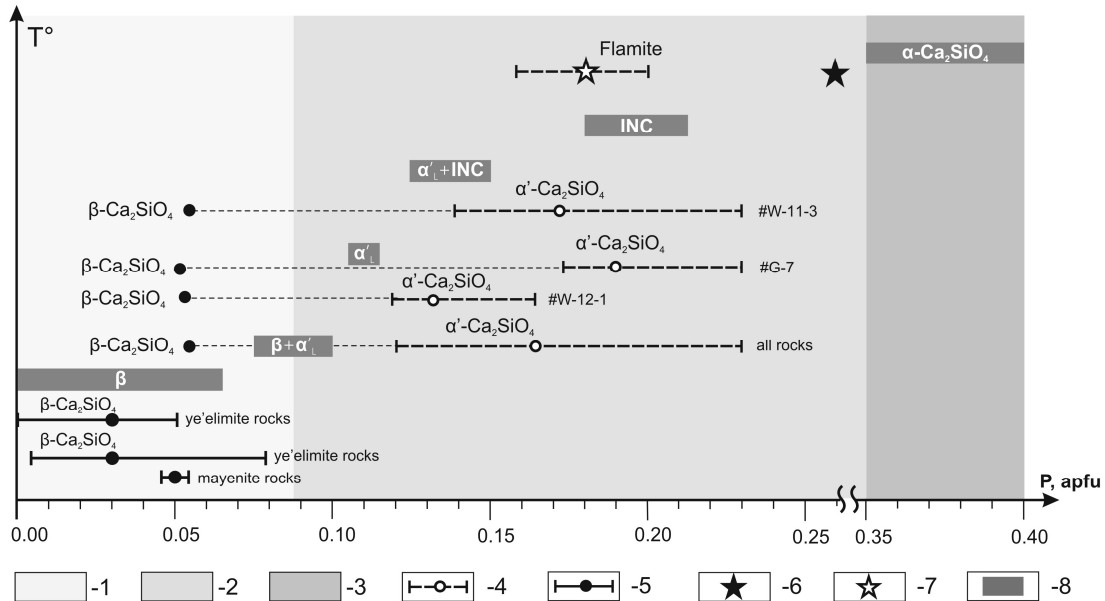

**Figure 11.** Compositional ranges of pure and P-doped synthetic $\beta\text{-Ca}_2\text{SiO}_4$, $\alpha'\text{-Ca}_2\text{SiO}_4$ and $\alpha\text{-Ca}_2\text{SiO}_4$ modifications and incommensurate phase (INC) [91] and their natural analogs from the Hatrurim larnite CM rocks [4,52,88], compared. 1 = field of complete $\alpha'\text{-Ca}_2\text{SiO}_4$-to-$\beta\text{-Ca}_2\text{SiO}_4$ transition; 2 = field of quenched of $\alpha'\text{-Ca}_2\text{SiO}_4$ modifications and incommensurate phase (INC); 3 = field of quenched $\alpha\text{-Ca}_2\text{SiO}_4$ modification; 4 = range (dash line) and average composition (open circle) of natural flamite ($\alpha'\text{-Ca}_2\text{SiO}_4$ modification), this study; 5 = range (solid line) and average composition (black circle) of natural larnite ($\beta\text{-Ca}_2\text{SiO}_4$ modification), this study; 6, 7 = flamite compositions after [88] (6) and [52] (7). 8 = ranges of pure and P-doped synthetic $\beta\text{-Ca}_2\text{SiO}_4$, $\alpha'\text{-Ca}_2\text{SiO}_4$ and $\alpha\text{-Ca}_2\text{SiO}_4$ modifications and incommensurate phase (INC), after [91].

Belites doped with $P_2O_5$ develop an orthorhombic incommensurate superstructure [89,90], which appears when the precursor $\alpha\text{-Ca}_2\text{SiO}_4$ (ss) fails to undergo the $\alpha\text{-Ca}_2\text{SiO}_4$ (ss)-to-$\alpha'\text{-Ca}_2\text{SiO}_4$ (ss) transition upon quenching. In this case, the grains of $\alpha\text{-Ca}_2\text{SiO}_4$ (ss) also consist of symmetrically related domains rotated 120° around the $c$-axis of the precursor. However, at a submicroscopic level, these domains have irregular boundaries and show mottled extinction under crossed polars [89]. The electron diffraction patterns reveal that the modulation is one-dimensional, and the modulation wavelength N correlates well with the content of the dopant expressed as P/(Si + P); N is a derivative of the unit vectors $a^*$ and $c^*$ expressed as $(1/N)a^* + c^*)$ which depends on the $Ca_2SiO_4$ (ss) composition being, for instance, N = 4 at P/(Si + P) = 0.148 [89]. These conclusions by [89] Fukuda (2001) prompt that true $\alpha'\text{-Ca}_2\text{SiO}_4$ (ss) in natural clinker-like systems should coexist with orthorhombic incommensurate superstructures, which form under control of both dopant content and cooling regime. It would be no surprise to discover $\alpha'\text{-Ca}_2\text{SiO}_4$ (ss) and orthorhombic incommensurate superstructures in neighbor areas, given that the heating and cooling regimes of the Hatrurim Fm CM rocks are highly variable (and thus high-gradient) and the occurrence of ultrahigh-temperature rocks (including paralavas) within the MZ

complexes is very local. The different regimes of cooling and quenching of $Ca_2SiO_4$-bearing rocks may be responsible for different symmetries of the $\alpha'$-$Ca_2SiO_4$ (ss) natural phases revealed previously [4,88]. Note that the samples studied in both cited works were oriented lamellar intergrowths of flamite and $\beta$-$Ca_2SiO_4$ with overlapping reflections in X-ray diffraction patterns. A very recent analysis of a P-, K-, and Na-rich single crystal of $Ca_2SiO_4$ (ss) from ultrahigh-temperature garnet paralavas has proven for the first time that they belong to orthorhombic incommensurate superstructures [59]. The authors concluded that flamite corresponds to natural P- and Na-stabilized orthorhombic $\alpha'_H$ -$Ca_2SiO_4$ with a commensurately modulated $4b$ parameter. Figure 11 perfectly illustrates the relationship between the structural types and the composition ranges of P-doped synthetic $Ca_2SiO_4$ (ss) and natural minerals from the larnite family.

4.2.2. Hydration Behavior of Natural P-, Na-, and K-Doped $\alpha'$-$Ca_2SiO_4$ and $\beta$-$Ca_2SiO_4$ Phases

Experiments on hydration of mayenite-bearing CM rocks (sample G7) under conditions simulating the industrial processes show that million-years-old natural analogs of Ca sulfoaluminate clinker have preserved high reactivity and followed a sequence of hydration similar to that in industrial Ca sulfoaluminate cement [96]. Natural mayenite can hydrate quicker (~4 h) than gehlenite (4–14 h) and P-, Na- and K-bearing larnite (24–40 h). Most of hydrates ($Al(OH)_3$ and some CASHs phases) are amorphous and the product of gehlenite hydration (strätlingite) forms alone as a phase of low-to-medium crystallinity, while high P contents (up to 11.5 wt % $P_2O_5$) poison the reactivity of $\alpha'$-$Ca_2SiO_4$ (ss). Thus, flamite behaves as an inert phase on the Earth's surface showing no evidence of hydration, unlike larnite which partly converts to calcium silicate hydrates (Figure 9B,C).

We tried to imitate hydration of $Ca_2SiO_4$-bearing rocks by water vapor upon reaching the dew point in simplified experiments. Larnite lamelli in polished plates of ye'elimite CM rocks immersed in water for 1 min developed etch twin sutures and acquired distinct grain surface microtopography, while all other minerals, including $\alpha'$-$Ca_2SiO_4$ (ss), remained intact. Therefore, natural P-, Na-, and K-doped larnite shows much higher hydration reactivity during the initial stage of 'cement' hydration than the coexisting $\alpha'$-$Ca_2SiO_4$ (ss) phase, i.e., the old natural rocks have the same relative reactivities as in the experimental systems [89]. Fragments of larnite rocks placed in a closed desiccator together with a vessel full of water (at the dew point conditions), exhibited signatures of dissolution in the upper 0.5 mm at the sites of moisture condensation after three months. The dissolution process was accompanied by leaching of Ca and its diffusion toward the surface in the form of $Ca(OH)_2$ leading to the formation of cavernous pseudomorphs consisting of X-ray amorphous (gel-like) matter with ~60 wt % $SiO_2$ and ~25 wt % CaO in the place of larnite grains. The samples became covered by a crust of microgranular calcite produced by carbonation of portlandite by atmospheric $CO_2$ in three to seven months (Figure 12), and was fully coated with a dense calcite crust in a year.

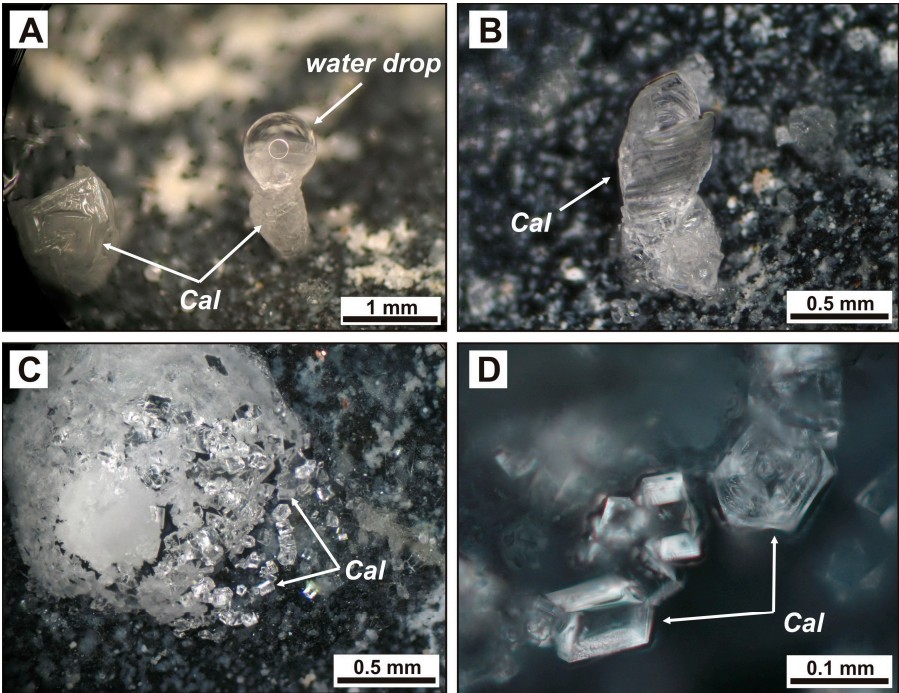

**Figure 12.** Products of experimental hydration of larnite CM rocks placed in a closed desiccator together with a water-filled vessel. A, B: calcite tube with a drop of water condensed (**A**) on its top and evaporated (**B**). (**C,D**) crust and individual crystals of fine calcite on the surface of hydrated larnite rocks kept for three to seven months in the desiccator. Cal = calcite.

**Bredigite** $Ca_7Mg(SiO_4)_4$ is quite frequently found in high-temperature contact aureoles that formed after the precursor Mg-enriched (usually dolomite-bearing) sediments, but is a relatively rare mineral [97–101]. On the other hand, this constituent is typical of periclase refractories, high-Mg cement clinker and metallurgical slags. Since long ago, there has been terminological controversy about bredigite [20,102]: the name is applied to two different compounds of $Ca_7Mg(SiO_4)_4$ and $\acute{\alpha}$-$Ca_2(SiO_4)$. First Tilley and Vincent (1948) [97] discovered bredigite in the Scawt Hill contact aureole in Northern Ireland and identified it as $\acute{\alpha}$-$Ca_2(SiO_4)$. Later, its composition was redetermined as $Ca_7Mg(SiO_4)_4$ by EPMA, but the name "bredigite" remained in use for the $\acute{\alpha}$-$Ca_2(SiO_4)$ compound in the petrography of industrial stones [1]. Currently, it is only $Ca_7Mg(SiO_4)_4$ that is referred to as bredigite [99,101]. The structure of bredigite from the Hatrurim larnite–ye'elimite-fluorellestadite rocks has only recently been fully resolved [101].

Fine grains of bredigite occur sporadically in the Hatrurim larnite rocks [4,25,51,54]. The Mg-poor CM samples we studied contain few bredigite grains, more often in the mayenitic rocks (with the lowest average MgO content 0.75 ± 0.07 wt % and the highest CaO/MgO = 71.1 ratio) than in the ye'elimite-bearing variety (with higher MgO 1.18 ± 0.33 wt % and lower CaO/MgO = 51.3 ratio). Most of Mg in both rock types commonly resides in magnesioferrite, Fe–Mg spinel, or periclase rather than in bredigite.

Bredigite reaches a rock-forming percentage (11.5%) in a single larnite–gehlenite sample (#Y-10-5), with 1.77 wt % MgO and CaO/MgO as low as 27.0 (Tables 2 and 8), where it occurs as fresh anhedral grains (10–25 μm in size) with sporadic tiny gehlenite inclusions (Figure 7H). Bredigite has an invariable composition of $(Ca_{7.00}Na_{0.04})_{\Sigma 7.04}(Mg_{0.94}Fe_{0.03})_{\Sigma 0.97}(Si_{3.82}Al_{0.09}P_{0.06}S_{0.02})_{\Sigma 3.99}O_{16}$ (Table 10), close to the ideal $Ca_7Mg(SiO_4)_4$. It contains no manganese (which is common to synthetic bredigite) and only ≤0.10 wt % BaO, unlike that from the larnite–ye'elimite–fluorellestadite rocks with up to 0.37 wt % BaO [101]. Note that Ba-bearing bredigite (analog of a synthetic phase, so-called Ba–Ca-bredigite) was reported recently from Jordanian MZ complexes [101].

Bredigite hydrates very slowly at 20 °C and is thus considered as an undesirable constituent of cement clinker (though more preferable than MgO periclase) [103]. In this respect, the presence/absence of bredigite in rocks of complex compositions is potentially of practical value: the formation of bredigite is suppressed in Mg-poor, Ca-rich and ($SO_4$)-enriched raw mixtures, but still more undesirable periclase forms instead, along with Fe–Mg spinel. Crystallization of bredigite in Al-enriched mixtures, where gehlenite is the predominant end-member in melilite solid solutions, is mainly maintained by very low Ca/Mg ratios.

**Ye'elimite**, a main phase in belite-based calcium sulfoaluminate clinker [104,105], was characterized in detail in some previous publications [4,76]. Its high hydration reactivity is responsible for its extreme rarity in natural environments, with the exception of the Hatrurim Fm CM rocks, in which ye'elimite was considered to be endemic [4,25,29,51,76] before it was discovered in metacarbonate xenoliths of Bellerberg volcano (East Eifel, Germany) [106] which are mineralogically similar to the Mottled Zone CM rocks.

Ye'elimite is rock-forming phase in all Hatrurim larnite rocks containing more than 1.4 wt % $SO_3$%, but is scarce in the mayenitic variety with 0.11–0.73 wt % $SO_3$ (Table 2). It always contains numerous inclusions of $Ca_2SiO_4$ and sporadic opaque minerals (Figures 7I and 8A). Ye'elimite commonly remains fresh but has microscopic vugs filled with aluminum hydroxide in the place of some grains. The composition of natural ye'elimite is close to $Ca_4Al_6SO_{16}$, with 46.16–49.12 wt % $Al_2O_3$, 34.98–38.49 wt % CaO, 10.93–13.69 wt % $SO_3$ (Tables 11 and 12).

**Table 11.** Mineral chemistry (WDS-EDS, wt %) of ye'elimite from Hatrurim larnite CM rocks.

| Sample | Y-5-1 | Y-6-3 | Y-8 | M5-30 | M5-31 | M4-217 | H-201 |
|---|---|---|---|---|---|---|---|
| *n* | 10 | 7 | 6 | 5 | 14 | 7 | 2 |
| $P_2O_5$ | n.a. | 0.26 | 0.32 | <0.02 | 0.21 | 0.21 | n.a. |
| $SiO_2$ | 0.83 | 1.91 | 1.45 | 0.21 | 0.94 | 0.64 | 0.68 |
| $Al_2O_3$ | 48.40 | 46.88 | 47.79 | 49.10 | 47.18 | 48.38 | 47.57 |
| $Fe_2O_3$ | 0.87 | 1.10 | 1.10 | 0.80 | 2.33 | 0.93 | 2.24 |
| MgO | n.a. | 0.17 | <0.02 | <0.02 | 0.07 | 0.04 | 0.12 |
| CaO | 36.03 | 35.27 | 36.20 | 36.15 | 35.77 | 35.83 | 36.16 |
| BaO | 0.53 | 0.91 | n.a. | n.a. | n.a. | n.a. | 0.14 |
| SrO | 0.26 | 0.35 | 0.31 | 0.52 | 0.33 | 0.65 | n.a. |
| $Na_2O$ | 0.16 | 0.05 | 0.08 | 0.03 | 0.23 | 0.09 | n.a. |
| $K_2O$ | n.a. | 0.13 | 0.17 | 0.02 | 0.08 | 0.05 | n.a. |
| $SO_3$ | 13.03 | 12.91 | 13.00 | 13.06 | 12.98 | 13.01 | 12.98 |
| Total | 100.12 | 99.95 | 100.41 | 99.89 | 100.11 | 99.82 | 99.87 |
| Formula based on 11 cations $M_4[T_6O_{12}][XO_4]$ | | | | | | | |
| P | | 0.023 | 0.028 | 0.000 | 0.018 | 0.018 | |
| Si | 0.085 | 0.197 | 0.148 | 0.021 | 0.097 | 0.066 | 0.069 |
| Al | 5.833 | 5.693 | 5.740 | 5.916 | 5.698 | 5.842 | 5.756 |
| Fe | 0.067 | 0.085 | 0.084 | 0.062 | 0.179 | 0.072 | 0.173 |
| $\Sigma[T]$ | 5.985 | 5.998 | 6.000 | 5.999 | 5.992 | 5.998 | 5.998 |
| Mg | | 0.026 | 0.000 | 0.000 | 0.010 | 0.006 | 0.018 |
| Ca | 3.948 | 3.894 | 3.952 | 3.960 | 3.927 | 3.934 | 3.977 |
| Ba | 0.021 | 0.037 | | | | | 0.006 |
| Sr | 0.016 | 0.021 | 0.018 | 0.031 | 0.020 | 0.039 | |
| Na | 0.030 | 0.009 | 0.014 | 0.005 | 0.042 | 0.016 | |
| K | | 0.017 | 0.022 | 0.003 | 0.010 | 0.007 | |
| $\Sigma[M]$ | 4.014 | 4.004 | 4.006 | 3.999 | 4.010 | 4.002 | 4.002 |
| S | 1.000 | 0.998 | 0.994 | 1.002 | 0.998 | 1.001 | 1.000 |

$TiO_2$, $Cr_2O_3$ and MnO are below the detection limits (<0.05 wt %). *n* = number of analyses; n.a. = not analyzed. $\Sigma[T]$ = P+Si+Al+Fe; $\Sigma[M]$ = Mg + Ca + Ba + Sr + Na + K. For more analyses, see [4,32].

**Table 12.** Average (*n* = 60) mineral chemistry (WDS-EDS, in wt %) of ye'elimite from Hatrurim larnite CM rocks.

| Component | Mean | *S* | Min | Max |
|---|---|---|---|---|
| $P_2O_5$ | 0.19 | 0.13 | <0.02 | 0.47 |
| $SiO_2$ | 0.89 | 0.71 | 0.12 | 3.90 |
| $Al_2O_3$ | 47.61 | 1.41 | 46.16 | 49.12 |
| $Fe_2O_3$ | 2.03 | 1.04 | 0.77 | 6.60 |
| CaO | 36.01 | 0.64 | 34.98 | 38.49 |
| BaO | 0.29 | 0.04 | 0.14 | 0.91 |
| SrO | 0.40 | 0.22 | <0.04 | 0.70 |
| $Na_2O$ | 0.14 | 0.09 | <0.03 | 0.32 |
| $K_2O$ | 0.07 | 0.03 | <0.01 | 0.17 |
| $SO_3$ | 12.80 | 0.67 | 10.93 | 13.69 |
| Total | 100.43 | | | |

Mean = mean value, *S* = standard deviation, Min = minimum value, Max = maximum value.

The key impurities are $Fe_2O_3$ (0.77–6.60 wt %, 2.03 wt % on average), $SiO_2$ (0.12–3.90 wt %, an average of 0.89 wt %), BaO (0.14–0.91 wt %, an average of 0.29 wt %), and SrO (within 0.70 wt %, 0.40 wt % on average); $K_2O$ (0.17 wt %), $P_2O_5$ (0.47 wt %), and $Na_2O$ (0.32 wt %) are rare components (Table 12); Ti, Mg, and V are within the detection limits. By analogy with the synthetic phase $Ca_4Al_{(6-2x)}Fe^{3+}_{2x}SO_{16}$ [105], natural ye'elimite can be classified as Ca sulfoaluminate doped with $Fe^{3+}$ ($x_{max}$ ~0.3).

Single-crystal analysis of natural ye'elimite reveals a cubic structure, $a$ = 9.1886(4) Å, $V_{uc}$ = 775.80(6) Å$^3$, space group $I\bar{4}3m$, which is consistent with published data on structural analysis of synthetic $Ca_4Al_6SO_{16}$ [107]. Based on the results of an X-ray powder diffraction study, Cuesta et al. [108] suggested that cubic ye'elimite would be stable at high temperature only and the symmetry of the $Ca_4Al_6SO_{16}$ phase reduced to *Pcc*2 during cooling. The results of the X-ray single crystal study of natural ye'elimite, however, do not confirm this inference.

Supergene alteration of larnite CM rocks produces micro-vugs partly filled with aluminum hydroxide in the place of ye'elimite grains. $Fe^{3+}$-bearing ye'elimite ($Ca_4Al_{5.7-5.9}Fe_{0.1-0.3}SO_{16}$) remains fresh after short immersion in water which would be enough for etch signatures in larnite grains. Prolonged natural hydration processes first transform ye'elimite into short-living phases (afwillite + ettringite ± portlandite ± tobermorite) stable at high pH > 11.5 of pore fluid, which then evolve into assemblages that are stable at lower pH ($CaCO_3$, gypsum, and $Al(OH)_3$).

**Mayenite–supergroup minerals** are major constituents in some larnite rocks of the Hatrurim Basin, where three minerals of this supergroup have been found so far [109]: fluormayenite, $Ca_{12}Al_{14}O_{32}(\square_4F_2)$, fluorkyuygenite $Ca_{12}Al_{14}O_{32}((H_2O)_4F_2)$ and chlorkyuygenite $Ca_{12}Al_{14}O_{32}((H_2O)_4Cl_2)$. Their compositions and properties were characterized in previous publications [25,26,32,109–111], and new data are summarized in Tables 13 and 14. The Hatrurim Basin is a typical locality for **fluorkyuygenite** [26,110], which is widespread in larnite rocks, whereas fluormayenite and **chlorkyuygenite** are of rare occurrence (Figure 7C,E,F, Figure 8D,E,I and Figure 9A,D–F). Some samples contain abundant **fluormayenite** grains with fluorkyuygenite rims, possibly produced by partial hydration of fluormayenite during the cooling of CM rocks and/or after the CM event, whereby vacant "zeolitic cavities" became filled by $H_2O$: $((\square_4F_2) \rightarrow ((H_2O)_4F_2))$. The transformation, however, did not dramatically change the mayenite structure [110]. Small percentages of chlorkyuygenite occur, together with CSHs, in the intergranular space and may be a retrograde phase (Figure 9A,E). Fluormayenite and fluorkyuygenite commonly contain minor amounts of $Fe_2O_3$ (1.53–4.41 wt %), $SiO_2$ (≤1.58 wt %), Cl (≤1 wt %), SrO, $SO_3$ and $P_2O_5$ (within 0.2–0.4 wt %); chlorkyuygenite contains 5.6 wt % $Fe_2O_3$ and 0.5 wt % $SiO_2$. Mayenite-supergroup minerals show high hydration activity and are easily replaced by katoite (from $Ca_3Al_2(SiO_4)_{1.5}(OH)_6$ to $Ca_3Al_2(OH)_{12}$) under supergene conditions [26].

**Table 13.** Mineral chemistry (WDS-EDS, wt %) of mayenite-supergroup minerals from Hatrurim CM larnite rocks.

| Sample | M5-30 | W-10-2 | M4-218 | M4-251 | M5-32 | M4-215 | M5-30 | W-10-2 | Y-2-1 | W-12-1 | G-7 | Y-9-1 |
|---|---|---|---|---|---|---|---|---|---|---|---|---|
| **Mineral** | Fmay | Fmay | Fky | Fky | Fky | Fky | Fky | Fky | Fky | Fky | Fky | Clky |
| *n* | 1 | 13 | 16 | 10 | 22 | 11 | 4 | 9 | 12 | 5 | 15 | 22 |
| $P_2O_5$ | 0.16 | 0.05 | 0.03 | 0.01 | 0.01 | 0.04 | 0.06 | 0.03 | 0.04 | n.a. | n.a. | 0.04 |
| $SO_3$ | 0.15 | 0.04 | <0.02 | 0.19 | 0.12 | 0.03 | 0.18 | 0.03 | 0.03 | n.a. | <0.02 | 0.04 |
| $SiO_2$ | 0.05 | 0.31 | 0.54 | 0.14 | 0.25 | 1.31 | 0.18 | 0.40 | 0.49 | 0.20 | 0.79 | 0.53 |
| $Al_2O_3$ | 49.04 | 47.79 | 45.50 | 46.72 | 46.38 | 46.27 | 47.23 | 46.09 | 43.98 | 45.76 | 45.55 | 42.53 |
| $Fe_2O_3$ | 1.78 | 2.75 | 2.77 | 2.01 | 2.21 | 1.72 | 1.56 | 2.58 | 3.63 | 2.87 | 3.17 | 5.63 |
| CaO | 47.35 | 46.62 | 44.58 | 44.77 | 44.48 | 44.90 | 45.03 | 44.90 | 44.08 | 45.31 | 44.77 | 43.59 |
| SrO | 0.28 | 0.15 | 0.11 | 0.10 | 0.12 | <0.02 | 0.14 | 0.11 | 0.06 | n.a. | <0.02 | 0.12 |
| $Na_2O$ | 0.07 | 0.07 | 0.04 | 0.12 | 0.24 | 0.05 | 0.10 | 0.07 | <0.02 | n.a. | <0.02 | 0.06 |
| $K_2O$ | 0.01 | 0.02 | 0.01 | 0.12 | 0.14 | 0.02 | 0.04 | 0.02 | 0.01 | n.a. | 0.20 | 0.02 |
| F | 1.90 | 1.56 | 2.73 | 2.13 | 1.99 | 2.72 | 1.70 | 1.58 | 1.52 | 1.84 | 1.93 | 0.25 |
| Cl | 0.23 | 0.69 | 0.08 | 0.39 | 0.30 | 0.04 | 0.18 | 0.67 | 0.99 | 0.29 | n.a. | 5.24 |
| $H_2O$ | 0.38 | 0.40 | 4.59 | 4.91 | 5.03 | 4.52 | 5.15 | 5.06 | 4.86 | 5.09 | 5.02 | 3.97 |
| Total | 101.40 | 100.45 | 100.98 | 101.60 | 101.26 | 101.63 | 101.54 | 101.53 | 99.71 | 101.36 | 101.42 | 102.02 |
| $O-(F,Cl)_2$ | 0.85 | 0.81 | 1.17 | 0.98 | 0.91 | 1.15 | 0.76 | 0.82 | 0.86 | 0.84 | 0.81 | 1.29 |
| Total | 100.55 | 99.63 | 99.82 | 100.62 | 100.36 | 100.48 | 100.78 | 100.72 | 98.84 | 100.53 | 100.61 | 100.73 |
| Formula based on 26 cations $X_{12}T_{14}O_{32-x}(OH)_{3x}[W_{6-3x}]$ | | | | | | | | | | | | |
| P | 0.032 | 0.010 | 0.006 | 0.002 | 0.003 | 0.008 | 0.012 | 0.006 | 0.008 | | | 0.009 |
| S | 0.027 | 0.007 | 0.000 | 0.035 | 0.022 | 0.005 | 0.033 | 0.006 | 0.005 | | 0.000 | 0.008 |
| Si | 0.012 | 0.074 | 0.135 | 0.034 | 0.062 | 0.324 | 0.044 | 0.098 | 0.126 | 0.050 | 0.194 | 0.135 |
| Al | 13.601 | 13.438 | 13.380 | 13.598 | 13.531 | 13.449 | 13.682 | 13.446 | 13.156 | 13.375 | 13.283 | 12.793 |
| Fe | 0.315 | 0.494 | 0.520 | 0.374 | 0.412 | 0.320 | 0.289 | 0.480 | 0.693 | 0.536 | 0.589 | 1.081 |
| Σ[T] | 13.986 | 14.022 | 14.041 | 14.042 | 14.030 | 14.105 | 14.059 | 14.036 | 13.987 | 13.961 | 14.067 | 14.026 |
| Ca | 11.938 | 11.917 | 11.917 | 11.846 | 11.797 | 11.865 | 11.859 | 11.910 | 11.988 | 12.039 | 11.870 | 11.920 |
| Sr | 0.038 | 0.020 | 0.016 | 0.014 | 0.017 | 0.000 | 0.021 | 0.015 | 0.009 | | 0.000 | 0.018 |
| Na | 0.034 | 0.035 | 0.021 | 0.058 | 0.113 | 0.022 | 0.049 | 0.032 | 0.012 | | 0.000 | 0.030 |
| K | 0.004 | 0.006 | 0.004 | 0.039 | 0.043 | 0.005 | 0.012 | 0.007 | 0.003 | | 0.063 | 0.007 |
| Σ[X] | 12.014 | 11.978 | 11.958 | 11.958 | 11.970 | 11.892 | 11.941 | 11.964 | 12.013 | 12.039 | 11.933 | 11.974 |
| F | 1.414 | 1.178 | 2.152 | 1.660 | 1.560 | 2.123 | 1.320 | 1.238 | 1.222 | 1.440 | 1.508 | 0.202 |
| Cl | 0.090 | 0.280 | 0.033 | 0.163 | 0.124 | 0.016 | 0.076 | 0.282 | 0.425 | 0.123 | | 2.267 |
| OH | 0.602 | 0.635 | 0.014 | 0.266 | 0.324 | 0.293 | 0.768 | 0.605 | 0.480 | 0.447 | 0.690 | 0.303 |
| $H_2O$ | | | 3.814 | 3.911 | 3.992 | 3.568 | 3.836 | 3.875 | 3.872 | 3.990 | 3.802 | 3.229 |
| Σ[W] | 2.105 | 2.094 | 6.000 | 6.000 | 6.000 | 6.000 | 6.000 | 6.000 | 6.000 | 6.000 | 6.000 | 6.000 |

$TiO_2$ (<0.02 wt %), MnO (<0.02 wt %), MgO (<0.02 wt %) and BaO (<0.05 wt %) are below the detection limits. *n* = number of analyses; n.a. = not analyzed. Σ[T] = P+S+Si+Al+Fe; Σ[X] = Ca + Sr + Na + K; Σ[W] = F + Cl + OH + $H_2O$. Mineral names are abbreviated as Fmay = fluormayenite; Fky = fluorkyuygenite; Clky = chlorkyuygenite. Formulas are calculated according to [109,110]. For more data, see [32,109,110].

**Table 14.** Average mineral chemistry (WDS-EDS, wt %) of mayenite-supergroup minerals from Hatrurim larnite CM rocks.

| Mineral | Fluormayenite (n = 14) | | | | Fluorkyuygenite (n = 104) | | | |
|---|---|---|---|---|---|---|---|---|
| n | Mean | *S* | Min | Max | Mean | *S* | Min | Max |
| $P_2O_5$ | 0.06 | 0.06 | <0.02 | 0.16 | 0.03 | 0.03 | <0.02 | 0.12 |
| $SO_3$ | 0.05 | 0.06 | <0.03 | 0.22 | 0.06 | 0.08 | <0.03 | 0.31 |
| $SiO_2$ | 0.29 | 0.28 | 0.05 | 1.26 | 0.51 | 0.42 | 0.05 | 1.58 |
| $Al_2O_3$ | 47.88 | 0.73 | 46.35 | 49.04 | 45.84 | 1.01 | 41.95 | 47.69 |
| $Fe_2O_3$ | 2.68 | 0.32 | 1.78 | 2.98 | 2.57 | 0.66 | 1.53 | 4.41 |
| CaO | 46.67 | 0.32 | 46.29 | 47.35 | 44.66 | 0.58 | 43.36 | 47.18 |
| SrO | 0.16 | 0.09 | 0.02 | 0.29 | 0.08 | 0.09 | <0.04 | 0.36 |
| $Na_2O$ | 0.07 | 0.03 | <0.03 | 0.12 | 0.09 | 0.10 | <0.03 | 0.36 |
| $K_2O$ | 0.02 | 0.02 | <0.01 | 0.05 | 0.07 | 0.07 | <0.01 | 0.20 |
| F | 1.59 | 0.18 | 1.37 | 1.90 | 2.07 | 0.49 | 1.13 | 3.11 |
| Cl | 0.66 | 0.14 | 0.23 | 0.78 | 0.37 | 0.33 | 0.01 | 1.03 |
| $H_2O$ | 0.40 | 0.11 | 0.21 | 0.57 | 4.95 | 1.12 | 4.28 | 8.76 |
| Total | 100.52 | | | | 101.30 | | | |
| $O-(F,Cl)_2$ | 0.82 | 0.07 | 0.73 | 0.96 | 0.84 | 0.33 | 0.00 | 1.33 |
| Total | 99.70 | | | | 100.46 | | | |

*n* = number of samples, Mean = mean value, *S* = standard deviation, Min = minimum value, Max = maximum value.

### 4.2.3. Fluorapatite-Fluorellestadite and Ternesite–Silicocarnotite Solid Solutions

The Hatrurim Fm $Ca_2SiO_4$-bearing rocks are remarkable due to their high diversity of Ca- and P-bearing minerals, especially fluorapatite-fluorellestadite solid solutions (ss), which are found as rock-forming minerals (15–25%) in ye'elimite–larnite samples but only as accessories (≤3%) in the $(SO_4)$-poor mayenite variety (Table 2). Fluorapatite-fluorellestadite ss exist as large poikilitic crystals (from 50 to 200–400 μm), with numerous inclusions of larnite, ye'elimite, and opaque phases (Figures 7I and 8A,B). It lacks chemical zonation, but main anion substitutions within every rock sample have rather large ranges of 10.67–28.29 wt % $P_2O_5$, 7.82–16.58 wt % $SO_3$, 5.48–14.20 wt % $SiO_2$, and 1.92–3.63 wt % F, while $V_2O_5$ contents are as low as 0.12–0.58 wt %; CaO varies from 54.99 to 56.70 wt % and SrO is from 0.17–0.83 wt % (Tables 15 and 16). The space group of fluorellestadite long remains debatable [112,113] but has been determined recently by single-crystal analysis for $Ca_{9.99}Si_{2.52}S_{2.04}P_{1.05}\,Al_{0.02}O_{24}F_{2.03}$, which gives $a = 9.42250(2)$ Å; $c = 6.93026(18)$ Å; V = 533.14(2) Å$^3$; and $P6_3/m$ (R1 = 0.0357; wR2 = 0.0712; GooF = 1.055).

**Table 15.** Mineral chemistry (WDS-EDS, wt %) of fluorapatite and fluorellestadite from Hatrurim larnite CM rocks.

| Type | Y | Y | Y | Y | M | M | M | M | M | M | Gh | Gh |
|---|---|---|---|---|---|---|---|---|---|---|---|---|
| Sample | Y-5-1 | Y-6-3 | Y-8 | M5-30 | Y-9-1 | W-10-2 | CONCR | CONCR | M4-215 | Y-2-1 | W-11-3 | Y-10-5 |
| Mineral | Ap | Ap | Els | Els | Ap | Ap | Ap | Ap | Ap | Ap | Ap | Ap |
| *n* | 9 | 10 | 7 | 2 | 4 | 24 | 4 | 1 | 1 | 7 | 3 | 2 |
| $P_2O_5$ | 19.17 | 24.27 | 13.39 | 6.15 | 14.83 | 30.98 | 35.76 | 23.72 | 40.81 | 33.58 | 37.53 | 29.31 |
| $V_2O_5$ | 0.31 | 0.47 | 0.33 | n.a. | 0.35 | 0.44 | n.a. | n.a. | n.a. | 0.35 | 0.52 | 0.50 |
| $SO_3$ | 12.60 | 9.46 | 16.01 | 21.32 | 15.60 | 5.62 | 2.71 | 17.83 | 0.54 | 4.12 | 0.82 | 6.27 |
| $SiO_2$ | 10.00 | 7.73 | 12.31 | 15.17 | 11.57 | 5.35 | 3.59 | 2.07 | 0.65 | 3.89 | 3.29 | 6.28 |
| FeO | 0.40 | 0.47 | 0.31 | n.a. | 0.36 | 0.10 | 0.04 | 0.02 | <0.02 | 0.12 | 0.19 | 0.33 |
| CaO | 55.58 | 55.49 | 55.44 | 55.63 | 55.74 | 55.50 | 55.95 | 55.37 | 56.04 | 55.59 | 55.23 | 55.91 |
| SrO | n.a. | 0.67 | 0.50 | 0.83 | n.a. | 0.46 | 0.35 | 0.27 | 0.13 | 0.19 | 0.83 | n.a. |
| $Na_2O$ | n.a. | <0.02 | 0.03 | 0.13 | n.a. | 0.02 | 0.03 | 0.02 | <0.02 | 0.02 | n.a. | n.a. |
| $K_2O$ | n.a. | n.a. | n.a. | n.a. | n.a. | 0.05 | <0.01 | 0.03 | <0.01 | <0.01 | n.a. | n.a. |
| F | 3.63 | 2.57 | 2.67 | 1.92 | 2.89 | 2.76 | 3.00 | 2.95 | 3.20 | 3.05 | 2.39 | 3.16 |
| Cl | n.a. | n.a. | 0.10 | n.a. | 0.23 | 0.01 | n.a. | n.a. | n.a. | 0.01 | n.a. | n.a. |
| Total | 101.67 | 101.12 | 101.09 | 101.13 | 101.58 | 101.27 | 101.44 | 102.28 | 101.36 | 100.91 | 100.82 | 101.75 |
| O-(F,Cl)$_2$ | 1.53 | 1.08 | 1.15 | 0.81 | 1.27 | 1.16 | 1.26 | 1.24 | 1.35 | 1.29 | 1.01 | 1.33 |
| Total | 100.14 | 100.04 | 99.94 | 100.32 | 100.31 | 100.11 | 100.18 | 101.04 | 100.02 | 99.63 | 99.81 | 100.43 |
| Formula based on 10 cations in the Ca atoms $M_{10}[ZO_4]_{6\times2}$ | | | | | | | | | | | | |
| P | 2.71 | 3.41 | 1.89 | 0.86 | 2.09 | 4.37 | 5.02 | 3.37 | 5.74 | 4.75 | 5.31 | 4.12 |
| V | 0.03 | 0.05 | 0.04 | 0.00 | 0.04 | 0.05 | | | | 0.04 | 0.06 | 0.06 |
| S | 1.58 | 1.18 | 2.00 | 2.65 | 1.95 | 0.70 | 0.34 | 2.24 | 0.07 | 0.52 | 0.10 | 0.78 |
| Si | 1.67 | 1.28 | 2.05 | 2.51 | 1.93 | 0.89 | 0.60 | 0.35 | 0.11 | 0.65 | 0.55 | 1.04 |
| Σ[Z] | 5.99 | 5.92 | 5.97 | 6.02 | 6.00 | 6.02 | 5.95 | 5.96 | 5.92 | 5.95 | 6.02 | 6.00 |
| Fe | 0.06 | 0.07 | 0.04 | | 0.05 | 0.01 | 0.01 | 0.00 | 0.00 | 0.02 | 0.03 | 0.05 |
| Ca | 9.94 | 9.87 | 9.90 | 9.88 | 9.95 | 9.93 | 9.95 | 9.96 | 9.99 | 9.96 | 9.89 | 9.95 |
| Sr | | 0.06 | 0.05 | 0.08 | | 0.04 | 0.03 | 0.03 | 0.01 | 0.02 | 0.08 | |
| Na | | 0.00 | 0.01 | 0.04 | | 0.01 | 0.01 | 0.01 | 0.00 | 0.00 | | |
| K | | | | | | 0.01 | 0.00 | 0.01 | 0.00 | 0.00 | | |
| Σ[M] | 10.00 | 10.00 | 10.00 | 10.00 | 10.00 | 10.00 | 10.00 | 10.00 | 10.00 | 10.00 | 10.00 | 10.00 |
| F | 1.91 | 1.35 | 1.40 | 1.00 | 1.52 | 1.46 | 1.57 | 1.56 | 1.68 | 1.61 | 1.26 | 1.66 |
| Cl | | | 0.03 | | 0.07 | 0.00 | | | | 0.00 | | |

*n* = number of analyses; MnO and MgO are below the detection limits (<0.02 wt %); n.a. = not analyzed; Ap = fluorapatite, Els = fluorellestadite, Y = ye'elimite-bearing larnite rocks; M = mayenite-bearing larnite rocks; Gh = gehlenite-rich rocks. For more data on fluorapatite-fluorellestadite from Hatrurim rocks, see [4,26,32].

The intermediate phases of **ternesite–silicocarnotite** series, with a ternesite/silicocarnotite ratio of 90:10 to 50:50, were found as 50 μm × 10 μm prismatic grains in two ye'elimite-bearing samples only. Main impurities are V (up to 0.33 wt % $V_2O_5$) and Sr (up to 0.16 wt % SrO) (Table 17). A detailed characteristic of silicocarnotite–ternesite ss from the Hatrurim complex outcrops was presented previously [32,114].

**Table 16.** Average mineral chemistry (WDS-EDS, wt %) of fluorapatite and fluorellestadite from Hatrurim ye'elimite– and mayenite-bearing larnite CM rocks.

| Mineral | Fluorellestadite ($n$ = 25) | | | | Fluorapatite ($n$ = 60) | | | |
|---------|------|------|------|------|------|------|------|------|
| Component | Mean | $S$ | Min | Max | Mean | $S$ | Min | Max |
| $P_2O_5$ | 11.43 | 3.84 | 6.12 | 15.60 | 28.37 | 6.37 | 15.11 | 39.68 |
| $V_2O_5$ | 0.29 | 0.14 | <0.03 | 0.49 | 0.39 | 0.14 | <0.03 | 0.62 |
| $SO_3$ | 17.41 | 2.64 | 13.63 | 21.45 | 7.13 | 3.87 | 0.65 | 15.53 |
| $SiO_2$ | 13.04 | 1.47 | 11.51 | 15.21 | 6.32 | 2.57 | 1.73 | 11.94 |
| FeO | 0.32 | 0.25 | <0.02 | 0.69 | 0.20 | 0.21 | <0.02 | 0.75 |
| CaO | 55.49 | 0.20 | 55.12 | 55.76 | 55.56 | 0.26 | 55.07 | 56.27 |
| SrO | 0.58 | 0.16 | 0.37 | 0.86 | 0.47 | 0.22 | 0.03 | 0.85 |
| $Na_2O$ | 0.06 | 0.11 | <0.03 | 0.26 | <0.03 | 0.02 | <0.03 | 0.07 |
| $K_2O$ | <0.01 | | | | 0.03 | 0.05 | <0.01 | 0.21 |
| F | 2.56 | 0.40 | 1.75 | 3.12 | 2.90 | 0.47 | 2.20 | 4.02 |
| Cl | 0.14 | 0.10 | <0.01 | 0.33 | 0.02 | 0.05 | <0.01 | 0.21 |
| Total | 101.31 | | | | 101.41 | | | |
| $O-(F,Cl)_2$ | 1.10 | 0.19 | 0.74 | 1.39 | 1.22 | 0.20 | 0.93 | 1.69 |
| Total | 100.21 | | | | 100.19 | | | |

$n$ = number of samples, Mean = mean value, $S$ = standard deviation, Min = minimum value, Max = maximum value.

**Table 17.** Mineral chemistry (WDS, wt %) of intermediate phases of the silicocarnotite–ternesite series from Hatrurim larnite CM rocks, compared with holotype silicocarnotite and ternesite.

| Number | 1 | 2 | 3 | 4 | 5 | 6 | 7 | 8 |
|--------|---|---|---|---|---|---|---|---|
| Sample | M5-30 | M5-31 | YV-567 | YV-415 | Ideal | Ideal | Ideal | Ideal |
| $n$ | 10 | 9 | 16 | 6 | | | | |
| $P_2O_5$ | 2.95 | 14.33 | 27.94 | 25.81 | 29.42 | 0.00 | 14.74 | 2.95 |
| $V_2O_5$ | 0.09 | 0.33 | 0.51 | n.a. | n.a. | n.a. | n.a. | n.a. |
| $SO_3$ | 14.94 | 8.07 | 0.32 | 1.85 | 0.00 | 16.66 | 8.31 | 14.99 |
| $SiO_2$ | 23.09 | 18.21 | 12.70 | 13.62 | 12.45 | 25.00 | 18.72 | 23.74 |
| $TiO_2$ | 0.28 | 0.12 | n.a. | n.a. | n.a. | n.a. | n.a. | n.a. |
| $Cr_2O_3$ | 0.08 | <0.02 | n.a. | n.a. | n.a. | n.a. | n.a. | n.a. |
| $Al_2O_3$ | 0.07 | <0.02 | n.a. | n.a. | n.a. | n.a. | n.a. | n.a. |
| FeO | 0.07 | 0.07 | n.a. | n.a. | n.a. | n.a. | n.a. | n.a. |
| CaO | 58.24 | 57.92 | 57.24 | 57.61 | 58.13 | 58.34 | 58.23 | 58.32 |
| SrO | 0.16 | 0.06 | 0.17 | 0.13 | n.a. | n.a. | n.a. | n.a. |
| $Na_2O$ | 0.07 | 0.11 | n.a. | n.a. | n.a. | n.a. | n.a. | n.a. |
| Total | 100.04 | 99.31 | 98.88 | 99.02 | 100.00 | 100.00 | 100.00 | 100.00 |
| Formula based on 12 oxygens $M_5[TO_4]_2[TO_4]$ | | | | | | | | |
| P | 0.20 | 0.98 | 1.92 | 1.77 | 2.00 | 0.00 | 1.00 | 0.20 |
| V | 0.00 | 0.02 | 0.03 | | | | | |
| S | 0.90 | 0.49 | 0.02 | 0.11 | 0.00 | 1.00 | 0.50 | 0.90 |
| Si | 1.85 | 1.48 | 1.03 | 1.11 | 1.00 | 2.00 | 1.50 | 1.90 |
| Al | 0.01 | 0.00 | | | | | | |
| Ti | 0.02 | 0.01 | | | | | | |
| $\Sigma[T]$ | 2.99 | 2.98 | 3.00 | 2.99 | 3.00 | 3.00 | 3.00 | 3.00 |
| Cr | 0.00 | 0.00 | | | | | | |
| Fe | 0.00 | 0.00 | | | | | | |
| Ca | 5.01 | 5.03 | 4.99 | 5.01 | 5.00 | 5.00 | 5.00 | 5.00 |
| Sr | 0.01 | 0.00 | 0.01 | 0.01 | | | | |
| Na | 0.01 | 0.02 | | | | | | |

$n$ = number of analyses; n.a. = not analyzed; $K_2O$ (<0.01 wt %)MnO (<0.02 wt %), MgO (<0.02wt %); BaO (<0.05 wt %), Cl (<0.01 wt %), F (<0.06 wt %), $As_2O_5$ (<0.03 wt %) are below the detection limits. 1–2 = intermediate phases from ye'elimite-bearing CM rocks, central Hatrurim Basin (this study); 3–4 = holotype silicocarnotite from melilite rocks, central Hatrurim Basin [114]; 5 = ideal silicocarnotite $Ca_5(PO_4)_2(SiO_4)$; 6 = ideal ternesite $Ca_5(SiO_4)_2(SO_4)$; 7 = ideal $Ca_5(SiO_4)_{1.5}(PO_4)(SO_4)_{0.5}$; 8 = ideal $Ca_5(SiO_4)_{1.9}(SO_4)_{0.9}(PO_4)_{0.2}$. $\Sigma[T]$ = P+V+S+Si+Al+Ti.

The minerals of the two ss series remain intact even in strongly hydrated and carbonated varieties of CM rocks where clinker minerals have been replaced by calcite and CSHs (±aragonite, vaterite, ettringite, gypsum). Therefore, these phases should inevitably appear as ballast during industrial annealing of mixtures containing fluorine and excess phosphorus, in addition to the $P_2O_5$ amount consumed by $Ca_2SiO_4$ modifications, and will never participate in hydration reactions.

$Fe^{3+}$-rich **perovskite–supergroup minerals** are widespread main or minor phases in the Hatrurim larnite rocks (Table 2). They are members of the pseudobinary perovskite–brownmillerite series, in which six minerals have been identified to date, including four new phases: brownmillerite $Ca_2(Al,Fe)FeO_5$, srebrodolskite $Ca_2FeFeO_5$, shulamitite $Ca_3TiFeAlO_8$ and sharyginite $Ca_3TiFeFeO_8$, nataliakulikite $Ca_4Ti_2FeFeO_{11}$ and $Fe^{3+}$-rich perovskite $CaTi_{1-2x}Fe_{2x}O_{3-x}$ ($x <0.25$) [25–28,106,115]. They can be classified as layered perovskites according to the material science terminology [116–120] and belong to anion deficient perovskites (brownmillerite subgroup, non-stoichiometric perovskites group) according to the recent nomenclature of the perovskite supergroup [121]. Surprisingly, the Hatrurim Basin is now the only CM area where all members of the perovskite–brownmillerite family have been found (Figures 7 and 8; Table 18).

**Table 18.** Mineral chemistry (wt %) of perovskite–supergroup minerals (anion deficient perovskites, brownmillerite subgroup) in Hatrurim larnite CM rocks, Israel.

| Type | Y | Y | Y | M | M | M | M | Y | Y | M | M | Gh | Gh | M |
|---|---|---|---|---|---|---|---|---|---|---|---|---|---|---|
| Sample | Y-6-3 | Y-6-3 | Y-8 | W-10-2 | Y-2-1 | G-7 | W-12-1 | Y-6-3 | Y-6-3 | W-10-2 | G-7 | W-11-3 | W-11-3 | W-10-2 |
| Mineral | Srb | Brm | Srb | Brm | Brm | Brm | Brm | Shr | Shr | Shu | Shu | Ntk | Prv | Prv |
| $n$ | 18 | 4 | 8 | 22 | 7 | 11 | 13 | 5 | 2 | 1 | 7 | 49 | 6 | 7 |
| $SiO_2$ | 0.58 | 0.99 | 0.68 | 0.67 | 0.64 | 1.05 | 0.72 | 1.08 | 0.60 | 0.98 | 1.04 | 5.05 | 3.81 | 2.86 |
| $TiO_2$ | 0.31 | 0.65 | 0.30 | 2.64 | 3.07 | 2.97 | 3.41 | 19.35 | 16.59 | 20.97 | 19.47 | 29.04 | 33.36 | 39.31 |
| $ZrO_2$ | n.a. | n.a. | n.a. | n.a. | n.a. | n.a. | n.a. | n.a. | 2.52 | n.a. | 0.49 | 0.68 | 0.74 | 0.55 |
| $Nb_2O_5$ | n.a. | n.a. | n.a. | n.a. | n.a. | n.a. | n.a. | n.a. | n.a. | n.a. | n.a. | 0.04 | 0.06 | n.a. |
| $Cr_2O_3$ | 0.25 | 0.60 | <0.02 | 0.66 | 1.14 | 0.45 | 2.34 | <0.02 | n.a. | n.a. | 0.48 | 0.08 | 0.07 | 0.41 |
| $Al_2O_3$ | 4.39 | 11.04 | 6.72 | 13.48 | 10.85 | 13.78 | 15.40 | 5.93 | 5.93 | 7.88 | 9.33 | 2.07 | 1.92 | 3.13 |
| $Fe_2O_3$ | 51.61 | 41.89 | 48.51 | 36.03 | 38.46 | 34.76 | 30.88 | 30.13 | 30.69 | 25.47 | 25.46 | 14.23 | 17.66 | 11.68 |
| FeO* | 0.24 | 0.48 | 0.11 | 1.03 | 1.24 | 1.54 | 1.31 | n.a. | n.a. | 1.00 | 0.10 | 5.47 | | |
| MnO | 0.31 | 0.52 | 0.32 | n.a. | n.a. | n.a. | n.a. | 0.65 | 0.84 | n.a. | n.a. | 0.07 | 0.03 | n.a. |
| MgO | 0.26 | 0.50 | 0.61 | 0.68 | 0.66 | 0.58 | 0.84 | <0.02 | 0.45 | n.a. | 0.30 | <0.02 | <0.02 | n.a. |
| CaO | 42.33 | 43.89 | 42.63 | 44.64 | 44.19 | 44.66 | 45.27 | 42.79 | 42.63 | 42.55 | 43.33 | 42.10 | 41.98 | 42.44 |
| SrO | n.a. | n.a. | n.a. | n.a. | n.a. | n.a. | n.a. | n.a. | n.a. | n.a. | n.a. | 0.27 | 0.27 | n.a. |
| Total | 100.30 | 100.59 | 99.89 | 99.95 | 100.39 | 99.98 | 100.31 | 99.93 | 100.24 | 98.97 | 100.01 | 99.10 | 99.76 | 100.38 |
| Formula based on | 4 cations and 5 oxygens | | | | | | | 6 cations and 8 oxygens | | | | 8 cations 11 oxygens | 2 cations | |
| Si | 0.03 | 0.04 | 0.03 | 0.03 | 0.03 | 0.04 | 0.03 | 0.07 | 0.04 | 0.06 | 0.07 | 0.45 | 0.08 | 0.06 |
| Al | 0.23 | 0.55 | 0.34 | 0.66 | 0.54 | 0.68 | 0.75 | 0.46 | 0.46 | 0.61 | 0.71 | 0.22 | 0.05 | 0.08 |
| $Fe^{3+}$ | 1.70 | 1.33 | 1.58 | 1.13 | 1.22 | 1.09 | 0.95 | 1.48 | 1.51 | 1.25 | 1.23 | 0.95 | 0.30 | 0.19 |
| $Fe^{2+}$ | 0.01 | 0.02 | 0.00 | 0.04 | 0.04 | 0.05 | 0.05 | | | 0.05 | 0.01 | 0.41 | | |
| Ti | 0.01 | 0.02 | 0.01 | 0.08 | 0.10 | 0.09 | 0.11 | 0.95 | 0.82 | 1.03 | 0.94 | 1.93 | 0.56 | 0.65 |
| Zr+Nb | | | | | | | | | 0.08 | | 0.02 | 0.03 | 0.01 | 0.01 |
| Cr | 0.01 | 0.02 | 0.00 | 0.02 | 0.04 | 0.01 | 0.08 | 0.00 | 0.00 | | 0.02 | 0.01 | 0.00 | 0.01 |
| Mn | 0.01 | 0.02 | 0.01 | | | | | 0.04 | 0.05 | | | 0.01 | 0.00 | |
| Mg | 0.02 | 0.03 | 0.04 | 0.04 | 0.04 | 0.04 | 0.05 | 0.00 | 0.04 | | 0.03 | 0.00 | 0.00 | |
| Ca + Sr | 1.99 | 1.98 | 1.98 | 2.00 | 2.00 | 1.99 | 1.99 | 3.00 | 3.00 | 2.99 | 2.98 | 4.01 | 1.00 | 1.00 |

$n$ = number of analyses; $Na_2O$ (<0.02 wt %) and $V_2O_3$ (<0.03 wt %) are below the detection limits; n.a. = not analyzed. Y = ye'elimite–larnite rocks; M = mayenite–larnite rocks; Gh = gehlenite-rich rocks. Minerals are abbreviated as Srb = srebrodolskite $CaFeFeO_5$; Brm = brownmillerite $Ca(Fe,Al)AlO_5$; Shr = sharyginite $Ca_3TiFeFeO_8$; Shu = shulamitite $Ca_3Ti(Fe,Al)AlO_8$; Ntk = nataliakulikite $Ca_4Ti_2FeFeO_{11}$; Prv = $Fe^{3+}$-rich perovskite $CaTi_{1-2x}Fe_{2x}O_{3-x}$ ($x < 0.25$). FeO and $Fe_2O_3$ are calculated on charge balance.

### 4.2.4. Phase Relations in the System $Ca_2Fe_2O_5$–$CaTiO_3$

The synthetic compounds of the $CaTiO_3$–$Ca_2Fe_2O_5$ series have been exhaustively studied since the 1960–1970s for specific properties of oxygen-deficient Fe-rich perovskites, such as superconductivity, oxygen ionic conductivity, and electronic conductivity. Numerous compounds corresponding to the general formula $CaTi_{1-2x}Fe_{2x}O_{3-x}$ ($0 \leq x \leq 0.5$) were synthesized within this system [116]. However, only four orthorhombic compounds are fully ordered in oxygen vacancies and appear to be the most stable: $CaTiO_3$ ($x = 0$), $Ca_4Ti_2Fe_2O_{11}$ ($x = 0.25$), $Ca_3TiFe_2O_8$ ($x = 0.33$) and $Ca_2Fe_2O_5$ ($x = 0.5$) [116–120,122–124]. Perovskite and brownmillerite are never in equilibrium in any assemblage, as one can see in the $CaTiO_3$–$CaFeO_{2.5}$ phase diagram (Figure 13), because their stability fields are separated by that of the $Ca_3TiFe_2O_8$ + $Ca_4Ti_2Fe_2O_{11}$ assemblage. In the same way, the $Ca_4Ti_2Fe_2O_{11}$ phase is never in equilibrium with $Ca_2Fe_2O_5$, but equilibrium is possible with the $Ca_3TiFe_2O_8$ phase or with the $Ca_3TiFe_2O_8$ + $Ca_2Fe_2O_5$ assemblage. The intermediate phases $Ca_4Ti_2Fe_2O_{11}$ and $Ca_3TiFe_2O_8$ should be stoichiometric and free from both cation and oxygen vacancies [116]. Any deviation of bulk compositions from the Ti:Fe stoichiometry of both $Ca_4Ti_2Fe_2O_{11}$ and $Ca_3TiFe_2O_8$ phases leads to the appearance of nanoscale biphasic intergrowths of $Ca_4Ti_2Fe_2O_{11}$ + Fe–perovskite, $Ca_3TiFe_2O_8$ + $Ca_4Ti_2Fe_2O_{11}$ or $Ca_3TiFe_2O_8$ + $Ca_2Fe_2O_5$. In addition to these four compounds, the ordered $Ca_5TiFe_4O_{13}$ phase occurs in the $Ca_3TiFe_2O_8$–$Ca_2Fe_2O_5$ region [21].

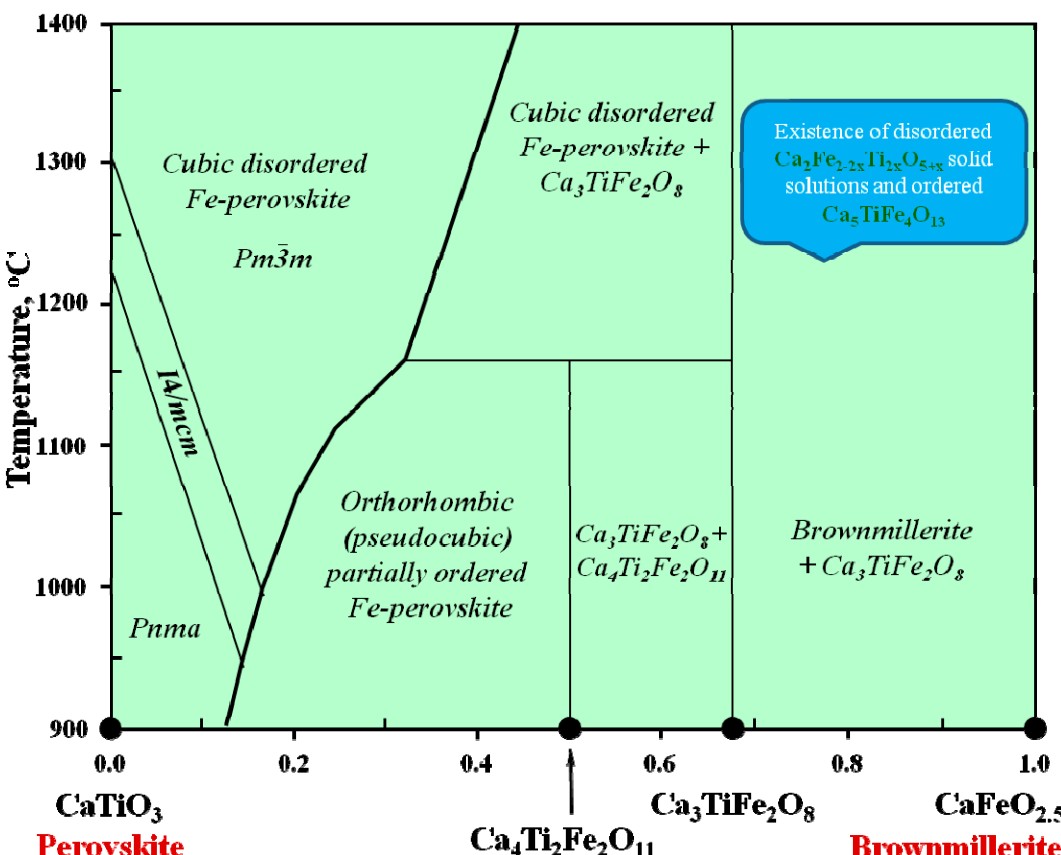

**Figure 13.** Phase diagram for the $CaTiO_3$–$CaFeO_{2.5}$ system, modified after [21,122–124].

### 4.2.5. Diversity of Perovskite–supergroup Minerals in the Hatrurim Larnite Rocks

Different types of larnite-bearing CM assemblages in the Hatrurim Basin (Table 2) may contain a single mineral of the perovskite–brownmillerite series (usually brownmillerite, rarely shulamitite or sharyginite), or two such minerals (brownmillerite/srebrodolskite + shulamitite/sharyginite, Fe-rich perovskite + shulamitite/sharyginite, Fe-rich perovskite + nataliakulikite) [4,25–27,86,106], which all

fit the phase diagram $CaTiO_3$–$CaFeO_{2.5}$ (Figure 13). The Fe-rich perovskite + shulamitite/sharyginite + brownmillerite/srebrodolskite assemblage was not found in the Hatrurim larnite rocks, although individual grains of perovskite and brownmillerite may occur in some rock samples. Such triple association was observed only in metacarbonate xenoliths of the Bellerberg volcano, Germany [26,115]. In both cases, it does not contradict with the phase diagram (Figure 13) because perovskite and brownmillerite have no immediate contacts (Figure 8D,E,I).

The sharyginite–shulamitite and srebrodolskite–brownmillerite non-ideal solid solutions can be well identified in the Hatrurim larnite CM rocks. Fe-rich and Al-poor opaque species, such as sharyginite and srebrodolskite, are more common to ye'elimite-rich larnite rocks, whereas Al-rich brownmillerite and shulamitite more often occur in the mayenite-bearing variety (Table 2). The reason is that the two rock types have different bulk contents of $Fe_2O_3$ (4.45 against 3.17 wt %, correspondingly) which resides, as $Fe^{3+}$, mainly in these opaques. On the other hand, Fe-rich perovskites in the mayenitic variety have $Fe^{3+}$- and Al-rich compositions. In gehlenite–larnite rocks, where most of Al and $Fe^{3+}$ are concentrated by melilite-group minerals, nataliakulikite and perovskite are scarce, while magnesioferrite is the main opaque phase.

The compositions of numerous perovskite–supergroup minerals from different Hatrurim areas were reported in some previous publications [1,4,25–28,32,86,106], and new data on Fe-rich supergroup members are presented in Tables 18–20 and Figure 14. The perovskite–supergroup minerals from the Hatrurim larnite CM rocks contain different amounts of Mn, Mg, Si, Cr, Zr, Nb and REE impurities: perovskite, nataliakulikite and sharyginite/shulamitite are more enriched in Si, Zr, Nb and REE but depleted in Mg relative to brownmillerite/srebrodolskite [25]. Manganese is low (<1 wt % MnO) because the sedimentary protoliths are depleted in Mn (Table 5). Unlike its natural analogs, brownmillerite from industrial cement clinkers commonly contains up to 10 wt % of impurities in total, and its average composition corresponds to $Ca_2Al(Fe,Mn)_{0.6}Mg_{0.2}Si_{0.15}Ti_{0.05}O_5$ [20,25,34].

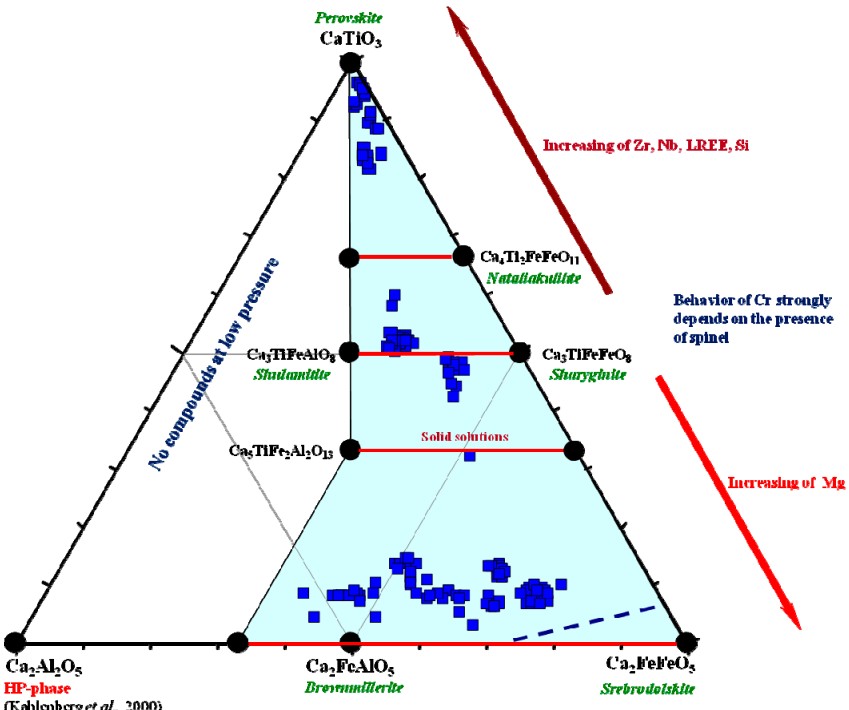

**Figure 14.** Mineral chemistry for perovskite–supergroup minerals from the Hatrurim larnite CM rocks in $CaTiO_3$–$Ca_2Fe_2O_5$–$Ca_2Al_2O_5$ coordinates, modified after [25]. Blue field shows natural compositions. Data are from this study (Table 18) and previous publications [4,25,26,32].

**Table 19.** Average mineral chemistry (WDS-EDS, wt %) of brownmillerite subgroup minerals from Hatrurim larnite CM rocks.

| Rock | Ye'elimite–Larnite Rocks | | | | | | | | Mayenite–Larnite Rocks | | | |
|---|---|---|---|---|---|---|---|---|---|---|---|---|
| Mineral | Srebrodolskite (*n* = 30) | | | | Brownmillerite (*n* = 4) | | | | Brownmillerite (*n* = 65) | | | |
| | Mean | S | Min | Max | Mean | S | Min | Max | Mean | S | Min | Max |
| $SiO_2$ | 0.61 | 0.18 | 0.28 | 0.96 | 0.99 | 0.20 | 0.71 | 1.16 | 0.75 | 0.22 | 0.43 | 1.71 |
| $TiO_2$ | 0.30 | 0.18 | <0.02 | 0.69 | 0.65 | 0.04 | 0.62 | 0.67 | 2.93 | 0.43 | 2.17 | 4.07 |
| $Cr_2O_3$ | 0.04 | 0.06 | <0.02 | 0.25 | 0.60 | 0.33 | 0.37 | 0.83 | 1.07 | 0.79 | 0.38 | 3.58 |
| $Al_2O_3$ | 5.11 | 2.36 | 0.49 | 10.30 | 11.04 | 1.59 | 9.52 | 13.05 | 13.64 | 1.66 | 10.56 | 17.59 |
| $Fe_2O_3$ | 50.66 | 3.05 | 44.19 | 56.64 | 41.89 | 1.27 | 40.82 | 43.42 | 34.97 | 3.07 | 26.46 | 39.43 |
| FeO* | 0.20 | 0.30 | <0.02 | 1.13 | 0.48 | 0.27 | 0.15 | 0.55 | 1.15 | 0.33 | 0.18 | 1.99 |
| MnO | 0.31 | 0.05 | 0.21 | 0.44 | 0.52 | 0.04 | 0.49 | 0.54 | <0.02 | | | |
| MgO | 0.33 | 0.17 | 0.20 | 0.74 | 0.50 | 0.16 | 0.40 | 0.73 | 0.70 | 0.15 | 0.46 | 1.09 |
| CaO | 42.42 | 0.38 | 41.51 | 43.00 | 43.89 | 0.28 | 43.67 | 44.30 | 44.72 | 0.45 | 43.71 | 45.69 |
| Total | 99.98 | | | | 100.54 | | | | 99.94 | | | |

*n* = number of samples, Mean = mean value, *S* = standard deviation, Min = minimum value, Max = maximum value; FeO and $Fe_2O_3$ are calculated by charge balance.

**Table 20.** Average mineral chemistry (WDS-EDS, wt %) of perovskite–supergroup minerals from Hatrurim larnite CM rocks.

| Rock | Mayenite–Larnite Rocks | | | | | | | | | | | |
|---|---|---|---|---|---|---|---|---|---|---|---|---|
| Mineral | Sharyginite (*n* = 7) | | | | Shulamitite (*n* = 8) | | | | Perovskite (*n* = 13) | | | |
| | Mean | S | Min | Max | Mean | S | Min | Max | Mean | S | Min | Max |
| $SiO_2$ | 0.96 | 0.30 | 0.56 | 1.33 | 1.04 | 0.10 | 0.90 | 1.20 | 2.86 | 0.17 | 2.59 | 3.14 |
| $TiO_2$ | 18.66 | 1.31 | 16.45 | 19.95 | 19.47 | 0.15 | 19.32 | 20.97 | 39.31 | 0.84 | 37.95 | 40.83 |
| $ZrO_2$ | 0.70 | 1.30 | <0.02 | 3.10 | <0.02 | | | | 0.55 | 0.11 | 0.43 | 0.69 |
| $Cr_2O_3$ | <0.02 | | | | 0.48 | 0.06 | 0.38 | 0.57 | 0.41 | 0.09 | 0.35 | 0.51 |
| $Al_2O_3$ | 5.93 | 0.33 | 5.42 | 6.35 | 9.33 | 0.37 | 7.88 | 9.92 | 3.13 | 0.22 | 2.82 | 3.51 |
| $Fe_2O_3$ | 30.27 | 0.30 | 29.93 | 30.85 | 25.57 | 0.45 | 25.15 | 26.59 | 11.68 | 0.45 | 10.84 | 12.22 |
| FeO* | | | | | 0.04 | 0.35 | 0.01 | 1.00 | | | | |
| MnO | 0.70 | 0.15 | 0.49 | 0.87 | <0.02 | | | | <0.02 | | | |
| MgO | 0.09 | 0.20 | 0.00 | 0.45 | 0.04 | 0.11 | <0.02 | 0.30 | <0.02 | | | |
| CaO | 42.75 | 0.16 | 42.55 | 43.00 | 43.33 | 0.11 | 42.55 | 43.48 | 42.44 | 0.20 | 42.17 | 42.66 |
| Total | 100.06 | | | | 99.30 | | | | 100.38 | | | |

*n* = number of samples, Mean = mean value, *S* = standard deviation, Min = minimum value, Max = maximum value; FeO* and $Fe_2O_3$ are calculated by charge balance.

The Fe-perovskite + sharyginite/shulamitite paragenesis in the Hatrurim larnite CM rocks indicates their formation at temperatures no lower than 1170–1200 °C (see the $CaTiO_3$–$Ca_2Fe_2O_3$ diagram in Figure 13). Thus, this assemblage can be used as a mineral thermometer for high-temperature–low-pressure metacarbonate rocks [25], as well as the newly discovered natural assemblage of Fe-perovskite + nataliakulikite [27]. The respective temperature estimates agree with previous reconstructions for larnite rocks [4].

Ti-rich phases (perovskite and nataliakulikite) are more resistant in the retrograde process than sharyginite/shulamitite, and all four phases have weaker hydration reactivity than Fe-rich brownmillerite/srebrodolskite. Yet, natural brownmillerite has been less reactive in prolonged supergene alteration of CM rocks in the Negev Desert than in its synthetic analogs (Figure 9A,B,F), and it often remains the only preserved clinker phase after retrograde alteration of larnite and spurrite varieties. Its grains become partly replaced by a brown isotropic X-ray amorphous phase of Ca–Al-ferrite-hydrate ($Ca(Al,Fe)_2O_4 \cdot nH_2O$) only in strongly weathered ye'elimite rocks that bear abundant ettringite, $CaCO_3$, and CSHs [1,26,76].

### 4.2.6. Spinel-Supergroup Minerals

Minerals of this supergroup are among the main opaque phases in the larnite rocks. According to the recent nomenclature [125], they belong to the oxyspinel group and are classified as spinel, magnesioferrite, and magnetite (Figure 7C,D,H and Figure 8B,E–G). Most of compositions correspond to the magnesioferrite–spinel solid solution with minor amounts of CaO (0.5–1.5 wt %), MnO (≤0.6 wt %), NiO (up to ≤1.7 wt %), and ZnO (≤1.7 wt %), and sometimes also CuO (≤0.4 wt %) (Table 21; Figure 15). Cr-rich spinel (~24 wt % $Cr_2O_3$) and magnetite (<1.5 w t% $TiO_2$) have been found in the mayenite-bearing variety, while magnesioferrite is of rare occurrence. The distribution of Cr in different opaques is mainly controlled by the abundance of Fe–Mg spinel, which is a main Cr host in the high-temperature Hatrurim CM rocks [86]. Ye'elimite-bearing rocks usually contain Al-rich magnesioferrite (10.56–19.59 wt % $Al_2O_3$), whereas Al-poor magnesioferrite (2.33–3.55 wt % $Al_2O_3$) is common to gehlenite–larnite rocks (Figure 15).

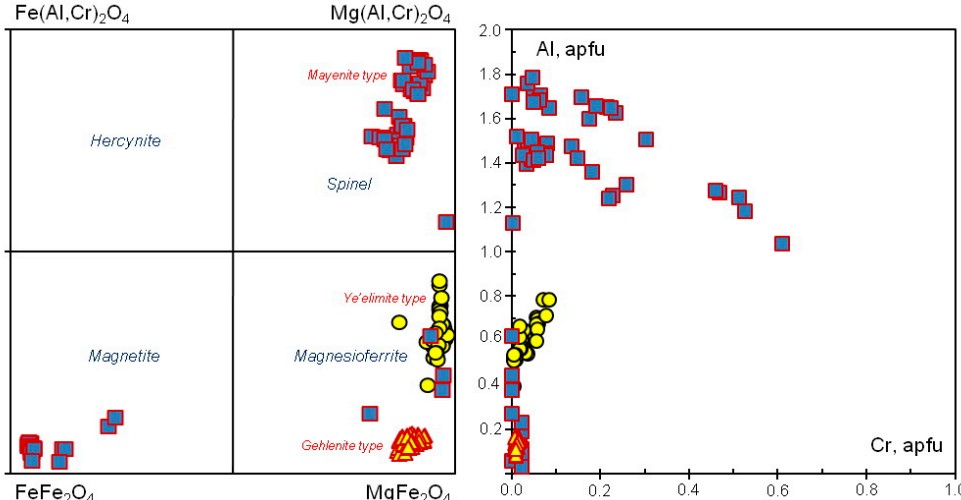

**Figure 15.** Composition variations of spinel-group minerals from the Hatrurim larnite CM rocks. Circles, squares, and triangles are ye'elimite–, mayenite–, and gehlenite-bearing larnite rocks, respectively.

**Melilite–group minerals** in the studied Hatrurim rocks are represented by gehlenite-dominant varieties (Table 22). Individual grains are commonly homogeneous in composition. Optical and chemical zoning occurs only in some gehlenite-rich rocks and the core-to-rim variations show mainly the increasing of ferrigehlenite end-member. Akermanite, ferroakermanite and natromelilite end-members are subordinate or negligible (Table 22; Figures 7 and 8F). The mineral is relatively resistant to supergene alteration. The main products of gehlenite hydration are strätlingite or hydrogarnet, which are commonly the phases of low-to-medium crystallinity (Figure 9C).

**Table 21.** Mineral chemistry (wt %) of spinel-group minerals in Hatrurim larnite CM rocks, Israel.

| Type | Sample | Phase | n | TiO$_2$ | Cr$_2$O$_3$ | V$_2$O$_3$ | Al$_2$O$_3$ | Fe$_2$O$_3$ | FeO | Mn$_2$O$_3$ | MnO | MgO | CaO | NiO | ZnO | CuO | Total | Ti | Cr+V | Al | Fe$^{3+}$ | Fe$^{2+}$ | Mn$^{3+}$ | Mn$^{2+}$ | Mg | Ca | Ni+Zn+Cu |
|---|---|---|---|---|---|---|---|---|---|---|---|---|---|---|---|---|---|---|---|---|---|---|---|---|---|---|---|
| Y | H-201 | Mfr | 18 | 0.21 | 1.05 | n.a. | 16.73 | 58.48 | 0.06 | | 0.14 | 19.75 | 0.79 | 1.68 | 1.07 | n.a. | 99.96 | 0.01 | 0.03 | 0.61 | 1.36 | 0.00 | | 0.00 | 0.91 | 0.03 | 0.07 |
| | | *S* | | *0.05* | *0.18* | | *0.93* | *1.09* | *0.06* | | *0.02* | *0.14* | *0.15* | *0.15* | *0.19* | *n.a.* | | | | | | | | | | | |
| Y | M5-31 | Mfr | 9 | 0.24 | 2.65 | <0.03 | 19.59 | 53.76 | 0.19 | | 0.11 | 20.16 | 0.93 | 0.67 | 1.72 | 0.08 | 100.11 | 0.01 | 0.06 | 0.70 | 1.23 | 0.01 | | 0.00 | 0.91 | 0.03 | 0.06 |
| | | *S* | | *0.03* | *0.45* | *0.00* | *1.69* | *2.44* | *0.26* | | *0.01* | *0.15* | *0.18* | *0.05* | *0.26* | *0.03* | | | | | | | | | | | |
| Y | Y-8 | Mfr | 1 | <0.02 | 0.18 | n.a. | 10.56 | 67.14 | 0.09 | | 0.40 | 19.67 | 1.41 | n.a. | 0.42 | n.a. | 99.87 | 0.00 | 0.01 | 0.39 | 1.60 | 0.00 | | 0.01 | 0.93 | 0.05 | 0.01 |
| Y | YV-411 | Mfr | 6 | 0.26 | 0.74 | n.a. | 15.64 | 60.07 | 0.16 | | 0.62 | 19.91 | 1.04 | 0.48 | 0.91 | n.a. | 99.81 | 0.01 | 0.02 | 0.57 | 1.40 | 0.00 | | 0.02 | 0.92 | 0.03 | 0.03 |
| | | *S* | | *0.05* | *0.06* | | *0.24* | *0.36* | *0.21* | | *0.10* | *0.14* | *0.23* | *0.06* | *0.14* | | | | | | | | | | | | |
| Y | Y-6-3 | Mfr | 2 | <0.02 | 0.70 | n.a. | 17.86 | 58.49 | 1.83 | | n.a. | 20.17 | 1.14 | n.a. | n.a. | n.a. | 100.18 | 0.00 | 0.02 | 0.64 | 1.34 | 0.05 | | | 0.92 | 0.04 | |
| Y | Y-5-1 | Mfr | 7 | <0.02 | 0.27 | n.a. | 14.32 | 60.23 | 0.06 | 2.54 | | 19.68 | 1.21 | 0.59 | 1.39 | n.a. | 100.29 | 0.00 | 0.01 | 0.52 | 1.41 | 0.00 | 0.06 | | 0.91 | 0.04 | 0.05 |
| | | *S* | | *0.00* | *0.07* | | *0.33* | *0.48* | *0.04* | *0.10* | | *0.09* | *0.11* | *0.15* | *0.18* | | | | | | | | | | | | |
| M | H-401 | Mag | 31 | 1.05 | 0.61 | 0.12 | 2.39 | 64.14 | 29.98 | | 0.14 | 1.16 | 0.46 | n.a. | n.a. | n.a. | 100.04 | 0.03 | 0.02 | 0.11 | 1.81 | 0.94 | | 0.00 | 0.07 | 0.02 | |
| | | *S* | | *0.45* | *0.14* | *0.05* | *0.81* | *1.37* | *1.87* | | *0.05* | *0.91* | *0.18* | | | | | | | | | | | | | | |
| M | M4-251 | Spl | 9 | 0.10 | 6.10 | <0.03 | 53.03 | 12.81 | 4.23 | | 0.10 | 22.66 | 0.47 | 0.18 | 0.71 | 0.05 | 100.43 | 0.00 | 0.13 | 1.62 | 0.25 | 0.09 | | 0.00 | 0.88 | 0.01 | 0.02 |
| | | *S* | | *0.04* | *7.52* | *0.00* | *8.13* | *1.49* | *0.90* | | *0.01* | *1.47* | *0.18* | *0.12* | *0.04* | *0.08* | | | | | | | | | | | |
| M | M4-218 | Spl | 13 | 0.13 | 5.44 | <0.03 | 44.56 | 22.59 | 5.16 | | 0.10 | 21.37 | 0.50 | 0.08 | 0.33 | n.a. | 100.26 | 0.00 | 0.12 | 1.42 | 0.46 | 0.12 | | 0.00 | 0.86 | 0.01 | 0.01 |
| | | *S* | | *0.03* | *3.80* | *0.01* | *3.64* | *1.74* | *1.06* | | *0.01* | *0.77* | *0.09* | *0.01* | *0.07* | | | | | | | | | | | | |
| | | Mfr | 1 | 0.05 | n.a. | n.a. | 7.04 | 70.13 | 4.90 | | 0.59 | 16.60 | 1.18 | n.a. | n.a. | n.a. | 100.49 | 0.00 | | 0.27 | 1.73 | 0.13 | | 0.02 | 0.81 | 0.04 | |
| M | M4-215 | Spl | 12 | 0.15 | 13.19 | <0.03 | 49.64 | 9.77 | 3.30 | | 0.03 | 23.52 | 0.41 | <0.02 | 0.10 | <0.03 | 100.11 | 0.00 | 0.27 | 1.53 | 0.19 | 0.07 | | 0.00 | 0.92 | 0.01 | 0.00 |
| | | *S* | | *0.06* | *8.10* | *0.00* | *8.55* | *3.05* | *0.64* | | *0.02* | *0.84* | *0.11* | *0.00* | *0.04* | *0.00* | | | | | | | | | | | |
| M | G-7 | Spl | 3 | n.a. | 1.88 | n.a. | 44.42 | 26.43 | 4.44 | | n.a. | 21.14 | 0.87 | 0.86 | n.a. | n.a. | 100.04 | | 0.04 | 1.42 | 0.54 | 0.10 | | | 0.86 | 0.03 | 0.02 |
| | | Mfr | 1 | n.a. | <0.02 | n.a. | 17.35 | 60.24 | 0.02 | | 0.40 | 20.89 | 1.30 | n.a. | n.a. | n.a. | 100.19 | | 0.00 | 0.62 | 1.38 | 0.00 | | 0.01 | 0.95 | 0.04 | |
| M | W-10-2 | Spl | 5 | 0.21 | 2.19 | n.a. | 44.70 | 25.22 | 4.96 | | n.a. | 20.61 | 0.79 | 0.93 | 0.75 | n.a. | 100.37 | 0.01 | 0.05 | 1.43 | 0.52 | 0.11 | | | 0.83 | 0.02 | 0.04 |
| | | *S* | | *0.48* | *0.68* | | *0.44* | *1.10* | *1.22* | | | *0.63* | *0.17* | *0.06* | *0.18* | | | | | | | | | | | | |
| M | Y-9-1 | Mfr | 2 | <0.02 | <0.02 | n.a. | 11.11 | 67.33 | 0.11 | | <0.02 | 20.85 | 0.67 | n.a. | n.a. | n.a. | 100.06 | 0.00 | 0.00 | 0.41 | 1.59 | 0.00 | | 0.00 | 0.98 | 0.02 | |
| M | CONCR | Spl | 1 | n.a. | 0.11 | n.a. | 34.33 | 41.10 | 0.03 | | 0.11 | 23.55 | 0.48 | n.a. | n.a. | n.a. | 99.71 | | 0.00 | 1.13 | 0.87 | 0.00 | | 0.00 | 0.98 | 0.02 | |
| Geh | Y-10-5 | Mfr | 4 | <0.02 | 0.32 | n.a. | 2.33 | 75.69 | 1.77 | | 1.15 | 17.64 | 1.14 | <0.02 | <0.03 | n.a. | 100.05 | 0.00 | 0.01 | 0.09 | 1.90 | 0.05 | | 0.03 | 0.88 | 0.04 | 0.00 |
| Geh | W-11-3 | Mfr | 45 | <0.02 | 0.41 | n.a. | 3.55 | 74.01 | 1.30 | | 0.27 | 17.25 | 1.50 | 0.81 | 0.66 | 0.44 | 100.23 | 0.00 | 0.01 | 0.14 | 1.85 | 0.04 | | 0.01 | 0.85 | 0.05 | 0.04 |
| | | *S* | | *0.04* | *0.09* | | *0.41* | *0.49* | *0.62* | | *0.05* | *0.40* | *0.16* | *0.10* | *0.14* | *0.09* | | | | | | | | | | | |

Formula is based on three cations and four oxygens. Fe$_2$O$_3$, FeO, Mn$_2$O$_3$ and MnO are calculated by charge balance. Y = ye'elimite-bearing rocks; M = mayenite-bearing rocks; Gh = gehlenite-rich rocks. Symbols for minerals: Mfr = magnesioferrite; Mag = magnetite; Spl = spinel; *n* = number of analyses; n.a. = not analyzed; *S* = standard deviation for *n* > 5.

**Table 22.** Mineral chemistry (WDS-EDS, wt %) of gehlenite from Hatrurim larnite CM rocks.

| Type | Gh | Gh | Gh | Gh | M | M | M | M | Y |
|---|---|---|---|---|---|---|---|---|---|
| Sample | W-11-3 | W-11-3 | Y-10-5 | W-10-2 | G-7 | H-401 | CONCR | M4-251 | YV-410 |
| Position | core | rim | | | | | | | |
| *n* | 36 | 7 | 5 | 1 | 9 | 7 | 1 | 4 | 7 |
| $SiO_2$ | 22.17 | 21.60 | 25.05 | 21.97 | 22.11 | 26.16 | 22.28 | 21.71 | 25.92 |
| $TiO_2$ | 0.04 | <0.01 | <0.01 | <0.01 | <0.01 | 0.09 | 0.11 | 0.28 | 0.09 |
| $Al_2O_3$ | 32.91 | 28.05 | 29.54 | 34.89 | 34.55 | 26.68 | 34.32 | 35.58 | 26.68 |
| $Fe_2O_3$ | 4.25 | 10.48 | 2.12 | 2.50 | 2.37 | 4.47 | 2.13 | 1.87 | 5.30 |
| FeO | 0.28 | 0.18 | 1.09 | 0.28 | 0.69 | 1.34 | 0.51 | 0.17 | 0.59 |
| MgO | 0.28 | 0.35 | 1.78 | 0.10 | <0.02 | 1.41 | 0.08 | 0.00 | 1.41 |
| CaO | 40.26 | 39.17 | 40.24 | 40.43 | 40.33 | 39.07 | 40.15 | 40.53 | 39.44 |
| SrO | n.a. | n.a. | n.a. | n.a. | n.a. | 0.18 | 0.21 | 0.24 | 0.18 |
| $Na_2O$ | <0.02 | <0.02 | n.a. | n.a. | n.a. | 0.49 | 0.07 | <0.02 | 0.49 |
| $K_2O$ | <0.01 | <0.01 | n.a. | n.a. | n.a. | 0.24 | 0.03 | <0.01 | 0.26 |
| Sum | 100.20 | 98.79 | 99.61 | 99.92 | 99.81 | 99.69 | 99.66 | 100.18 | 100.44 |
| Formula based on 5 cations and 7 oxygens | | | $A_2B[T_2O_7]$ | | | | | | |
| Si | 1.026 | 1.027 | 1.158 | 1.011 | 1.020 | 1.217 | 1.029 | 0.996 | 1.204 |
| $Al^{IV}$ | 0.974 | 0.973 | 0.842 | 0.989 | 0.980 | 0.783 | 0.971 | 1.004 | 0.796 |
| $\Sigma[T]$ | 2.000 | 2.000 | 2.000 | 2.000 | 2.000 | 2.000 | 2.000 | 2.000 | 2.000 |
| $Al^{VI}$ | 0.822 | 0.598 | 0.768 | 0.903 | 0.898 | 0.679 | 0.897 | 0.920 | 0.664 |
| Ti | 0.001 | | | | | 0.003 | 0.004 | 0.009 | 0.003 |
| $Fe^{3+}$ | 0.148 | 0.375 | 0.074 | 0.086 | 0.082 | 0.157 | 0.074 | 0.064 | 0.185 |
| $Fe^{2+}$ | 0.011 | 0.007 | 0.042 | 0.011 | 0.027 | 0.052 | 0.020 | 0.007 | 0.023 |
| Mg | 0.019 | 0.025 | 0.123 | 0.007 | | 0.098 | 0.006 | | 0.098 |
| $\Sigma[B]$ | 1.002 | 1.005 | 1.007 | 1.007 | 1.007 | 0.990 | 1.000 | 1.001 | 0.973 |
| Ca | 1.998 | 1.995 | 1.993 | 1.993 | 1.993 | 1.947 | 1.987 | 1.993 | 1.963 |
| Sr | | | | | | 0.005 | 0.006 | 0.006 | 0.005 |
| Na | | | | | | 0.044 | 0.006 | | 0.044 |
| K | | | | | | 0.015 | 0.002 | | 0.015 |
| $\Sigma[A]$ | 1.998 | 1.995 | 1.993 | 1.993 | 1.993 | 2.010 | 2.000 | 1.999 | 2.027 |
| End members | | | | | | | | | |
| $Ca_2AlAlSiO_7$ | 82.04 | 59.53 | 76.32 | 89.65 | 89.18 | 62.78 | 88.95 | 91.95 | 62.37 |
| $Ca_2Fe^{3+}AlSiO_7$ | 4.78 | 37.31 | 7.31 | 8.59 | 8.18 | 15.82 | 7.40 | 6.44 | 19.05 |
| $Ca_2MgSi_2O_7$ | 1.94 | 2.47 | 12.20 | 0.68 | 0.00 | 9.91 | 0.55 | 0.00 | 10.03 |
| $Ca_2Fe^{2+}Si_2O_7$ | 1.23 | 0.70 | 4.17 | 1.08 | 2.64 | 5.62 | 2.33 | 1.60 | 2.67 |
| $NaCaAlSi_2O_7$ | 0.00 | 0.00 | 0.00 | 0.00 | 0.00 | 5.87 | 0.77 | 0.01 | 5.89 |

MnO, $Cr_2O_3$ and $P_2O_5$ are below the detection limits (<0.02 wt %). n = number of analyses; n.a. = not analyzed. Gh = gehlenite-rich rocks; M = mayenite-bearing rocks; Y = ye'elimite-bearing rocks. FeO and $Fe_2O_3$ are calculated by charge balance. Core = core of grains; rim = rim of grains. $\Sigma[T] = Si + Al^{IV}$; $\Sigma[B] = Al^{VI}+Ti+ Fe^{3+}+ Fe^{2+}+Mg$; $\Sigma[A] = Ca+Sr+Na+K$.

**Periclase** is minor to accessory in the Hatrurim larnite CM rocks and is generally more common to the mayenite variety (Table 2). It exists as interstitial 5–10 μm round anhedral grains (Figures 7F and 8F,I), which are most often fresh, homogeneous, and filled with numerous magnesioferrite inclusions. It belongs to the MgO–FeO ss, with 2.55–23.93 wt % FeO (~16 wt % on average), and up to 3.50 wt % NiO and 2.17 wt % ZnO (Table 23). Periclase from the ye'elimite variety is richer in ZnO and NiO and poorer in Fe than that from mayenite—and gehlenite-bearing rocks.

Note that periclase from Jordanian spurrite marbles derived from Zn- and Ni-enriched Muwaqqar Fm sedimentary protoliths, on the contrary, contain small amounts of FeO (≤0.46 wt %) but high ZnO (15.24–23.13 wt %), NiO (4.80–5.59 wt %), and CuO (2.54–4.06 wt %). The average compositions of the MgO–FeO and MgO–ZnO solid solutions are $(Mg_{0.88}Fe_{0.10}Zn_{0.01}Ni_{0.01})O$ and $(Mg_{0.85}Zn_{0.12}Ni_{0.03}Cu_{0.02})O$, respectively. Ferropericlase is common in high-temperature contact marbles [99], while natural Zn-rich MgO–ZnO ss (up to 0.12 mol. % ZnO) have been discovered for

the first time. This unusual chemistry of periclase is due to the geochemical specificity of Zn-enriched and Fe-depleted sedimentary parent rocks [78]. Synthetic (Mg,Zn)O ss contains up to 0.3 mol. % ZnO at $T$ = 1100 °C [126].

**Table 23.** Average mineral chemistry (WDS-EDS, wt %) of periclase in Hatrurim larnite CM rocks and periclase from Jordanian CM marbles, compared.

| Locality | Hatrurim Basin | | | | | | Jordan | | | |
|---|---|---|---|---|---|---|---|---|---|---|
| Type | M | M | Y | Gh | Gh | Gh | Spurrite marble | | | |
| Sample | M5-32 | M4-215 | YV-412 | W-10-2 | W-10-2 | W-10-2 | Mean | S | Min | Max |
| *n* | 10 | 7 | 4 | 2 | 1 | 2 | 15 | | | |
| Cr$_2$O$_3$ | 0.21 | <0.02 | n.a. | n.a. | n.a. | n.a. | n.a. | n.a. | n.a. | n.a. |
| Al$_2$O$_3$ | 0.20 | <0.02 | n.a. | n.a. | n.a. | n.a. | n.a. | n.a. | n.a. | n.a. |
| FeO | 20.10 | 23.93 | 2.55 | 7.66 | 14.77 | 23.84 | 0.43 | 0.04 | 0.37 | 0.46 |
| MnO | 0.18 | 0.10 | n.a. | n.a. | n.a. | n.a. | n.a. | n.a. | n.a. | n.a. |
| MgO | 75.38 | 75.22 | 90.69 | 87.85 | 80.67 | 74.07 | 70.00 | 2.02 | 66.45 | 71.51 |
| CaO | 0.44 | 0.45 | 0.78 | 0.45 | 0.60 | 0.78 | <0.01 | | | |
| NiO | 1.98 | 0.22 | 3.50 | 2.50 | 2.32 | n.a. | 5.13 | 0.35 | 4.80 | 5.59 |
| ZnO | 1.45 | <0.03 | 2.17 | 1.00 | 1.01 | n.a. | 19.77 | 2.84 | 15.24 | 23.13 |
| CuO | <0.03 | <0.03 | n.a. | n.a. | n.a. | n.a. | 3.34 | 0.54 | 2.54 | 4.06 |
| Total | 99.93 | 99.92 | 99.68 | 99.46 | 99.37 | 98.68 | 98.67 | | | |
| Formula based on 1 cation | | | | | | | | | | |
| Cr | 0.00 | 0.00 | | | | | | | | |
| Al | 0.00 | 0.00 | | | | | | | | |
| Fe | 0.13 | 0.15 | 0.01 | 0.05 | 0.09 | 0.15 | 0.00 | | | |
| Mn | 0.00 | 0.00 | | | | | | | | |
| Mg | 0.85 | 0.84 | 0.95 | 0.93 | 0.89 | 0.84 | 0.85 | | | |
| Ca | 0.00 | 0.00 | 0.01 | 0.00 | 0.00 | 0.01 | | | | |
| Ni | 0.01 | 0.00 | 0.02 | 0.01 | 0.01 | | 0.03 | | | |
| Zn | 0.01 | 0.00 | 0.01 | 0.01 | 0.01 | | 0.12 | | | |
| Cu | 0.00 | 0.00 | | | | | 0.02 | | | |

SiO$_2$ (<0.01 wt %), TiO$_2$ (<0.01 wt %) and V$_2$O$_5$ (<0.03 wt %) are below the detection limits. *n* = number of samples, Mean = mean value, *S* = standard deviation, Min = minimum value, Max = maximum value. Y = ye'elimite–larnite rocks; M = mayenite–larnite rocks; Gh = gehlenite-rich larnite rocks.

Periclase is an undesirable constituent of clinker, and its content is often limited in many specifications of cements [20,103,127]. Its presence indicates excess Mg in the raw materials relative to the equilibrium compositions of ferrite [20]. In natural rocks akin to the products of clinker hardening, periclase hydrates rather slowly to yield brucite, which leads to local expansion and cracking of the hydrated rock.

### 4.2.7. Accessory Mineralization

The Hatrurim larnite CM rocks contain accessory fine-grained Fe, Cu, K and Ca sulfides, Cu and Ag selenides, and Ca–Zr and Ca–U double oxides. The accessory mineralization is quite scarce unlike that of the Jordanian spurrite CM marbles which store rich and diverse sulfide/selenide (Zn, Cd, Ni, Mo, Ag) and/or oxide (Zn, Cd, U, Ni) accessories [3,6–8].

**Sulfides and selenides** Fine grains of pyrrhotite occur in all types of larnite rocks, whereas chalcopyrite, oldhamite and K–Fe sulfide are found as single fine particles only in the mayenite-bearing variety (Table 2). Two rare selenides of Cu$_2$Se (berzelianite or bellidoite) and eucairite CuAgSe were found in a ye'elimite-bearing sample with 96 ppm Se (much higher than the average over all samples <14 ppm). The Cu$_2$Se phase (56.75 wt % Cu and 41.11 wt % Se) also contains up to 0.30 wt % S and 0.35 wt % Fe, whereas eucairite is stoichiometric CuAgSe with 26.35 wt % Cu, 42.64 wt % Ag, and 30.45 wt % Se.

**U-bearing oxides** The Hatrurim mayenite– and ye'elimite–larnite rocks with ~5 to 62 ppm U commonly bear double Ca–U(VI) oxides with $CaUO_4$ stoichiometry (vorlanite or protovorlanite) (Figure 8I) occurring as small (≤15 μm) interstitial anhedral grains. The mineral, with the U:Ca ratio highest among double Ca–U(VI) oxides, appears in rocks with only ~5 ppm U, and its percentage increases proportionally to the bulk U content in the rocks. The $CaUO_4$ phase is commonly partly altered and replaced by hydrated Ca uranates close to X-phase ($Ca_2UO_5 \cdot 2–3H_2O$). Secondary hydrated Ca urinates after primary Ca–U(VI) oxides (with Si, Fe, Al and F impurities) were previously found in U-rich CM marbles from the Tulul Al Hammam area in central Jordan [6]. Among these, the so-called X-phase ($Ca_2UO_5 \cdot 2–3H_2O$) immobilizes most of U(VI) [14]. The long-term stability of both phases in supergene environments is confirmed by systematic findings of the $CaUO_4$ phase partly converted into $Ca_2UO_5 \cdot 2-3H_2O$ in the Mottled Zone CM rocks [6,14,29].

Lakargiite (Ca(Zr,Ti,U)$O_3$), another accessory mineral, likewise can incorporate U as a main impurity [6]. Although larnite CM rocks from different Hatrurim localities have relatively high bulk Zr contents of ~64–71 ppm, lakargiite (Ca(Zr,Ti,U)$O_3$), which is typical of different high-temperature-low pressure metacarbonate rocks [6,128], was found only as sporadic clots of fine (1–2 μm) inclusions in sharyginite (Figure 8H). It is rare because Zr impurity mostly incorporates into Ti-rich perovskite–supergroup minerals (Table 18; Figure 14). Lakargiite, including its U-enriched variety, is highly resistant to weathering.

### 4.2.8. Ba Mineralization

**Barite–hashemite solid solutions** occur sporadically in the Hatrurim larnite CM rocks. Barite is a primary phase in the interstitial space of fresh larnite rocks, whereas hashemite is found in the matrix of secondary CSHs, ettringite, etc. The $CrO_3$ and $SO_3$ contents have large ranges, as in barite–hashemite solid solutions from other Hatrurim outcrops [129,130]. These minerals also contain SrO (0.7–1.9 wt %), CaO (0.5–1.6 wt %) and rarer FeO (≤0.7 wt %) impurities (Table 24).

**Table 24.** Representative WDS-EDS analyses (wt %) of minerals of barite–hashemite series in Hatrurim larnite CM rocks.

| Type | Y | Y | Y | Y | Y | Y | Gh | Gh |
|---|---|---|---|---|---|---|---|---|
| Sample | YV-411 | Y-6-3 | Y-6-3 | Y-8-1 | Y-5-1 | M5-31 | W-11-3 | W-11-3 |
| Mineral | Barite | Barite | Barite | Barite | Barite | Barite | Barite | Hashemite |
| n | 7 | 3 | 2 | 2 | 5 | 3 | 7 | 6 |
| BaO | 64.00 | 62.98 | 61.85 | 60.23 | 63.43 | 61.58 | 62.76 | 61.45 |
| SrO | 0.59 | 0.79 | 0.90 | 1.91 | 0.65 | 2.06 | 0.75 | 0.44 |
| CaO | 0.76 | 0.75 | 0.92 | 1.59 | 0.84 | 0.86 | 0.67 | 0.53 |
| FeO | 0.04 | 0.67 | n.a. | 0.50 | 0.67 | n.a. | n.a. | n.a. |
| $CrO_3$ | 0.87 | 3.28 | 18.64 | 8.38 | 3.14 | 0.48 | 16.77 | 28.20 |
| $SO_3$ | 33.23 | 31.71 | 17.37 | 26.80 | 31.47 | 34.30 | 18.91 | 9.40 |
| Total | 99.49 | 100.17 | 99.67 | 99.40 | 100.20 | 99.28 | 99.86 | 100.03 |
| Formula based on 4 oxygens | | | | | | | | |
| Ba | 0.98 | 0.95 | 0.98 | 0.92 | 0.96 | 0.93 | 1.00 | 0.99 |
| Sr | 0.01 | 0.02 | 0.02 | 0.04 | 0.01 | 0.05 | 0.02 | 0.01 |
| Ca | 0.03 | 0.03 | 0.04 | 0.07 | 0.03 | 0.04 | 0.03 | 0.02 |
| Fe | 0.00 | 0.02 | | 0.02 | 0.02 | | | |
| Cr | 0.02 | 0.08 | 0.46 | 0.20 | 0.07 | 0.01 | 0.41 | 0.70 |
| S | 0.97 | 0.92 | 0.53 | 0.79 | 0.92 | 0.99 | 0.58 | 0.29 |

*n* = number of analyses; n.a. = not analyzed; Y = ye'elimite–larnite rocks; M = mayenite–larnite rocks; Gh = gehlenite-rich larnite rocks.

Quite coarse (up to 50 μm) and abundant crystals (Figure 8B,C) of the **zadovite–group** minerals (from gazeevite $BaCa_6(SiO_4)_2(SO_4)_2O$ to intermediate members of the gazeevite–zadovite

(BaCa$_6$((SiO$_4$)(PO$_4$))(PO$_4$)$_2$F) series [131]) were found systematically only in samples from the Har Ye'elim area, where Ba reaches 5000 ppm.

## 5. Conclusions and Implications

### 5.1. Mineralogical Productivity of Larnite CM Rocks and Element Partitioning

By December 2018, there had been altogether 140 IMA-approved primary mineral species described in the peculiar high- and ultrahigh-temperature and low-pressure CM rocks of the Hatrurim Formation (Mottled Zone). Their mineralogical productivity was estimated as high as 4.67 (K = M$_{minerals}$/N$_{elements}$ = 140/30), higher than the 3.56 (K = 89/25) ratio over all samples of paralava and clinker-like Ca$_2$SiO$_4$-bearing rocks we have studied and seen in the literature. The mineralogy of the Hatrurim CM rocks varies from predominant oxygenated compounds (73 species, 82.0%) to sulfides + selenides + arsenides (10 species, 11.2%), phosphides (2 species, 2.3%), and native element forms (4 species, 4.5%) (Figure 16). The high mineralogical diversity is due to variations of element contents in the Ca-rich precursor sedimentary mixtures and of redox conditions during CM events.

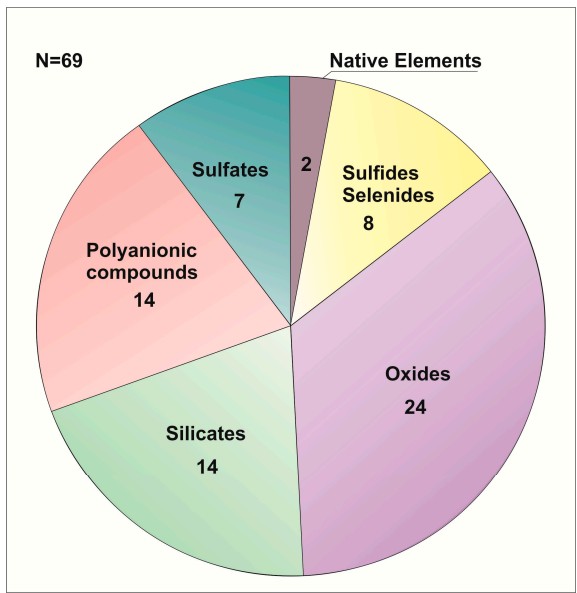

**Figure 16.** Distribution of chemical classes of minerals in the Hatrurim Fm larnite CM rocks. N is the number of species.

Our collection of totally decarbonated clinker-like Ca$_2$SiO$_4$-bearing rocks (mainly from five areas in the Hatrurim Basin) includes only thirty-eight mineral species where nineteen elements of interest are essential constituents. This is consistent with low productivity (K = 2) resulting from a narrow range of formation conditions and protolith compositions. All larnite rocks we analyzed formed by sintering of sediments from the same stratigraphic unit, mainly under oxidizing conditions. The list extends to sixty nine mineral species formed out of twenty-one elements (with a higher productivity of 3.14) when our data are combined with published evidence for rocks of the same type which likewise were derived from the Ghareb Fm sediments but belong to other Hatrurim Fm complexes or other localities within the Hatrurim Basin.

Correlation between the number of mineral species and the average concentration of respective elements was analyzed for data that represent clinker-like Ca$_2$SiO$_4$-bearing rocks using the algorithm of Christy (2015; 2018) [132,133], which was originally designed for global mineral data and crustal atomic abundances of elements. Average element abundances (A = atomic abundance in ppm) in the rocks plotted in log–log coordinates against the number of respective species (S) show a power–law relationship (Table S2; Figure 17). The minerals of clinker-like Ca$_2$SiO$_4$-bearing rocks mainly consist of

elements that are the most abundant in the upper crust but their descending order of species numbers differs from that revealed by Christy (2015; 2018) [132,133] and corresponds to the chemical features of organic-rich calcareous sedimentary precursors: 59 O, 49 Ca, 28 Si, 19 Fe, 18 S, 14 Al, 9 Mg, 8 Cu, 7 K, 6 F, 5 P, Ti and Ba, and 4 Na species (Figure 18). The $Ca_2SiO_4$-bearing rocks are also remarkable by the absence of C compounds (carbon escaped upon complete decarbonation of sediments at $T \geq 1000\,°C$) and by anhydrous compositions of primary CM minerals, except for mayenite supergroup minerals that contain $H_2O$ or (OH)-groups.

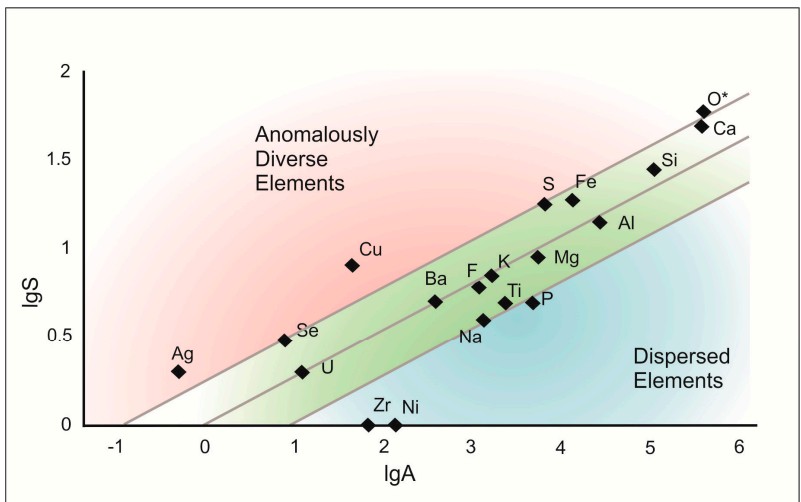

**Figure 17.** Number of mineral species (lgS) formed by essential constituent elements versus abundances of these elements in atomic ppm (lgA) in Hatrurim Fm larnite CM rocks. Lines on either side of the central trend present 95% confidence limits for the fit. Based on idea of A. Christy [132,133].

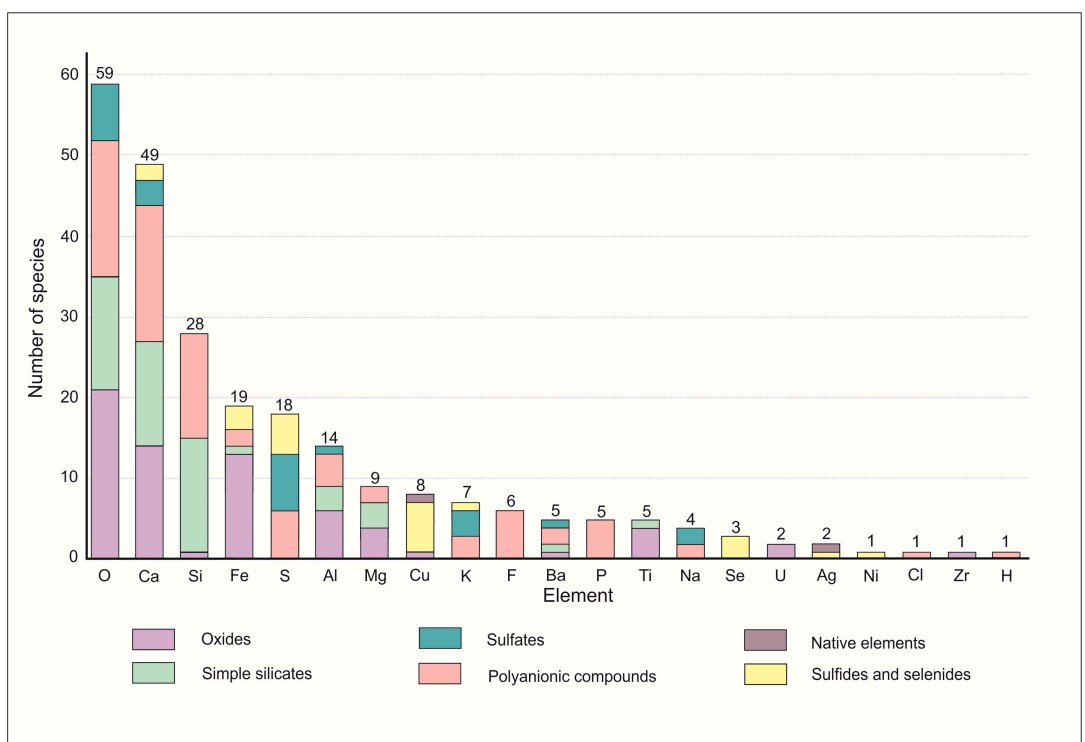

**Figure 18.** Mineralogical diversity of natural $Ca_2SiO_4$-bearing rocks and productivity of essential constituent elements.

The A–S relationships are linear, and seven elements (O, Ca, Fe, S, Cu, Se, and Ag) out of twenty-one principle constituents of mineral species form a large number of distinct mineral species in clinker-like rocks (Figure 17). Relative to the main trend, the number of species is the greatest for Cu (with an average content as low as ~43 ppm), and the second greatest for Fe, S and Se. The reason for the high productivity is that these elements are polyvalent and can exhibit chalcophile or lithophile behavior in different environments and form metallic, sulfide/selenide or oxygen-bearing compounds.

Trace elements in the analyzed $Ca_2SiO_4$-bearing rocks are commonly within tens to hundreds ppm, but enhanced local concentrations of Ba, Cu, Ni, Zr, U, Ag, and Se make relatively easy the formation of distinct minerals that occur as sporadic grains. The diversity of the Hatrurim mineral world is additionally supported by trace and minor elements which become mineral-forming species for the lack of the appropriate hosts in the Ca-rich ultra-high temperature-low pressure and commonly oxygenated environments. Namely, Ba forms five species, Se forms three species, U and Ag form two species each, and Ni and Zr form one species each (Figure 18). The P, Ti, Na (main), Ni and Zr (trace) elements have relatively low productivity in clinker-like rocks. Distinct phosphorus minerals in larnite rocks are few, because most of the bulk $P_2O_5$ content is bound (together with K and Na) in rock-forming $Ca_2SiO_4$ phases and in fluorapatite–fluorellestadite ss. For instance, larnites from mayenite– and ye'elimite-bearing rocks store ~37% and ~52% of the total P budget, respectively, while the rocks contain ~60% of larnite on average.

Summing up the wealth of currently available data on anhydrous clinker-like rocks from the Hatrurim Fm complexes, one can see that most of the compounds in anhydrous clinker-like rocks are chemically simple and consist of two or three elements: eight species of Ca, Fe and Mg oxides, Ca and Fe sulfides and Cu selenides, and twenty-six species of double oxides (Ca–Ti–O; Ca–Zr–O; Ca–U–O; Ca–Fe–O; Ca–Al–O; Fe–Mg–O; Fe–Ba–O; Cu–Fe–O; Mg–Al–O), simple Ca silicates (Ca–Si–O), sulphates (S–O) with one cation (Na, Ca, K, Ba), Cu–Ca and Cu–Fe sulfides (CuFeS and CaCuS), and one selenide species (AgCuSe). The compounds of four elements (nineteen species) include silicates (Si–O) with two cations (Ca–Mg, Ca–Al, Ba–Al), polyanion Ca compounds (Ca–Si–O–F, Ca–Si–O–S, Ca–Si–P–O, Ca–P–O–F, Ca–Al–O–Cl, Ca–Al–O–F), sulfates (Ca–Al–S–O, Ca–K–S–O, K–Na–S–O), and oxides (Ca–Al–Fe, Ca–Ti–Fe). Eight mineral species (including six newly discovered minerals) consist of five elements: two oxides shulamitite ($Ca_3TiFe^{3+}AlO_8$) and $Ca_2Mg_2Fe_{10}(Al,Fe)_4O_{25}$, silicate schorlomite ($Ca_3(Ti,Fe)_2((Si,Fe)O_4)_3$), polyanionic compounds fluorellestadite ($Ca_{10}((SO_4)_3(SiO_4)_3)_6F_2$), and two new minerals of dargaite ($BaCa_{12}(SiO_4)_4(SO_4)_2O_3$) and gazeevite ($BaCa_6(SiO_4)_2(SO_4)_2O$); fluorkyuygenite ($Ca_{12}Al_{14}O_{32}((H_2O)_4F_2)$) and unknown K–Fe–Cu–Ni–S sulfide. The six most complex compounds (among them four new minerals) consist of six elements: complex oxide nataliakulikite ($Ca_4Ti_2(Fe^{3+},Fe^{2+})(Si,Fe^{3+},Al)O_{11}$), silicates dorrite ($Ca_2(Mg_2Fe^{3+}_4)(Al_4Si_2O_{20})$) and khesinite ($Ca_4Mg_2Fe^{3+}_{10}O_4[(Fe^{3+}_{10}Si_2)O_{36}]$), ($PO_4$)-substituted silicate flamite ($Ca_{8-x}(Na,K)_x(SiO_4)_{4-x}(PO_4)_x$), and nabimusaite ($KCa_{12}(SiO_4)_4(SO_4)_2O_2F$).

Minerals in metacarbonate CM larnite rocks are mainly of cubic (25.8%) or orthorhombic (28.8%) symmetry; medium symmetric phases are less frequent but reach 27.3% in total, while monoclinic (13.6%) and triclinic (4.6%) symmetries typical of hydrates ($H_2O$ and/or (OH)-bearing minerals) are rare (Figure 19). This distribution differs from that for lithospheric minerals, where monoclinic phases are predominant (38%), but is similar to that for synthetic anhydrous inorganic compounds [134].

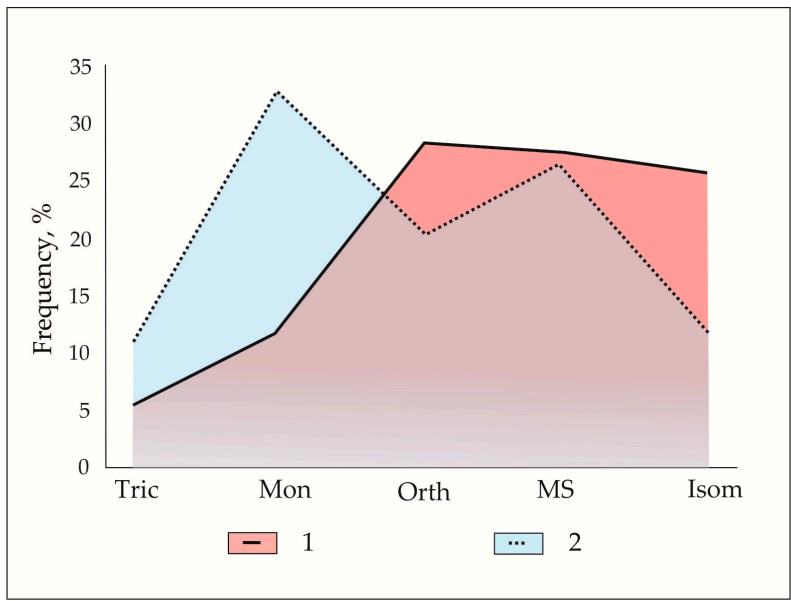

**Figure 19.** Symmetry distribution of mineral assemblages in the Hatrurim Fm larnite CM rocks (1) and the Earth lithosphere minerals (2), after [134], compared. Isom = isometric; Orth = orthorhombic; MS = medium symmetric phases (hexagonal, trigonal, tetragonal); Mon = monoclinic; Tric = triclinic.

### 5.2. Trace Elements Partitioning: Implications for TE Storage in Clinker Phases

Achtenbosch et al. (2003) [17] characterized in detail the trace-element compositions of raw meal, various types of cements (slag, blast furnace, pozzolanic, oil shales, limestone, aluminous, and mixed) and clinkers but noted the lack of respective data on oil shale and aluminous cements. This knowledge gap can be partly bridged by our results for clinker-like rocks derived from the bioproductive Maastrichtian Ghareb Fm oil shales. These CM rocks contain a number of redox-sensitive elements in average concentrations generally no lower than in cements and clinkers [17]: 174/239 vs. 50 ppm Cr, 157/126 vs. 28 ppm Ni, 12/18 vs. 9 ppm As, 800/480 vs. 540 ppm Mn, 72/50 vs. 43 ppm V, 52/34 vs. 46 ppm Cu, and 240/119 vs. 163 ppm Zn (the ratios are contents in ye'elimite to those in mayenite rocks). Natural clinker-like rocks contain U (13/11 ppm), which was not reported for cements [17], and are more depleted in some elements than the cements: 10/2 vs. 39 ppm Pb, <1 ppm vs. 1, 5, and 3 ppm Cd, Sn, and Sb, respectively. Thus, oil shale cements akin to their fly ashes [79] can be expected to contain V, Ni, Zn, Cu, which are common to organic-rich marine sediments of this type. The contents of As, Cd, Co, Mn, Mo, and U can vary broadly depending on the environment in which the oil shales were deposited. A bright example comes from the coeval Muwaqqar Fm oil shales from central Jordan with abnormal enrichment of Cd, Zn, Mo, Cr, U, Cr and Se, normal V and Ni, and very low Mn and Co [3,78]. The Jordanian spurrite CM marbles that formed after the Muwaqqar oil shales inherit this enrichment and store abundant and diverse accessories (ZnO, (Mg,Zn,Ni,Cu)O, (Ca,Cd)O, Ca–U oxides, $CaMoO_4$, (Zn,Cd)S, Cd(S,Se) CdS) [3,5–8]) which are absent from CM rocks derived from the Ghareb oil shales. The latter bear only few grains of vorlanite ($CaUO_4$), vapnikite ($Ca_3UO_6$), Cu oxides and sulfides, as well as periclase enriched in Zn and Ni. Therefore, it is reasonable to infer that the abovementioned TEs will fail to reach the threshold required for the formation of separate phases in industrial clinker production that will utilize oil shales with TE loading similar to that in the Ghareb Fm sediments.

The results for natural clinker-like rocks support the conclusion of Achtenbosch et al. (2003) [17] that the primary raw materials are the main source of Zn, Cd, Cu, Cr in cement. However, unlike most cements and concretes, the Cr-enrichment in our case is characteristic of the sediment [86,135–137].

In the clinker-like $Ca_2SiO_4$-bearing rocks of complex chemistry, both $\beta$-$Ca_2SiO_4$ and $\alpha'$-$Ca_2SiO_4$ modifications store most of the alkalis (Na, K) and P. The respective element incorporation ratios ($K_{El}$ = (El)$_{mineral}$/(El)$_{rock}$) are the highest for $\alpha'$-$Ca_2SiO_4$ ($K_K$ = 4.1; $K_P$ = 3.1 and $K_{Na}$ = 1.9).

Normalization also highlights the Na specialization of larnite ($K_{Na}$ = 1.2–1.7; $K_K$ = 0.2–1.1) instead of K specialization of flamite (Table 25).

**Table 25.** Partitioning of major and trace elements among rock-forming minerals identified in Hatrurim larnite CM rocks.

| Mineral | Element Incorporation Ratio ($K_{EL}$) | | | | | | | | | | | |
|---|---|---|---|---|---|---|---|---|---|---|---|---|
| | **Al** | **Si** | **P** | **Ca** | **Na** | **K** | **Fe** | **S** | **F** | **Sr** | **Cr** | **Ba** |
| Larnite (Y) | 0.0 | 1.4 | 0.6 | 1.2 | 1.2 | 0.1 | 0.2 | 0.0 | 0.0 | 0.3 | 0.7 | 0.0 |
| Larnite (M) | 0.0 | 1.4 | 0.9 | 1.2 | 1.7 | 1.1 | 0.1 | n.d. | 0.0 | 1.2 | 0.7 | 0.0 |
| $\alpha'$-Ca$_2$SiO$_4$(ss) | 0.0 | 1.3 | 3.1 | 1.1 | 1.9 | 4.1 | 0.1 | n.d. | 0.0 | 0.5 | 2.7 | 0.0 |
| Fluorellestadite | 0.0 | 0.6 | 5.4 | 1.1 | 0.2 | n.d. | 0.5 | 9.0 | **16.6** | 2.9 | 0.0 | 0.0 |
| Fluorapatite | 0.0 | 0.3 | **13.3** | 1.1 | 0.0 | 0.1 | 0.3 | 3.7 | **18.8** | 2.4 | 0.0 | 0.0 |
| Fluormayenite | 4.5 | 0.0 | 0.0 | 0.9 | 0.2 | 0.0 | 0.8 | 0.1 | 8.6 | 0.9 | 0.0 | 0.0 |
| Fluorkyuygenite | 4.3 | 0.0 | 0.0 | 0.8 | 0.0 | 0.2 | 0.8 | 0.2 | **11.3** | 0.4 | 0.0 | 0.0 |
| Ye'elimite | 4.5 | 0.0 | 0.1 | 0.7 | 0.4 | 0.1 | 0.5 | 4.6 | 0.0 | 1.9 | 0.0 | 0.0 |
| Barite (Y) | 0.0 | 0.0 | 0.0 | 0.0 | 0.0 | 0.0 | 1.0 | **10.4** | 0.0 | 5.4 | **130.0** | **441.0** |
| Barite (M) | 0.0 | 0.0 | 0.0 | 0.0 | 0.0 | 0.0 | 0.0 | **145.5** | 0.0 | 4.3 | 0.0 | **1314** |

$K_{El}$ = Element$_{mineral}$/Element$_{rock}$; n.d. = not detective. Y = ye'elimite–larnite rocks; M = mayenite–larnite rocks. $K_{Ti}$ = 0.3 (larnite—$\alpha'$-Ca$_2$SiO$_4$); $K_{Mg}$ = 0.1 (larnite from mayenite–larnite rocks); $K_V$ = 16.7 (fluorellestadite), $K_V$ = 22.5 (fluorapatite). Bold highlights $K_{El}$ of more than 10.

Ye'elimite, mayenite and accessory spinel have the same $K_{Al}$ = 4.3–4.5, higher than in gehlenite ($K_{Al}$ ~ 3). Opaque minerals store lesser amounts of Al ($K_{Al}$ = 1.1–1.5 for brownmillerite and magnesioferrite and $K_{Al}$ < 1 for Ti-rich members and srebrodolskite). Titanium is exclusively present in Ti-rich members of perovskite supergroup minerals $K_{Ti}$ = 44.7–94.1). Almost all Fe and Mg resides in opaques ($K_{Fe}$ = 3.7–57.5; $K_{Mg}$ = 1.6–29.7) and in periclase ($K_{Fe}$ = 23.0; $K_{Mg}$ = 81.1), which is the main host of Zn ($K_{Zn}$ = 47.3) and Ni ($K_{Ni}$ = 115.3). The incorporation ratios of Zn and Ni are also high in spinel ($K_{Zn}$ = 26.3 and $K_{Ni}$ = 24.2) and magnesioferrite from ye'elimite rocks ($K_{Zn}$ = 36.9 and $K_{Ni}$ = 43.8), but magnesioferrite from the mayenite variety is free from both Zn and Ni, most likely because periclase consumes the two elements in this rock type. As was previously reported by [18], hatrurite (Ca$_3$SiO$_5$) is another important host of Zn and Cu ($K_{Cu}$ = 3.4 and $K_{Zn}$ = 4), besides brownmillerite, whereas Ca$_2$SiO$_4$ modifications are not involved in Zn and Cu accumulation. The predominance of larnite instead of hatrurite in the Hatrurim clinker-like rocks is most likely responsible for the complex Zn partitioning between several hosts and for the formation of distinct Cu minerals even at quite low Cu contents in rocks (34 and 52 ppm in mayenite and ye'elimite rocks, respectively). Mn mainly resides in magnesioferrite (up to 1.15 wt % MnO).

The Cr budget is mostly distributed among opaque minerals: spinel ($K_{Cr}$ = 81.3), magnetite ($K_{Cr}$ = 10.3), magnesioferrite ($K_C$ = 20.9 for ye'elimite rocks), brownmillerite ($K_{Cr}$ = 13.4–18.1), shulamitite ($K_{Cr}$ = 8.1), perovskite ($K_{Cr}$ = 6.9), and nataliakulikite ($K_{Cr}$ = 2.6). Baryte–hashemite ss is the next widespread accessory that stores much Cr as (CrO$_4$)$^{2-}$-groups with $K_{Cr}$ up to 130. Our data agree with the inference from Achtenbosch et al. (2003) [17] that Cr was incorporated mostly into the $\alpha'$-Ca$_2$SiO$_4$ modification of CM rocks ($K_{Cr}$ = 2.7) rather than into β-Ca$_2$SiO$_4$ ($K_{Cr}$ = 0.7). On the other hand, the ability of Cr to form distinct minerals in the Hatrurim Fm rocks is prominent in supergene environments producing diverse endemic Cr mineralization [86,129,130,135–137].

Meanwhile, strontium, although being present in relatively high amounts (an average of ~1500–1700 ppm), does not form distinct phases but is rather dispersed in almost all types of mineral matrix: larnite ($K_{Sr}$ = 0.3 for ye'elimite and 1.2 for mayenite rock varieties, respectively); $\alpha'$-Ca$_2$SiO$_4$ modification ($K_{Sr}$ = 0.5); mayenite supergroup minerals ($K_{Sr}$ = 0.4–0.9); perovskite and nataliakulikite ($K_{Sr}$ = 1.6); ye'elimite ($K_{Sr}$ = 1.9), with the highest content in fluorapatite–fluorellestadite ss ($K_{Sr}$ = 2.4–2.9) and barite ($K_{Sr}$ = 4.4–5.4) (Tables 25 and 26).

**Table 26.** Partitioning of major and trace elements among opaque minerals identified in Hatrurim larnite CM.

| Mineral | Element Incorporation Ratio ($K_{EL}$) | | | | | | | | | | |
|---|---|---|---|---|---|---|---|---|---|---|---|
| | Ti | Al | Fe | Mg | Ca | Cr | Ni | Zn | Zr | Nb | Sr |
| Magnesioferrite (Y) | 0.3 | 1.5 | **14.5** | **16.6** | 0.0 | **20.9** | **43.8** | **36.9** | 0.0 | 0.0 | 0.0 |
| Magnesioferrite (M) | 0.3 | 1.1 | **22.7** | **26.1** | 0.0 | 0.0 | 0.0 | 0.0 | 0.0 | 0.0 | 0.0 |
| Magnetite | 2.5 | 0.2 | **57.5** | 1.6 | 0.0 | **10.3** | 0.0 | 0.0 | 0.0 | 0.0 | 0.0 |
| Spinel | 0.4 | 4.3 | **11.8** | **29.7** | 0.0 | **81.3** | **24.2** | **26.3** | 0.0 | 0.0 | 0.0 |
| Periclase | 0.0 | 0.0 | **23.0** | **81.1** | 0.0 | 0.0 | **115** | **47.3** | 0.0 | 0.0 | 0.0 |
| Perovskite | **94.1** | 0.3 | 3.7 | 0.0 | 0.8 | 6.9 | 0.0 | 0.0 | **63.4** | **24.3** | 1.6 |
| Nataliakulikite | **61.8** | 0.2 | **11.9** | 0.0 | 0.9 | 2.6 | 0.0 | 0.0 | **69.1** | **21.6** | 1.6 |
| Sharyginite | **44.7** | 0.6 | 9.5 | 0.1 | 0.8 | n.d. | 0.0 | 0.0 | **80.6** | 0.0 | 0.0 |
| Shulamitite | **46.6** | 0.9 | 8.0 | 0.1 | 0.8 | 8.1 | 0.0 | 0.0 | 0.0 | 0.0 | 0.0 |
| Brownmillerite (Y) | 1.6 | 1.0 | **10.6** | 0.4 | 0.8 | **13.4** | 0.0 | 0.0 | 0.0 | 0.0 | 0.0 |
| Brownmillerite (M) | 7.0 | 1.3 | **12.4** | 0.9 | 0.8 | **18.1** | 0.0 | 0.0 | 0.0 | 0.0 | 0.0 |
| Srebrodolskite | 0.8 | 0.5 | **12.0** | 0.3 | 0.8 | 0.2 | 0.0 | 0.0 | 0.0 | 0.0 | 0.0 |

$K_{El}$ = Element$_{mineral}$/Element$_{rock}$; n.d. = not detective. Y = ye'elimite–larnite rocks; M = mayenite–larnite rocks. $K_{Si}$ = 0.1 (perovskite), 0.2 (nataliakulikite); $K_V$ = 8.9 (magnetite). Bold highlights $K_{El}$ of more than 10.

Zirconium also reaches high concentrations in the larnite CM rocks but forms only sporadic grains of lakargiite, because most of the Zr budget is likewise dispersed in Ti-rich perovskite–supergroup minerals: sharyginite ($K_{Zr}$ = 80.6), nataliakulikite (69.1), and Fe-rich perovskite (63.4). Nataliakulikite and perovskite are also the main revealed hosts of Nb ($K_{Nb}$ = 21.7 and 24.3 respectively).

The analysis of natural CM rocks with clinker mineralogy shows that sintering of calcareous marine sediments (mainly oxidative) at T ~ 1000–1200 °C gave rise to diverse mineral assemblages with regular partitioning of major and trace elements between different solids. The fractionation was mainly controlled by the chemical properties of the elements, namely valence and steric factors. The element incorporation ratios allow quantitative comparison of the efficiency of different mineral hosts. In clinker-like assemblages, chemically complex solid solutions are typical of "omnivorous" perovskite, antiperovskite, and NaCl structural types, and especially of pseudobinary perovskite–brownmillerite series, nabimusaite-supergroup minerals, and periclase (Mg,Fe,Ni,Zn)O. Other structural types show restricted chemical flexibility. A set of trace elements, including relatively abundant Sr, Mn, Cr, V and Zn, as well as dispersed Cd, As, Pb, Sb, Mo, Co, Ga, and Nb, fail to form distinct mineral species, and thus their concentrations in the compositionally complex rocks we study must be below their threshold limits, whereas Cu, U, Ba, Se, and Ag are above these limits. For industrial clinkers, Achtenbosch et al. (2003) [17] previously noted preferable bonding of hardly volatile trace elements (Ba, Co, Cr, Cu, Ni, Zn, V) in main hydraulic clinker phases (hatrurite, belite, and Ca aluminate–ferrite). However, the absence of hatrurite from natural CM rocks and preferable incorporation of Cr, Mn, Ni, Zn into opaques and V into polyanionic compounds with low reactivity, instead of larnite, ye'elimite and mayenite with high hydration reactivity, are responsible for relatively slow TE redistribution during long-term supergene alteration of the clinker-like rocks in the Negev Desert environment.

**Supplementary Materials:** The following are available online at http://www.mdpi.com/2075-163X/9/8/465/s1, Table S1: Mineral chemistry (WDS-EDS, wt %) of larnite and $\alpha'$-Ca$_2$SiO$_4$ phase from Hatrurim larnite rocks. Table S2: Number of mineral species and mean, minimum, maximum values of bulk (in wt %) and trace-element (in ppm) compositions of Hatrurim larnite CM rocks.

**Author Contributions:** Project idea: E.V.S., S.N.K. and V.V.S.; Mineralogy and petrography: E.V.S., S.N.K. and V.V.S.; Field work: E.V.S. and V.V.S.; SEM analyses: V.A.D. and N.S.K.; EPMA analyses: E.N.N.; Formal analysis: V.A.D. and N.S.K.; XRD analysis, and preconditioning of samples, and interpretation: Y.V.S. and R.L.; Visualization, A.S.D.; Writing: E.V.S., S.N.K., and V.V.S.

**Funding:** This research was funded by the Russian Science Foundation, grant number 17-17-01056.

**Acknowledgments:** This year-long work would have been impossible without the effort, support, and encouraging interest of many our colleagues from different countries. Mottled Zone complexes have been studied through long

and fruitful collaboration with Hani Khoury from the University of Jordan (Amman, Jordan). Yevgeny Vapnik from the Ben-Gurion University (Negev, Israel) was of great help in organizing field trips to the Hatrurim Basin in 2004, 2005, and 2007 and contributed a lot to our understanding of the Hatrurim phenomenon. Our idea of this phenomenon was largely inspired by Yehoshua Kolodny from the Jerusalem University (Israel) and Avohua Burg from the Geological Survey of Israel who kindly shared their knowledge and experience. We are highly appreciative of the valuable comments and suggestions of two anonymous reviewers. Our thanks are extended to T. Perepelova from the Institute of Geology and mineralogy (Novosibirsk) for cooperation and helpful advice. Maarten Broekmans from the Geological Survey of Norway was our guide in the world of cements and introduced us to the community of cement experts. Samples are courtesy of Mikhail Murashko from the Sankt-Petersburg University. Andrew Christy turned our attention to the concept of mineralogical diversity. The study was supported by the Russian Science Foundation, grant 17-17-01056.

**Conflicts of Interest:** The authors declare no conflict of interest.

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
