# Peer review of "Mineralogical Diversity of Ca2SiO4-Bearing Combustion Metamorphic Rocks in the Hatrurim Basin: Implications for Storage and Partitioning of Elements in Oil Shale Clinkering"

_minerals, doi:10.3390/min9080465_

Round 1
Reviewer 1 Report
The manuscript by Ella Sokol with colleagues discloses the results of the first detailed study of Ca2SiO4-bearing assemblages of the Hatrurim Formation since fundamental work by Gross (1977). Pyrometamorphic suite of the Hatrurim Formation seems to be the best object for in-depth studies of clinker minerals, where all these cement phases are represented by (relatively) well-crystallized crystals. Therefore, the present work by Sokol et al. would be of interest to a broad scientific and industrial community.
I would recommend to accept this article, but after semi-minor revision, concerning the Introduction.
Although Ca2SiO4-bearing rocks constitute a signficant part of the Hatrurim Formation, a specific mineral assemblages in other rock types (melilitic, pyroxene etc.) should be breafly (~1 page) mentioned in the beginning of the introduction, so that the reader could see that the Hatrurim Fm is not composed solely of Ca2SiO4-bearing rocks. I believe, embedding of one more page into 69-page manuscript will not extend the volume of the article.
The next thing concerns the age of the Hatrurim Formation. The reported determinations give the values between 16 Ma and 200 Ka (Kolodny et al., Chemical Geology 385 (2014) 140–155) - please correct and make appropriate reference to this work.
Author Response
Manuscript ID: minerals-559757
Type of manuscript: Article
Title: Mineralogical Diversity of Ca2SiO4-bearing Combustion Metamorphic Rocks in the Hatrurim Basin: Implications for Storage and Partitioning of Elements in Oil Shale Clinkering
From: Ella Sokol, Svetlana Kokh, Victor Sharygin, Victoria Danilovsky, Yurii Seryotkin, Ruslan Liferovich, Anna Deviatiiarova, Elena Nigmatulina, Nikolay Karmanov
To: Reviewer #1.
Dear colleague,
Herewith we’re submitting the revised manuscript entitled “Mineralogical Diversity of Ca2SiO4-bearing Combustion Metamorphic Rocks in the Hatrurim Basin: Implications for Storage and Partitioning of Elements in Oil Shale Clinkering” by Ella Sokol, Svetlana Kokh, Victor Sharygin, Victoria Danilovsky, Yurii Seryotkin, Ruslan Liferovich, Anna Deviatiiarova, Elena Nigmatulina, Nikolay Karmanov to be considered for publication in Minerals.
On behalf of all coauthors of minerals-559757 I express our gratitude for your comments, suggestions, and improvements, which were extremely useful. In the new version we have considered overwhelming majority of your remarks and suggestions.
Please, find the list of specific corrections below.
Author’s answers to reviewer #1 comments.
Q.: Reviewer’s comment: I would recommend to accept this article, but after semi-minor revision, concerning the Introduction.
Although Ca2SiO4-bearing rocks constitute a signficant part of the Hatrurim Formation, a specific mineral assemblages in other rock types (melilitic, pyroxene etc.) should be breafly (~1 page) mentioned in the beginning of the introduction, so that the reader could see that the Hatrurim Fm is not composed solely of Ca2SiO4-bearing rocks. I believe, embedding of one more page into 69-page manuscript will not extend the volume of the article.
Ps.10-11, paragraph 3.3 in revised version. The paragraph 3.3. "Combustion metamorphic rocks within the Hatrurim Basin» has been renamed and completely rewritten. The brief characterization of the spatial distribution and mineral composition of the main groups of CM rocks of the Hatrurim Basin was introduced into the paragraph. We believe this additional information to be appropriate in this section. Since Introduction is focused on the issues of mineralogy and trace element composition of industrial cement clinkers, this extraneous material may break its integrity. We’d be grateful if the reviewer and editor will support our decision. (Lines 250-295 in revised version).
Q.: Reviewer’s comment: The next thing concerns the age of the Hatrurim Formation. The reported determinations give the values between 16 Ma and 200 Ka (Kolodny et al., Chemical Geology 385 (2014) 140–155) - please correct and make appropriate reference to this work.
A: The corresponding corrections have been made. The sentence was rewritten as ‘According to 40Ar/39Ar, K/Ar and 230Th–234U dating of different MZ rocks [11,66-67], the fluid flow and further gas ignition occasionally occurred between 16 and 0.2 Ma (mainly in the interval 7–0.5 Ma), which is the time span of the most active Dead Sea rifting [68,69]’. The reference on the paper Kolodny et al., Chemical Geology 385 (2014) 140–155) has been added into the list of references. (Lines 210-213 in revised version).
REFERENCES
A.: List of references has been actualized.
The authors are deeply grateful to the anonymous reviewer for goodwill and priceless help with the manuscript improvement.
24.07.2019
Dr. Svetlana Kokh

Reviewer 2 Report
Review of the paper by Sokol et al.
General comments:
The paper presents a truly exhaustive summary of the authors geochemical and mineralogical investigations of The Hatrurim rocks over the past 15 years. Therefore it represents a highly useful summary of the research done on these pyrometamorphic rocks, which in my mind is highly welcome to anyone interested in pyrometamorphism. Also the connex to industrial cement clinkers makes this paper very valuable. Therefore I have only very few comments (mostly concerning spelling and grammar, indicated in the annotated manuscript). I recommend publications of the work with only very minor revisions.
Specific comments:
One should use the term electron probe microanalyis (EPMA) instead of EMPA throughout the paper.
Page 2, line 67: Please give chemical formulae of these minerals!
Page 7, line 149: Which Cameca Camebax is it: SX50 or SX100. Please be specific!
Page 7, line 151: How many seconds for peak and background measurements!
Page 8, line 210: Better fluid injection or fluid flow..... Avoid impact!
The Tables should be uniform! Sometimes you give the analyses in lines and in other Tables you give them in columns.
The font in Table S1 is not uniform!

Author Response
Manuscript ID: minerals-559757
Type of manuscript: Article
Title: Mineralogical Diversity of Ca2SiO4-bearing Combustion Metamorphic Rocks in the Hatrurim Basin: Implications for Storage and Partitioning of Elements in Oil Shale Clinkering
From: Ella Sokol, Svetlana Kokh, Victor Sharygin, Victoria Danilovsky, Yurii Seryotkin, Ruslan Liferovich, Anna Deviatiiarova, Elena Nigmatulina, Nikolay Karmanov
To: Reviewer #2.
Dear colleague,
Herewith we’re submitting the revised manuscript entitled “Mineralogical Diversity of Ca2SiO4-bearing Combustion Metamorphic Rocks in the Hatrurim Basin: Implications for Storage and Partitioning of Elements in Oil Shale Clinkering” by Ella Sokol, Svetlana Kokh, Victor Sharygin, Victoria Danilovsky, Yurii Seryotkin, Ruslan Liferovich, Anna Deviatiiarova, Elena Nigmatulina, Nikolay Karmanov to be considered for publication in Minerals.
On behalf of all coauthors of minerals-559757 I express our gratitude for your comments, suggestions, and improvements, which were extremely useful. In the new version we have considered overwhelming majority of your remarks and suggestions.
Please, find the list of specific corrections below.
Author’s answers to reviewer #2 comments.
Editing of English language and style (fine/minor).
A: All the reviewer's corrections and suggestions about editing of English language and style have been thankfully accepted and taken into account in the revised version of the manuscript.
Specific comments:
Q.: Reviewer’s comment: One should use the term electron probe microanalyis (EPMA) instead of EMPA throughout the paper.
A.: Respective corrections were made throughout the manuscript.
Q.: Reviewer’s comment: Page 2, line 67 in initial version: Please give chemical formulae of these minerals!
A.: P.2, line 67 in revised version. Respective corrections were made.
Q.: Reviewer’s comment: Page 7, line 149: Which Cameca Camebax is it: SX50 or SX100. Please be specific!
A.: P.7, line 149 in revised version. EPMA were performed using a Camebax Micro (Cameca Ltd) microprobe. Respective correction was made.
Q.: Reviewer’s comment: Page 7, line 151: How many seconds for peak and background measurements!
A.: P.7, line 151 in revised version. The count time was 20 s (10 s peak and 10 s background). Respective correction was made.
Q.: Reviewer’s comment: Page 8, line 210: Better fluid injection or fluid flow..... Avoid impact!
A.: P.8, line 210 in revised version. The phrase “fluid impact and further gas ignition occasionally occurred…” was rewritten as ‘fluid flow and further gas ignition occasionally occurred…’
Q.: Reviewer’s comment: The Tables should be uniform! Sometimes you give the analyses in lines and in other Tables you give them in columns.
A.: The design of the tables is individual to make them as compact as possible. We find it expedient to save the initial format, and we ask the reviewer to approve our suggestion.
Q.: Reviewer’s comment: The font in Table S1 is not uniform!
A.: Table S1 is revised in accordance with reviewer’s suggestions. All respective corrections have been made.
ADDITIONAL COMMENTS ELABORATED IN SEPARATE DOCUMENT (attachments)
Q.: P. 1 , line 17 in initial version. Remark.
A.: P.1, line 17 in revised version. The phrase “geochemical image of natural Ca2SiO4-bearing…” was rewritten as ‘geochemical survey of natural Ca2SiO4-bearing…’
Q.: P. 1 , line 40 in initial version. Remark.
A.: P.1, line 40 in revised version. The phrase “… high trace-element (Zn, Cd, U, Ni, Cr) loading [1–8]” was rewritten as ‘…high trace-element (Zn, Cd, U, Ni, Cr) content [1–8]’
Q.: P. 1 , line 40 in initial version. Remark.
A.: P.1, line 42 in revised version. The phrase “clinkers are studied in 4 Ma to 100 Kyr …” was rewritten as ‘… clinkers are studied in the 4 Ma to 100 Kyr…’
Q.: P. 2 , line 45 in initial version. Remark.
A.: P.2, line 45 in revised version. The phrase “…at temperatures 1 300-1 500 °C …” was rewritten as ‘…at temperatures of 1 300-1 500 °C …’
Q.: P. 2 , line 48 in initial version. Remark.
A.: P.2, line 48 in revised version. The phrase “…as well as metallurgical wastes highly loaded by …” was rewritten as ‘…as well as metallurgical wastes highly rich in …’
Q.: P. 2 , line 49 in initial version. Remark.
A.: P.2, line 49 in revised version. The phrase “Heavy metals become bound in cement clinker…” was rewritten as ‘ Heavy metals are incorporated in cement clinker…’
Q.: P. 2 , line 51 in initial version. Remark.
A.: P.2, line 51 in revised version. The phrase “… cation substitutions in the structure …” was rewritten as ‘ … cation substitutions into the structure …’
Q.: P. 2 , line 56 in initial version. Remark.
A.: P.2, line 56 in revised version. The phrase “…an essential issue of the modern material science” was rewritten as ‘…an essential issue of modern material science’
Q.: P. 2 , line 59 in initial version. Remark.
A.: P.2, line 59 in revised version. The phrase “…crystalline clinker phases is very complicated…” was rewritten as ‘…crystalline clinker phases is mostly complicated…’
Q.: P. 2 , line 79 in initial version. Remark.
A.: P.2, lines 79-80 in revised version. The phrase “…to provide a general geochemical image of natural Ca2SiO4-bearing CM rocks…” was rewritten as ‘…to provide a geochemical survey of natural Ca2SiO4-bearing CM rocks…’
Q.: P. 2 , line 83 in initial version. Remark.
A.: P.2, line 83 in revised version. The phrase “…the contents of major and trace elements” was rewritten as ‘…the contents of their major and trace elements’
Q.: P. 6 , line 114 in initial version. Remark.
A.: P.6, line 114 in revised version. The phrase “…were analyzed by the X-ray powder diffraction…” was rewritten as ‘…were analyzed by X-ray powder diffraction …’
Q.: P. 8 , line 185 in initial version. Remark.
A.: P.6, lines 184-185 in revised version. The phrase “…the Upper Cretaceous – Low Tertiary section…” was rewritten as ‘…the Upper Cretaceous – Low Paleogene section…’
Q.: P. 8 , line 192 in initial version. Remark.
A.: P.8, line 192 in revised version. The phrase “…marly sediments (Т = 1000-1500 °С) are of two main groups…” was rewritten as ‘…marly sediments (Т = 1000-1500 °С) can be grouped into two main groups…’
Q.: P. 8 , line 193 in initial version. Remark.
A.: P.8, line 193 in revised version. The phrase “…such as hornfels…” was rewritten as ‘…such as hornfelses…’
Q.: P. 8 , line 194 in initial version. Remark.
A.: P.8, line 194 in revised version. The phrase “…diverse melt rocks …” was rewritten as ‘…diverse molten rocks …’
Q.: P. 8 , line 199 in initial version. Remark.
A.: P.8, line 199 in revised version. The phrase “The Hatrurim Fm CM rocks have been often interpreted as products …” was rewritten as ‘The Hatrurim Fm CM rocks have often been interpreted as products …’
Q.: P. 8 , line 202 in initial version. Remark.
A.: P.8, line 202 in revised version. The phrase “…CM events…” was rewritten as ‘…CM overprints…’
Q.: P. 8 , line 205 in initial version. Remark.
A.: P.8, line 205-206 in revised version. The sentence “Once being exposed to air, methane could react with atmospheric oxygen and ignite spontaneously.” was rewritten as ‘Once exposed to air, methane reacts with atmospheric oxygen and ignites spontaneously.’
Q.: P. 8 , line 207 in initial version. Remark.
A.: P.8, line 207 in revised version. The phrase “…and even local melting…” was rewritten as ‘…and even resulted in local melting…’
Q.: P. 8 , line 208 in initial version. Remark.
A.: P.8, line 208 in revised version. The phrase “…gases left imprint …” was rewritten as ‘…gases left their imprint …’
Q.: P. 8 , line 211 in initial version. Remark.
A.: P.8, lines 210-213 in revised version. The sentence was totally rewritten as ‘According to 40Ar/39Ar, K/Ar and 230Th–234U dating of different MZ rocks [11,66-67], fluid flow and further gas ignition occasionally occurred between 16 and 0.2 Ma (mainly in the interval 7–0.5 Ma), which is the time span of the most active Dead Sea rifting [68,69]’.
Q.: P. 8 , line 223 in initial version. Remark.
A.: P.8, lines 223-224 in revised version. The phrase “The Upper Cretaceous – Lower Tertiary…” was rewritten as ‘The Upper Cretaceous – Lower Paleogene…’
Q.: P. 10 , line 254 in initial version. Remark.
A.: Ps.10-11, paragraph 3.3 in revised version. The paragraph 3.3. 'Combustion metamorphic rocks within the Hatrurim Basin’ was completely rewritten according the suggestions of Reviewer #1.
lines 250-295 in revised version.
Q.: P. 53 , line 973 in initial version. Remark.
A.: P.54, line 1005 in revised version. The phrase “The Hatrurim larnite CM rocks bear…” was rewritten as ‘The Hatrurim larnite CM rocks contain…’
Q.: P. 61 , line 1173 in initial version. Remark.
A.: P.62, line 1205 in revised version. The phrase “The Cr budget is mostly dispersed among opaque minerals …” was rewritten as ‘The Cr budget is mostly distributed among opaque minerals…’
REFERENCES
A.: List of references has been actualized.
The authors are deeply grateful to the anonymous reviewer for goodwill and priceless help with the manuscript improvement.
24.07.2019
Dr. Svetlana Kokh
